

# Factorization and global symmetries in holography

Francesco Benini[1,2,3], Christian Copetti[1,2] and Lorenzo Di Pietro[2,4]

**1** SISSA, Via Bonomea 265, 34136 Trieste, Italy
**2** INFN, Sezione di Trieste, Via Valerio 2, 34127 Trieste, Italy
**3** ICTP, Strada Costiera 11, 34151 Trieste, Italy
**4** Dipartimento di Fisica, Università di Trieste,
Strada Costiera 11, 34151 Trieste, Italy

## Abstract

We consider toy models of holography arising from 3d Chern-Simons theory. In this context a duality to an ensemble average over 2d CFTs has been recently proposed. We put forward an alternative approach in which, rather than summing over bulk geometries, one gauges a one-form global symmetry of the bulk theory. This accomplishes two tasks: it ensures that the bulk theory has no global symmetries, as expected for a theory of quantum gravity, and it makes the partition function on spacetimes with boundaries coincide with that of a modular-invariant 2d CFT on the boundary. In particular, on wormhole geometries one finds a factorized answer for the partition function. In the case of non-Abelian Chern-Simons theories, the relevant one-form symmetry is non-invertible, and its "gauging" corresponds to the condensation of a Lagrangian anyon.



# 1 Introduction and summary

According to the holographic principle [1,2] and the AdS/CFT correspondence [3–5], a theory of quantum gravity in asymptotically anti-de-Sitter (AdS) spacetime is dual to a conformal field theory (CFT) placed on the conformal boundary of that spacetime. Various examples of low-dimensional gravities, however, appeared in the literature in the last few years, which are dual not to a conventional unitary quantum system, but rather to an average (with respect to some measure) over a family of quantum systems. A smoking gun for that is the lack of factorization of the partition function in the presence of disconnected boundaries. If the spacetime on which the dual boundary theory lives is of the form $X = X_1 \sqcup \ldots \sqcup X_n$ (a disconnected sum), the lack of factorization amounts to

$$\mathcal{Z}_{\text{grav}}[X] \neq \mathcal{Z}_{\text{grav}}[X_1] \cdots \mathcal{Z}_{\text{grav}}[X_n], \tag{1.1}$$

where $\mathcal{Z}_{\text{grav}}[X]$ is the gravitational partition function with fixed boundary conditions $X$. This can be associated to the presence of non-trivial (Euclidean) spacetime wormholes connecting different boundaries.

One example [6–9] is provided by two-dimensional Jackiw-Teitelboim (JT) dilaton gravity [10, 11], dual to the Sachdev-Ye-Kitaev (SYK) model [12, 13], namely a family of quantum mechanical systems with a large number of fermions, averaged over the couplings among those fermions. A complete calculation of the path integral on the gravity side was possible, confirming the duality to an ensemble average [14]. A different example [15,16] is provided by a three-dimensional Euclidean gravity defined as a sum over hyperbolic geometries, in which the small fluctuations around each geometry are described by an Abelian Chern-Simons (CS) theory. Its dual is the family of free conformal field theories of $D$ compact bosons, averaged over their Narain moduli space [17]. Variations of this example have been considered, *e.g.*, in [18–23]. The non-factorized contribution due to off-shell wormholes in 3d gravity was studied in [24].

In view of these examples, it is natural to wonder whether there exist two types of holographic correspondences — the standard one and the averaged one — and thus of gravitational theories. Since an average over unitary theories is not a unitary quantum system, at least not in the traditional sense, one asks whether the corresponding gravitational theories suffer from

some pathology that prevents them from being UV complete.[1] Various approaches to this question have been taken, *e.g.*, in [26–32].

Within the class of solvable 3d gravitational theories defined in terms of CS theories with compact gauge group, we propose an answer. We notice that those theories possess a global symmetry, generated by the topological line operators in the theory [33]. Abelian CS theories have a conventional 1-form symmetry, while in the non-Abelian case the full 1-form symmetry is non-invertible [34].[2] This is in tension with the expectation that quantum gravity should not have any global symmetry [35–39]. Thus, we propose a connection between the presence of global symmetries in the gravitational theory, and averaging in the boundary theory.[3]

Indeed, a simple way to remove the global symmetry is to gauge a suitable non-anomalous subgroup thereof. In the Abelian case this is a standard gauging of a discrete 1-form symmetry, that may be implemented by coupling the theory to dynamical gauge fields, while in the non-Abelian case one needs to resort to anyon condensation [41,42] — a sort of analog of gauging for non-invertible symmetries. In both cases, after gauging, the bulk Chern-Simons theory acquires the following pleasant properties:

1) The theory becomes trivial in the bulk, in the sense that the Euclidean partition function on any (oriented) closed 3-manifold equals 1. After all, this is what we expect from a holographic theory: the degrees of freedom only live at the boundary.

2) Given a (possibly disconnected) 2d boundary and a boundary condition, the Euclidean partition function on any 3-manifold with that boundary gives the same result (this follows from point 1). Chern-Simons theory is a generally-covariant theory [43,44], in the sense that its partition function on closed 3-manifolds does not depend on a choice of metric — but it does depend on the topology. After gauging, the theory becomes independent from the bulk topology as well.

3) The partition function of the gravitational theory is defined as the one on an arbitrary 3-manifold with the given boundary conditions (not as a sum over all possible geometries, or some subset thereof), because of point 2). Factorization in the case of disconnected boundaries immediately follows.

4) The partition function with boundary conditions equals the partition function of a single and well-defined boundary CFT. The details of such a CFT are encoded in the bulk gauge group and in the specific chosen gauging of the 1-form symmetry.

5) What we described extends to correlation functions of the boundary theory, and hinges on the fact that all lines are transparent in the bulk (because of point 1).

The properties 2) and 3) have already been observed in a much more complicated example, namely string theory on $AdS_3 \times S^3 \times T^4$ in the tensionless limit, after performing the full non-perturbative path integral [45,46]. A property analogous to 3) was also observed in [47] for the SYK model reduced to one dimension [48]. The above observations clarify the relationship between averaging and global symmetries, at least in the class of models under consideration. The bulk theory has a (possibly non-invertible) global 1-form symmetry if and only if it has non-trivial topological line operators. In that case, the partition function with fixed boundary conditions depends on the topology of the chosen bulk 3-manifold, and it is natural to define the gravitational theory as a sum over those topologies. This leads to a lack of factorization, *i.e.*, to averaging over a family of boundary CFTs. This can be rephrased in the language of

---

[1]In the case of JT gravity, it has recently been noticed [25] that a certain non-local deformation can resolve the lack of factorization and lead to well-defined quantum mechanical systems with discrete spectrum.

[2]Besides, in both cases there usually are 0-form symmetries as well.

[3]A connection between averaging in the boundary theory and the presence of global symmetries in the bulk theory (with a focus on 2d gravity) has also been proposed in [40] from a different point of view.

Coleman's $\alpha$-states [49] (see also [50, 51]) with each $\alpha$-state being the dual to a specific CFT. In our class of models, each CFT is built from the 2d left- and right-moving chiral algebras dual [44] to the bulk CS theory before gauging.[4]

After gauging, we obtain a bulk theory which is holographically dual — in the standard sense — to a specific 2d rational conformal field theory (RCFT) at a finite value of the central charge, defined in terms of its holomorphic and anti-holomorphic halves and a prescription for gluing them. The relation between 3d topological quantum field theories (TQFTs) and the holomorphic half of 2d RCFTs has been known for a long time [44, 52–56]. Besides, the formulation of modular-invariant parings between the holomorphic and anti-holomorphic sectors of a CFT in the language of TQFT, in particular in terms of Frobenius algebra objects (that we review in Section 4.1), has also been understood [57]. The latter has been more physically interpreted in terms of topological interfaces and boundary conditions in TQFT [58, 59] and in terms of anyon condensation [60].[5]

In this paper we elaborate on the interpretation in terms of gauging, showing that it makes more transparent the construction of generally-covariant and topology-independent theories — *i.e.*, of gravitational theories — in terms of CS theory.[6] As we already mentioned, exploiting the categorical formulation (in terms of Frobenius algebras) of the notion of gauging of (possibly higher-form) symmetries and its non-invertible analog, anyon condensation, is very useful in order to treat non-Abelian Chern-Simons theories. See [65–71] for interesting recent works that explore the gauging of non-invertible symmetries.

The paper is organized as follows. In Section 2 we describe the main ideas in the simplest example: a free compact scalar at rational radius. This serves as a gentle introduction to most of the relevant concepts. It is shown explicitly how to recover a well-defined dual RCFT by gauging a maximal (Lagrangian) subgroup of the one-form symmetry in the bulk, and how this gauging factorizes wormhole contributions automatically. In Section 3 we discuss the extension to more general Abelian theories with $U(1)^D$ chiral algebras. In Section 4 we discuss the extension to the non-Abelian case, for which we implement the machinery of anyon condensation. We describe how the relevant concepts on discrete gauging naturally extend to this case, by introducing the condensation of commuting Lagrangian anyons [42]. The properties of such objects are then used explicitly to show factorization of wormhole geometries and the projection to a well-defined RCFT. We conclude in Section 5 with implications and possible extensions of our results. Technical parts are collected in appendices.

## 2 The free compact scalar

The simplest example in which we can exhibit the main physical ideas of this paper is that of the 2d CFT of a free compact real scalar field. We will first review a few facts about such a CFT, and then move on to its holographic bulk description.

### 2.1 2d CFT

We consider a free massless compact real boson $\varphi$, with action[7]

$$S = \frac{1}{8\pi} \int d^2\sigma \, \partial_\mu \varphi \, \partial^\mu \varphi \, . \tag{2.1}$$

---

[4]This was conjectured in [18, 21] from the point of view of the boundary CFT.

[5]In the context of 2d gravity, it has been understood that $\alpha$-eigenstates can be related to certain types of brane insertions [61–63].

[6]Ref. [64] described a different approach to promote the correspondence between Chern-Simons theory and the holomorphic half of a RCFT, to a holographic duality to a full CFT.

[7]We follow the notation of [72] with $g = 1/4\pi$.

The theory enjoys a $U(1)$ symmetry that shifts $\varphi$, which is enhanced to $\mathfrak{u}(1) \times \mathfrak{u}(1)$ current algebra by the holomorphic currents $\partial \varphi$ and $\bar{\partial} \varphi$. We identify $\varphi \cong \varphi + 2\pi R$ (where $R$ is dimensionless). The Euclidean torus partition function can be written as

$$Z(R, \tau) = \frac{\Theta(R, \tau)}{|\eta(\tau)|^2} \,, \tag{2.2}$$

where $\tau = \tau_1 + i\tau_2$ is the modular parameter of the torus, one defines $q = e^{2\pi i \tau}$ as usual, $\eta(\tau)$ is the Dedekind eta function, and

$$\Theta(R, \tau) = \sum_{n,w \in \mathbb{Z}} \exp\left[-2\pi\tau_2 \left(\frac{n^2}{R^2} + \frac{w^2 R^2}{4}\right) + 2\pi i \tau_1 nw\right] = \sum_{n,w \in \mathbb{Z}} q^{\frac{1}{2}\left(\frac{n}{R} + \frac{wR}{2}\right)^2} \bar{q}^{\frac{1}{2}\left(\frac{n}{R} - \frac{wR}{2}\right)^2} \tag{2.3}$$

is the Siegel-Narain theta function. The integers $n, w$ are the electric charge (or momentum) and the magnetic charge (or vorticity), respectively, under the $U(1)$ symmetry. From the partition function one reads off the dimension $\Delta$ and spin $s$ of the primary operators $e^{in\varphi/R} e^{iwR\tilde{\varphi}/2}$ (where $\tilde{\varphi}$ is the dual field such that $\partial_\mu \tilde{\varphi} = \epsilon_{\mu\nu} \partial^\nu \varphi$) of the current algebra:

$$\Delta_{n,w} = \frac{n^2}{R^2} + \frac{w^2 R^2}{4} \,, \qquad s_{n,w} = nw \,. \tag{2.4}$$

Electric/magnetic T-duality acts as $R \leftrightarrow 2/R$. The left- and right-moving dimensions of primary operators are

$$h_{n,w} = \frac{1}{2}\left(\frac{n}{R} + \frac{wR}{2}\right)^2 \,, \qquad \bar{h}_{n,w} = \frac{1}{2}\left(\frac{n}{R} - \frac{wR}{2}\right)^2 \,. \tag{2.5}$$

The theory has an infinite number of primaries, whose dimensions, for generic $R$, are real numbers. However when $R^2 \in \mathbb{Q}$, the theory is a rational CFT: the dimensions are rational numbers, the chiral algebra is enhanced, and the primaries organize into a finite number of modules under the extended chiral algebra. The left-moving part of the extended chiral algebra is given by the fields with $\bar{h}_{n,w} = 0$, in other words one has to solve $n = \frac{R^2}{2} w$ for $n, w \in \mathbb{Z}$. Therefore one sets

$$\frac{R^2}{2} = \frac{p'}{p} \,, \qquad \text{with} \qquad p', p \in \mathbb{N} \text{ coprime} \,. \tag{2.6}$$

The solutions to $\bar{h}_{n,w} = 0$ are $n = p'\ell$, $w = p\ell$ with $\ell \in \mathbb{Z}$, and yield the spectrum $h = \frac{2p'p}{2}\ell^2$. This corresponds to the chiral algebra (see, *e.g.*, [73]):

$$\mathfrak{u}(1)_k \,, \qquad \text{with} \quad k = 2p'p \,. \tag{2.7}$$

Since $k \in 2\mathbb{N}$, this is a bosonic chiral algebra (all chiral fields have integer dimension).

The characters of $\mathfrak{u}(1)_k$ are

$$K_\lambda^{(k)}(\tau) \equiv \text{Tr}\, q^{L_0 - \frac{c}{24}} = \frac{1}{\eta(\tau)} \sum_{n \in \mathbb{Z}} q^{\frac{(kn+\lambda)^2}{2k}} \,, \tag{2.8}$$

for $\lambda = 0, \ldots, k-1$ (defined modulo $k$).[8] The modular transformations of the characters can be written as

$$K_\lambda^{(k)}(\gamma \cdot \tau) = \sum_{\mu=0}^{k-1} M_{\lambda\mu}^{(\gamma)} K_\mu^{(k)}(\tau) \,, \tag{2.9}$$

---

[8]The characters for $\lambda$ and $-\lambda$ (corresponding to charge conjugate representations) are identical, however they could be distinguished by considering the refined characters with a fugacity for the $U(1)$ symmetry. For the sake of simplicity, we will not do that here.

where $\gamma \in SL(2,\mathbb{Z})$ and $\gamma \cdot \tau$ is its action on the modular parameter $\tau$. One finds

$$
\begin{aligned}
T &: \tau \mapsto \tau + 1 \,, & T_{\lambda\mu} &= e^{2\pi i \left[\frac{\lambda^2}{2k} - \frac{1}{24}\right]} \delta_{\lambda\mu} \,, \\
S &: \tau \mapsto -\frac{1}{\tau} \,, & S_{\lambda\mu} &= \frac{1}{\sqrt{k}} e^{-2\pi i \frac{\lambda\mu}{k}} \,.
\end{aligned}
\tag{2.10}
$$

They satisfy $S^2 = C$ (where $C : \lambda \mapsto -\lambda$ is charge conjugation), $SS^\dagger = S^4 = \mathbb{1}$, and $(ST)^3 = C$.

The torus partition function can be written in terms of characters, by correctly combining left- and right-moving fields into physical fields. In order to construct the pairing of primaries, one finds integers $r_0, s_0$ such that

$$
pr_0 - p's_0 = 1
\tag{2.11}
$$

and then defines

$$
\omega = pr_0 + p's_0 \mod k \,.
\tag{2.12}
$$

Since $k \in 2\mathbb{N}$, it follows that $\omega^2 = 1 \mod 2k$. Therefore the map

$$
\omega : \lambda \mapsto \omega\lambda \,, \qquad \text{for} \qquad \lambda \in \mathbb{Z}_k
\tag{2.13}
$$

provides an involution on the set of integrable representations of the chiral algebra. This involution is an outer automorphism of the algebra that preserves the chiral dimension $h(\lambda) = \frac{\lambda^2}{2k}$ modulo 1, and therefore the $S$-matrix. With some algebra, one shows that[9]

$$
Z(R, \tau) = \sum_{\lambda=0}^{k-1} K_\lambda^{(k)}(\tau) \, \overline{K_{\omega\lambda}^{(k)}(\tau)} \,.
\tag{2.14}
$$

In particular, the pairing of representations $(\lambda, \omega\lambda)$ guarantees that all physical fields have integer spin, because the left and right dimensions agree modulo 1. Notice that for $R^2 \in 2\mathbb{N}$ one gets $k = R^2$ and $\omega = 1$, which yields a diagonal partition function. In the dual case $R^2 = \frac{4}{k}$ with $k \in 2\mathbb{N}$, one gets $k = 4/R^2$ and $\omega = -1$: in this case the partition function is non-diagonal, and the pairing of representations is through charge conjugation $\lambda \leftrightarrow k - \lambda$.

The partition function (2.14) is modular invariant. This is guaranteed by the expressions of $T, S$ in (2.10), as long as multiplication by $\omega$ is a group automorphism of $\mathbb{Z}_k$ that preserves the quadratic function

$$
q(\lambda) = \frac{\lambda^2}{2k} \mod 1
\tag{2.15}
$$

(the chiral dimension mod 1). This is equivalent to $\omega^2 = 1 \mod 2k$. In that case, the symmetric bilinear form $q(\lambda + \mu) - q(\lambda) - q(\mu) = \frac{\lambda\mu}{k} \mod 1$ is preserved as well.

The equations (2.6), (2.7), (2.11) and (2.12) provide a map from $R^2 \in \mathbb{Q}$ to the pair $(k, \omega)$. The inverse map (assuming that $\omega$, which is an element of the multiplicative group $\mathbb{Z}_k^*$ of integers mod $k$, preserves (2.15)) can be constructed as follows. If we restrict to momentum modes (i.e., to modes with equal left and right charge), then the minimal dimension is $h = \bar{h} = \frac{1}{2R^2}$. Momentum modes are solutions to $\lambda + kn = \omega\lambda + k\bar{n}$ for $n, \bar{n} \in \mathbb{Z}$, and given a solution for $\lambda$ there always is a solution in which $n = 0$. We conclude that $\frac{1}{2R^2}$ is equal to the minimal value of $\frac{\lambda^2}{2k}$ over those solutions, or in other words,

$$
R^2 = \frac{k}{\lambda^2} \,,
\tag{2.16}
$$

---

[9]Given the $SL(2,\mathbb{Z})$ matrix $S = \left(\begin{smallmatrix} p & p' \\ s_0 & r_0 \end{smallmatrix}\right)$, one performs the redefinition $\left(\begin{smallmatrix} N \\ -\bar{\ell} \end{smallmatrix}\right) = S\left(\begin{smallmatrix} n \\ w \end{smallmatrix}\right)$. The inverse transformation is $S^{-1} = \left(\begin{smallmatrix} r_0 & -p' \\ -s_0 & p \end{smallmatrix}\right)$. Then (2.2) takes the form $Z = \frac{1}{|\eta|^2} \sum_{N, \bar{\ell} \in \mathbb{Z}} q^{\frac{1}{2k}(N)^2} \bar{q}^{\frac{1}{2k}(k\bar{\ell} + \omega N)^2}$. Now rewrite $\sum_{N \in \mathbb{Z}}$ as $\sum_{\lambda=0}^{k-1} \sum_{\ell \in \mathbb{Z}}$ setting $N = k\ell + \lambda$. Shifting $\bar{\ell} \to \bar{\ell} - \omega\ell$, one obtains (2.14).

where $\lambda > 0$ is the minimal positive integer solution to

$$\lambda(\omega - 1) = 0 \mod k \, . \tag{2.17}$$

In particular notice that, for a given chiral algebra determined by $k$, there is a family of RCFTs — one for each choice of $\omega$ — which might be constructed from it.

**Higher genus.** The complex structure of a Riemann surface $\Sigma$ of genus $g \geq 1$ is described by a period matrix $\Omega$:[10] a symmetric $g \times g$ matrix, that we split into real and imaginary part as $\Omega_{ij} = x_{ij} + i\, y_{ij}$, with positive-definite imaginary part, $\mathbb{Im}\,\Omega = y > 0$. The Euclidean partition function on $\Sigma$ is

$$Z_\Sigma(R, \Omega) = \frac{\Theta(R, \Omega)}{\Phi} \, , \tag{2.18}$$

where

$$\Theta(R, \Omega) = \sum_{n, w \in \mathbb{Z}^g} \exp\left[ -2\pi\, y_{ij}\left( \frac{n^i n^j}{R^2} + \frac{w^i w^j R^2}{4} \right) + 2\pi i\, x_{ij} n^i w^j \right] \tag{2.19}$$

is the Siegel-Narain theta function. The denominator $\Phi$ can be written as [74]

$$\Phi = \left| \det{}'_\Gamma \bar{\partial}_0 \right| \, , \tag{2.20}$$

where $\bar{\partial}_0$ is the Dolbeault operator mapping functions to $(0, 1)$-forms, with zero-modes removed, while $\Gamma$ indicates a certain regularization (we provide more details in Appendix A). For $g > 1$, $\Phi$ suffers from the conformal anomaly and holomorphic factorization fails. However both $Z_\Sigma(R, \Omega)$, $\left( \det \mathbb{Im}\,\Omega \right)^{1/2} \Theta(R, \Omega)$, and $\left( \det \mathbb{Im}\,\Omega \right)^{1/2} \Phi$ are modular invariant under the $Sp(2g, \mathbb{Z})$ action of the mapping class group.

The theta function can be rewritten as

$$\Theta = \sum_{n, w \in \mathbb{Z}^g} \exp\left[ \pi i \left( \frac{n}{R} + \frac{wR}{2} \right)^\mathsf{T} \Omega \left( \frac{n}{R} + \frac{wR}{2} \right) - \pi i \left( \frac{n}{R} - \frac{wR}{2} \right)^\mathsf{T} \Omega^* \left( \frac{n}{R} - \frac{wR}{2} \right) \right] . \tag{2.21}$$

In the rational case $R^2 \in \mathbb{Q}$, with $(k, \omega)$ defined as before, following the same steps as in footnote 9, one obtains

$$Z_\Sigma = \sum_{\lambda \in (\mathbb{Z}_k)^g} \frac{1}{\Phi} \sum_{\ell \in \mathbb{Z}^g} \exp\left[ \frac{2\pi i}{2k} \left( k\ell + \lambda \right)^\mathsf{T} \Omega \left( k\ell + \lambda \right) \right] \sum_{\bar{\ell} \in \mathbb{Z}^g} \exp\left[ -\frac{2\pi i}{2k} \left( k\bar{\ell} + \omega\lambda \right)^\mathsf{T} \Omega^* \left( k\bar{\ell} + \omega\lambda \right) \right] . \tag{2.22}$$

This is a sum over higher-genus conformal blocks labelled by $\lambda$.

## 2.2 $U(1)_k \times U(1)_{-k}$ Chern-Simons theory

Two-dimensional current algebra is intimately related to three-dimensional Chern-Simons theory [44]. In particular [56], the quantization of Chern-Simons theory with gauge group $G$ at level $k$ on $D_2 \times \mathbb{R}$ (where $D_2$ is a two-dimensional spatial ball, or disk, while $\mathbb{R}$ is time) with holomorphic boundary conditions yields, when $G$ is a connected and simply-connected simple Lie group, the chiral WZW model $G_k$ [75], *i.e.*, the Hilbert space on $D_2$ is the Kac-Moody current algebra. Adding a Wilson line in an integral representation $\lambda$ along $\mathbb{R}$ through the disk, yields a Hilbert space which is the representation $\lambda$ of the current algebra [56]. If $G$ is not simply-connected, one obtains an extended chiral algebra [55]. In particular, $U(1)_k$

---

[10]Given a canonical basis $\{A_i, B_j\}$ of 1-cycles on $\Sigma$ with intersection numbers $(A_i, A_j) = (B_i, B_j) = 0$, $(A_i, B_j) = \delta_{ij}$, and a basis of holomorphic 1-forms $\omega_i$ such that $\oint_{A_i} \omega_j = \delta_{ij}$, one defines $\Omega_{ij} = \oint_{B_i} \omega_j$.

Cherns-Simons theory (with $k \in 2\mathbb{N}$) yields the $\mathfrak{u}(1)_k$ chiral algebra and its representations $\lambda \in \mathbb{Z}_k$.

Indeed, $U(1)_k$ CS theory is an Abelian topological quantum field theory (TQFT) with $k$ line operators, labelled by $\lambda \in \mathbb{Z}_k$. Under fusion, the lines reproduce the group structure of $\mathbb{Z}_k$, which is the one-form symmetry of the theory (in the terminology of [33]). The lines are the unitary defect operators that implement $\mathbb{Z}_k$ one-form symmetry transformations.

Euclidean path integrals of $U(1)_k$ CS theory on solid tori and higher-genus handlebodies, with holomorphic (conformal) boundary conditions on the boundary Riemann surface $\Sigma$, and with Wilson line insertions along the non-contractible cycles, have been addressed with a variety of approaches, see for instance [56, 76–79]. The boundary conditions fix the antiholomorphic part of the pull-back of the connection to the boundary, introducing a dependence on the complex structure of $\Sigma$ [56]:

$$A_{\bar{z}} d\bar{z} = \partial_{\bar{z}} \chi \, d\bar{z} + i\pi u (\mathbb{Im}\,\Omega)^{-1} \bar{\omega} \,. \tag{2.23}$$

Here $z$ is a local complex coordinate on $\Sigma$, $\omega_I$ for $I = 1,\ldots,g$ are a basis of holomorphic differentials that define $\Omega$, $\chi$ is a periodic function on $\Sigma$, and the vector $u_I$ of complex numbers fixes the harmonic part of the differential $A_{\bar{z}} d\bar{z}$. Then, with the insertion of Wilson lines parametrized by $\lambda \in (\mathbb{Z}_k)^g$ along the non-contractible cycles of the handlebody, the Euclidean path integral reads [79]:[11]

$$Z[A_{\bar{z}}, \lambda, \Omega] = \frac{(\det'\Delta_0)^{\frac{3}{4}}}{(\det'\Delta_1)^{\frac{1}{4}}} \, e^{\frac{k\pi}{2} u (\mathbb{Im}\,\Omega)^{-1} u} \, e^{\frac{k}{2\pi} \int_\Sigma d^2x \, \partial_z \chi \, \partial_{\bar{z}} \chi} \, \theta \begin{bmatrix} \lambda/k \\ 0 \end{bmatrix} (ku, k\Omega) \,. \tag{2.24}$$

Here

$$\theta \begin{bmatrix} a \\ b \end{bmatrix} (u, \Omega) = \sum_{n \in \mathbb{Z}^g} e^{\pi i (n+a)^\mathsf{T} \Omega (n+a) + 2\pi i (n+a)^\mathsf{T} (u+b)} \,, \qquad \text{with} \quad u \in \mathbb{C}^g \,, \quad a, b, \in \mathbb{R}^g \,. \tag{2.25}$$

On the other hand, $\Delta_0$ and $\Delta_1$ are Laplacian operators acting on scalars and one-forms, respectively. As long as we are dealing with only half of the theory, here $U(1)_k$, the definition of the determinants in (2.24) requires some care, reflecting the obstruction to holomorphic factorization in the boundary theory for $g > 1$. However, we are eventually interested in the $U(1)_k \times U(1)_{-k}$ CS theory with holomorphic/antiholomorphic boundary conditions for the two factors. As discussed in [16], in that case the determinants combine to give

$$\frac{(\det'\Delta_0)^{\frac{3}{2}}}{(\det'\Delta_1)^{\frac{1}{2}}} = \frac{1}{\Phi} \,. \tag{2.26}$$

The $\Phi$ appearing here is equal to the one defined in (2.20), in particular it gives $|\eta(\tau)|^{-2}$ in the case of $g = 1$, see Appendix A for more details. For the sake of simplicity, we simply take holomorphic boundary conditions $A_{\bar{z}} = 0$ for $U(1)_k$, and similarly antiholomorphic boundary conditions for $U(1)_{-k}$. With the insertion of Wilson lines $\lambda, \mu \in (\mathbb{Z}_k)^g$ for the two group factors, respectively, we obtain the Euclidean path-integral on a genus-$g$ handlebody:[12]

$$Z = \begin{cases} 1, & g = 0 \,, \\ K_\lambda^{(k)}(\tau) \overline{K_\mu^{(k)}(\tau)}, & g = 1 \,, \\ \dfrac{1}{\Phi} \left( \displaystyle\sum_{\ell \in \mathbb{Z}^g} e^{\frac{\pi i}{k} (k\ell+\lambda)^\mathsf{T} \Omega (k\ell+\lambda)} \right) \left( \displaystyle\sum_{\bar\ell \in \mathbb{Z}^g} e^{\frac{\pi i}{k} (k\bar\ell+\mu)^\mathsf{T} \Omega (k\bar\ell+\mu)} \right)^*, & g \geq 1 \,. \end{cases} \tag{2.27}$$

---

[11]We adopt here a slightly different regularization than in [79], more natural for the $U(1)_k \times U(1)_{-k}$ theory we discuss below, leading to a non-holomorphic factor in front of the Euclidean path integral.

[12]The ambiguity due to the 2d Euler counterterm is fixed by setting to 1 the result for $g = 0$, after which the normalization of the $g > 1$ partition functions is also fixed.

In the case $g = 1$ of a solid torus, we have expressed the path integral in terms of $\mathfrak{u}(1)_k$ characters (2.8).

The expressions above are not modular invariant, so a further prescription is needed to get a candidate physical RCFT dual. One possibility is to sum the expressions above over their modular images. This imitates the sum over handlebodies [80] for three-dimensional gravity. Such a setup has been studied, *e.g.*, in [18, 21, 81] with the conclusion that it gives rise to an *ensemble average* over different RCFTs with the same chiral algebras (*i.e.*, over different choices of $\omega$ in the language of Section 2.1). In the remainder of this section, instead, we will introduce an alternative procedure which selects a member of the ensemble and gives rise to a single dual RCFT.

## 2.3   Gauging a $\mathbb{Z}_k$ one-form symmetry

The $U(1)_k \times U(1)_{-k}$ CS theory has 1-form global symmetry $\mathbb{Z}_k \times \mathbb{Z}_k$. If we are going to regard this theory, being generally covariant in the language of [43,44], as a sort of three-dimensional theory of gravity, along the lines of [16], we encounter a tension with the expectation that theories of gravity should not have global symmetries (see, *e.g.*, [35–38]). We might hope that, if we remove the global symmetry, the behavior of the theory as a unitary quantum system improves.

A simple way to remove a global symmetry is to gauge it, that is, to couple it to dynamical gauge fields. Therefore, we would like to gauge the 1-form symmetry group, or a subgroup thereof. From the point of view of the spectrum of lines, this gauging was understood in [55]. The 1-form symmetry subgroup we gauge is given by a subset $\mathcal{A}$ of the simple lines. Such a subgroup is gaugeable, *i.e.*, its 't Hooft anomaly vanishes [33], if and only if the lines in $\mathcal{A}$ are mutually transparent — *i.e.*, they have trivial mutual braiding — and moreover they have integer spin.[13] The 1-form symmetry group is necessarily Abelian [33], and for Abelian lines the braiding is completely characterized by their spins:

$$B(\lambda, \mu) = e^{2\pi i \left[h(\lambda+\mu)-h(\lambda)-h(\mu)\right]} \,. \tag{2.28}$$

Here $h(\lambda)$ mod 1 is the spin of the line $\lambda$, which is equal to the chiral dimension mod 1 of any field in the integrable representation $\lambda$ of the associated 2d chiral algebra. Therefore, it is enough to require that all lines in the Abelian subgroup $\mathcal{A}$ have integer spin.

The algorithm in [55] tells us what the spectrum of lines after gauging is. In an Abelian theory, if we gauge a 1-form subgroup of order $p$, we reduce the number of lines by a factor of $p^2$. Thus, if we start from $\mathbb{Z}_k \times \mathbb{Z}_k$ and we want to get rid of all lines but the identity, then we should gauge a subgroup of order $k$. This is a *Lagrangian subgroup*, *i.e.*, a subgroup of lines with integer spin and whose order squares to the order of the group. Let us assume that we gauge a $\mathbb{Z}_k$ subgroup generated by the line $(1, \omega)$ for some $\omega \in \mathbb{Z}_k$.[14] The condition that the generator has integer spin boils down to

$$\omega^2 = 1 \quad \mod 2k \,. \tag{2.29}$$

We have already encountered this condition in Section 2.1. It implies that $\omega, k$ are coprime, and that such an $\omega \in \mathbb{Z}_k^*$ defines an involutive group automorphism of $\mathbb{Z}_k$ which pairs the lines of $U(1)_k$ with those of $U(1)_{-k}$. We will indicate the subgroup generated by $(1, \omega)$ as $\mathbb{Z}_k^{(\omega)}$. All

---

[13]The condition on the spin of the lines guarantees that, if we start from a bosonic theory (*i.e.*, a theory that does not depend on a spin structure), after gauging the theory is still bosonic. It is possible to gauge $\mathcal{A}$ even when some lines in it have semi-integer spin, however at the expense of making the theory fermionic (*i.e.*, dependent on a choice of spin structure).

[14]This is not the most general case. We discuss some generalizations in Section 3.2.

its lines have integer spin, and thus it can be gauged. This leads us to study

$$\mathcal{T}[k,\omega] \equiv \frac{U(1)_k \times U(1)_{-k}}{\mathbb{Z}_k^{(\omega)}} \qquad \text{Chern-Simons theory.} \qquad (2.30)$$

In particular, we are interested in its Euclidean partition and correlation functions.

Gauging of the 1-form symmetry can be described in two equivalent ways. The conventional point of view is that we couple the symmetry to an external background 2-form gauge field, and then make the latter dynamical. For a discrete (Abelian) symmetry $\mathcal{A}$, the second step is particularly simple because it reduces to a discrete sum over 1-form bundles. On an oriented closed three-manifold $M$, this is a sum over the singular cohomology group $H^2(M; \mathcal{A})$. The other point of view is that flat bundles can be engineered by inserting networks of symmetry defects, which here are the lines in $\mathcal{A}$. For Abelian lines, the networks can be broken into simple lines wrapping the non-contractible cycles of $M$. Therefore the sum over 1-form bundles can be expressed as a sum over the insertion of lines along the homology cycles of $M$, namely over $H_1(M; \mathcal{A}) \cong H^2(M; \mathcal{A})$.

We fix the normalization. Let $G = \widehat{\mathcal{A}}$ be the Pontriagyn dual to $\mathcal{A}$, namely the Abelian group of linear functions from $\mathcal{A}$ to $\mathbb{R}/\mathbb{Z}$, which is isomorphic to $\mathcal{A}$. For an oriented closed 3-manifold $M$, gauging of a discrete Abelian 0-form symmetry $G$ gives the partition function

$$Z^{\text{0-form gauged}} = \frac{1}{\left|H^0(M;G)\right|} \sum_{\alpha \in H^1(M;G)} Z[\alpha], \qquad (2.31)$$

where $Z[\alpha]$ is the partition function of the original theory coupled to the bundle $\alpha$. This could be equivalently written as a sum over the insertion of codimension-1 symmetry defects $\alpha \in H_2(M; G)$. The normalization is standard [82]: for each bundle, we divide by the number of automorphisms of that bundle, which, for $G$ Abelian, does not depend on the bundle and is the number of global gauge transformations, *i.e.*, transformations that are constant on each connected component of $M$. Similarly, gauging of a discrete (Abelian) 1-form symmetry $\mathcal{A}$ gives the partition function

$$Z^{\text{1-form gauged}} = \frac{\left|H^0(M;\mathcal{A})\right|}{\left|H^1(M;\mathcal{A})\right|} \sum_{\lambda \in H^2(M;\mathcal{A})} Z[\lambda]. \qquad (2.32)$$

This could be written as a sum over the insertion of line symmetry defects $\lambda \in H_1(M; \mathcal{A})$. We have divided by the number of global 1-form gauge transformations, but we have removed overcounting by 0-form gauge transformations of 1-form gauge transformations.

The normalizations in (2.31) and (2.32) are compatible. When we gauge a discrete Abelian 0-form symmetry $\widehat{\mathcal{A}}$, the new theory acquires a 1-form symmetry $\mathcal{A}$, and gauging the latter we get back the original theory (as in [83]). The simple identity

$$\frac{\left|H^0(M;\mathcal{A})\right|}{\left|H^1(M;\mathcal{A})\right|} \sum_{\beta \in H^2(M;\mathcal{A})} e^{2\pi i \int_M \gamma \cup \beta} \frac{1}{\left|H^0(M;\widehat{\mathcal{A}})\right|} \sum_{\alpha \in H^1(M;\widehat{\mathcal{A}})} e^{2\pi i \int_M \alpha \cup \beta} Z[\alpha] = Z[\gamma], \qquad (2.33)$$

where $\gamma \in H^1(M; \widehat{\mathcal{A}})$, expresses this fact. Here $\cup$ is the cup product, the bilinear form $\widehat{\mathcal{A}} \times \mathcal{A} \to \mathbb{R}/\mathbb{Z}$ is the natural one, and we used that $|H^1| = |H^2|$.

For holographic applications, we need a generalization of the gauging formula (2.32) to oriented three-manifolds $M$ with boundary, with Dirichelet boundary conditions $b$. The formula is the following:[15]

$$Z_b^{\text{1-form gauged}} = \left|\frac{H^0(M, \partial M \smallsetminus P)}{H^1(M, \partial M \smallsetminus P)}\right| \sum_{\substack{a \in H^2(M, \partial M \smallsetminus P) \\ i^*(a)=b}} Z[a]. \qquad (2.34)$$

---

[15]We are grateful to Pavel Putrov for explaining this formula to us. See also Section 5 of [84].

Here $\partial M$ is the two-dimensional boundary of $M$, $P$ is a set made of one point in each connected component of $\partial M$, $H^*(M, B)$ is the singular cohomology of $M$ relative to $B$,[16] $i$ is the inclusion map $\partial M \overset{i}{\hookrightarrow} M$, $b \in H^2(\partial M)$ is the boundary condition, while $Z[a]$ is the partition function of the original theory with background $a$. All cohomology groups take values in $\mathcal{A}$, that we have kept implicit. We provide a derivation of this formula in Appendix C. It is convenient to use homology — as opposed to cohomology — classes, since the former have a direct interpretation as line insertions. Using Poincaré duality, we obtain:

$$Z_b^{\text{1-form gauged}} = \left| \frac{H_3(M)}{H_2(M)} \right| \sum_{\substack{a \in H_1(M,P) \\ \partial a = b}} Z[a] \,, \tag{2.35}$$

where $H_*(M, P)$ is singular homology of $M$ relative to $P$. Here $b \in H_0(P)$, and the map $H_1(M, P) \overset{\partial}{\to} H_0(P)$ is in the long exact sequence for the pair $(M, P)$. Notice that $H_1(M, P)$ is larger than $H_1(M)$ because it includes relative 1-cycles going from one connected component of the boundary to another, however these extra 1-cycles are precisely fixed by the boundary conditions.

The theory $\mathcal{T}$ in (2.30) is completely trivial on closed three-manifolds. As we explained, it only contains a single transparent line. As a consequence, the Hilbert space is one-dimensional on any closed spatial two-manifold and thus $\mathcal{T}$ is an invertible TQFT. Besides, the partition function $Z_{\mathcal{T}}[M] = 1$ on any (oriented) closed three-manifold $M$, irrespective of its topology [86].[17] We interpret this as the fact that $\mathcal{T}$ is not just a generally-covariant theory: rather, it is independent both of a local metric *and* of the global topology, and therefore it behaves as a full-fledged theory of gravity without the need of extra sums over topologies. The total Hilbert space on closed manifolds is referred to as the "baby universes" Hilbert space [49–51], and in $\mathcal{T}$ it is one-dimensional. As we discuss in the following sections, things become more interesting in the presence of boundaries.

## 2.4 Partition functions and factorization

We proceed to compute the Euclidean partition function of the gauged Chern-Simons theory $\mathcal{T}[k, \omega] \equiv [U(1)_k \times U(1)_{-k}]/\mathbb{Z}_k^{(\omega)}$ on general (oriented) three-manifolds with boundary, and compare with 2d CFT. We focus on *factorization*: we show that the partition function of $\mathcal{T}$ on a three-manifold whose boundary is made of one or more connected components, is equal to the product of partition functions on handlebodies, each one having as boundary one of the connected components. In particular, the partition function of $\mathcal{T}$ on any oriented three-manifold with a given (connected or disconnected) boundary is the same, and it only depends on the boundary.

We will always take holomorphic and antiholomorphic boundary conditions for the gauge fields of the two factors $U(1)_k \times U(1)_{-k}$, respectively. We also define the so-called *algebra object*

$$\mathcal{A} = \bigoplus_{\lambda \in \mathbb{Z}_k} (\lambda, \omega\lambda) \,, \tag{2.36}$$

as the set of lines which generate the group to be gauged. Note that $\mathcal{A}$ is a line itself. To gauge the 1-form symmetry $\mathbb{Z}_k^{(\omega)}$ we use (2.35) with trivial boundary conditions, $b = 0$ (we discuss more general boundary conditions in Section 4.3). The sum over line insertions can be viewed as the insertion of the single line $\mathcal{A}$ on each generator of $H_1(M)$. With some abuse of notation

---

[16]See, *e.g.*, [85] for the definitions of relative singular cohomology and homology.

[17]In general, the invertible TQFT could be a multiple of the $(E_8)_1$ TQFT with $c_- = 8$. In our construction, as long as the left and right part before gauging are related by orientation reversal, the invertible TQFT is completely trivial. In the general case, it can be made trivial by stacking with $(E_8)_1$.

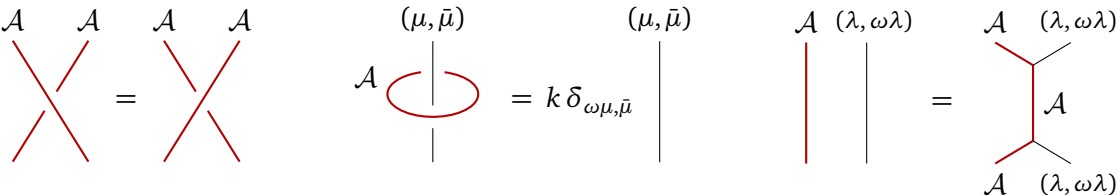

Figure 1: Basic properties of the Abelian Lagrangian algebra object $\mathcal{A}$.

we indicate as $\mathcal{A}$ both the 1-form symmetry group $\mathbb{Z}_k^{(\omega)}$ to be gauged, and the set of lines in CS theory that implement it.

The line $\mathcal{A}$ has the following properties:

1) $\mathcal{A}$ is transparent with respect to itself (this is the anomaly-free condition).

2) Wrapping $\mathcal{A}$ around a generic line $(\mu, \bar{\mu})$ is equal to $k \, \delta_{\omega\mu,\bar{\mu}}$ times that line, *i.e.*, it is proportional to a projector onto $\mathcal{A}$.

3) Parallel fusion of $\mathcal{A}$ with a line $(\lambda, \omega\lambda)$ in $\mathcal{A}$ gives back $\mathcal{A}$.

These properties are represented in Figure 1. We reviewed in (2.27) the partition function of $U(1)_k \times U(1)_{-k}$ CS theory on handlebodies with line insertions along the non-contractible cycles. Here we will first discuss a few simple examples of factorization, and argue for the general case at the end. We will sometimes use latin letters $a, b, c, \dots$ to denote the composite lines $(\mu, \bar{\mu})$ of the full theory. The orientation reversal of $\lambda$ will be denoted as $\check{\lambda}$.

**Genus $g = 0$.** We introduce the notation $\mathrm{S}\Sigma_g$ for a genus $g$ handlebody. Notice that $\mathrm{S}\Sigma_0 = \mathrm{S}S^2 = D_3$ is a solid ball, while $\mathrm{S}\Sigma_1 = \mathrm{S}T^2 = D_2 \times S^1$ is a solid torus. We also introduce the geometries $X_n = S^3 \setminus n\mathring{D}_3$: their boundaries $\partial X_n$ consist of $n$ disconnected $S^2$'s. Note that $X_1 = D_3$ is a solid ball, while $X_2 = S^2 \times I$ is a spherical cylinder, or "Euclidean wormhole", where $I = [0,1]$ is a closed interval. The homology groups are

$$H_3(X_n; \mathcal{A}) = 0 \,, \qquad H_2(X_n; \mathcal{A}) = \mathcal{A}^{n-1} \,, \qquad H_1(X_n, P; \mathcal{A})\big|_{b=0} = 0 \,. \qquad (2.37)$$

Note that $H_1(X_n, P; \mathcal{A}) = \mathcal{A}^{n-1}$, but the boundary condition $b = 0$ reduces it to zero. This implies that on this class of geometries the gauging is trivial, and it can only affect the normalization.

The partition function of $\mathcal{T}$ on a solid ball $X_1 = D_3$, according to (2.27), is

$$Z_{\mathcal{T}}[D_3] = Z_{\mathrm{CS}}[D_3] = 1 \,, \qquad (2.38)$$

in agreement with the CFT result.

Next, consider the cylinder $X_2 = S^2 \times I$. In order to compute $Z_{\mathrm{CS}}[S^2 \times I]$ we exploit the completeness relation. The Hilbert space on $S^2$ is one-dimensional and the unique state is produced by the path integral on $D_3$, therefore we can split the cylinder $S^2 \times I$ in two disks, up to a normalization factor $c_0$. In order to compute $c_0$, we split the cylinder twice: $Z_{\mathrm{CS}}[S^2 \times I] = c_0 Z_{\mathrm{CS}}[D_3] Z_{\mathrm{CS}}[D_3] = c_0^2 Z_{\mathrm{CS}}[D_3] Z_{\mathrm{CS}}[S^3] Z_{\mathrm{CS}}[D_3]$. This implies

$$c_0 = Z_{\mathrm{CS}}[S^3]^{-1} = S_{00}^{-1} = \mathcal{D} \,, \qquad (2.39)$$

where $\mathcal{D} = \sqrt{\sum_{a \in \mathcal{C}} d_a^2} = k$ is the total quantum dimension of the TQFT. For $\mathcal{A}$ Lagrangian, $|\mathcal{A}| = c_0$. Since $Z_{\mathcal{T}}[S^2 \times I] = \frac{1}{|\mathcal{A}|} Z_{\mathrm{CS}}[S^2 \times I]$, we conclude that

$$Z_{\mathcal{T}}[S^2 \times I] = Z_{\mathcal{T}}[D_3] Z_{\mathcal{T}}[D_3] = 1 \,. \qquad (2.40)$$

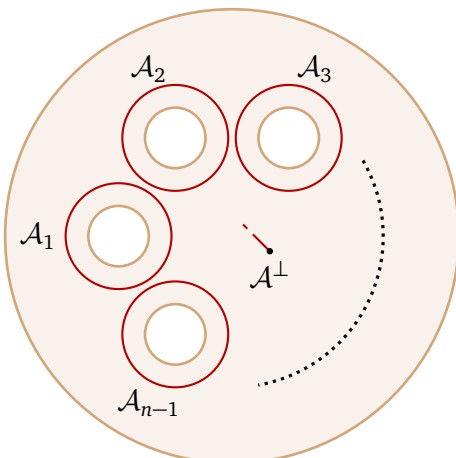

Figure 2: Line configuration for the gauging on $Y_n$. We draw $S^2 \setminus n\mathring{D}_2$, while the transverse $S^1$ is implicit. The $n-1$ lines denoted as $\mathcal{A}$ wrap one of the boundary circles, while $\mathcal{A}^\perp$ (drawn as a broken line) wraps the transverse $S^1$.

The partition function of $\mathcal{T}$ exhibits factorization, as expected in the boundary CFT.

The argument easily generalizes to all geometries $X_n$ by applying the completeness relation $n$ times, once around each of the $S^2$ boundaries. One obtains

$$Z_{\mathcal{T}}[X_n] = \frac{1}{|\mathcal{A}|^{n-1}} Z_{\mathrm{CS}}[X_n] = \frac{c_0^n}{|\mathcal{A}|^{n-1}} Z_{\mathrm{CS}}[D_3]^n Z_{\mathrm{CS}}[S^3] = Z_{\mathcal{T}}[D_3]^n = 1 \,, \qquad (2.41)$$

again exhibiting factorization.

**Genus $g = 1$.** We introduce the geometries $Y_n = (S^2 \setminus n\mathring{D}_2) \times S^1$: their boundaries $\partial Y_n$ consist of $n$ disconnected $T^2$'s. Note that $Y_1 = D_2 \times S^1 = \mathsf{S}T^2$ is a solid torus, while $Y_2 = T^2 \times I$ is a toroidal cylinder.[18] The homology groups are

$$H_3(Y_n; \mathcal{A}) = 0 \,, \qquad H_2(Y_n; \mathcal{A}) = \mathcal{A}^{n-1} \,, \qquad H_1(Y_n, P; \mathcal{A})\big|_{b=0} = \mathcal{A}^n \,. \qquad (2.42)$$

We can represent $Y_n$ with line insertions along $H_1$ as in Figure 2. In the case $n = 1$ of the solid torus, $H_1$ is generated by the non-contractible cycle and the partition function is

$$Z_{\mathcal{T}}[\mathsf{S}T^2] = Z_{\mathrm{CS}}[\mathsf{S}T^2; \mathcal{A}] = \sum_{a \in \mathcal{A}} Z_{\mathrm{CS}}[\mathsf{S}T^2; a] = Z(R, \tau) \,, \qquad (2.43)$$

where $Z(R, \tau)$ is the modular-invariant CFT torus partition function (2.14). In the middle we have a sum over the insertion of lines $a \equiv (\lambda, \omega\lambda) \in \mathcal{A}$ along the non-contractible cycle, which, according to (2.27), produce the characters of the representations $a$ in RCFT.

In order to discuss factorization, we need once again the completeness relation in CS theory, which allows us to perform surgery around a boundary component. The Hilbert space on $T^2$ has dimension equal to the total number of lines, and a basis of states — that we indicate as $|\mathsf{S}T^2; a\rangle$ — is produced by the path integral on a solid torus with line insertions $a$. The inner product between these states is obtained by taking two solid tori with opposite orientation, each with a line insertion, and gluing them along their boundaries. We obtain $S^2 \times S^1$ with two lines wrapping $S^1$ at different points on $S^2$. The resulting partition function

---

[18]In the context of a different approach to the factorization problem [87], Chern-Simons theory on this wormhole geometry was also studied recently in [88].

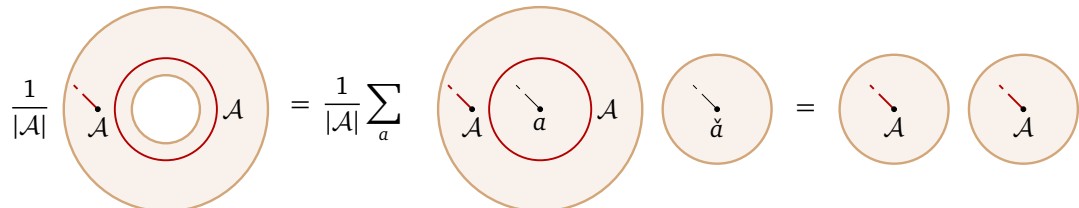

Figure 3: Surgery performed on $Y_2 = T^2 \times I$ (the transverse $S^1$ is kept implicit) in order to reduce it to $\mathsf{S}T^2 \times \mathsf{S}T^2$ and show factorization of the partition function of theory $\mathcal{T}$.

is $Z_{\mathrm{CS}}[S^2 \times S^1; \check{a}, b] = \langle \mathsf{S}T^2; a | \mathsf{S}T^2; b \rangle = \delta_{ab}$, expressing the orthonormality of states. The completeness relation reads

$$\mathbb{1}_{T^2} = \sum_a |\mathsf{S}T^2; a\rangle \langle \mathsf{S}T^2; a|, \tag{2.44}$$

where the sum is over all simple lines in the theory. In terms of the partition function of CS theory, it gives $Z_{\mathrm{CS}}[T^2 \times I] = \sum_a Z_{\mathrm{CS}}[\mathsf{S}T^2; a] Z_{\mathrm{CS}}[\mathsf{S}T^2; \check{a}]$.

Hence, in the case $n = 2$ of the toroidal cylinder (or wormhole), using the properties of the line $\mathcal{A}$ spelled after (2.36) and the completeness relation, we obtain:

$$\begin{aligned}
Z_{\mathcal{T}}[T^2 \times I] &= \frac{1}{|\mathcal{A}|} Z_{\mathrm{CS}}[T^2 \times I; \mathcal{A}, \mathcal{A}^\perp] = \frac{1}{|\mathcal{A}|} \sum_a Z_{\mathrm{CS}}[\mathsf{S}T^2; \mathcal{A}, (\mathcal{A}), a] Z_{\mathrm{CS}}[\mathsf{S}T^2; \check{a}] \\
&= \sum_{a \in \mathcal{A}} Z_{\mathrm{CS}}[\mathsf{S}T^2; \mathcal{A}, a] Z_{\mathrm{CS}}[\mathsf{S}T^2; \check{a}] = \sum_{a \in \mathcal{A}} Z_{\mathrm{CS}}[\mathsf{S}T^2; \mathcal{A}] Z_{\mathrm{CS}}[\mathsf{S}T^2; \check{a}] \\
&= Z_{\mathrm{CS}}[\mathsf{S}T^2; \mathcal{A}] Z_{\mathrm{CS}}[\mathsf{S}T^2; \mathcal{A}] = Z_{\mathcal{T}}[\mathsf{S}T^2] Z_{\mathcal{T}}[\mathsf{S}T^2].
\end{aligned} \tag{2.45}$$

Here $\mathcal{A}^\perp$ is inserted perpendicularly to $\mathcal{A}$. After performing surgery, a cycle becomes contractible in the bulk, and the line inserted around that cycle (which wraps $a$) is denoted by $(\mathcal{A})$. The main steps are graphically represented in Figure 3.

The same procedure can be applied inductively to $Y_n$, as is clear from Figure 2. Indeed, applying the completeness relation to the neighborhood of one boundary $T^2$, one detaches a copy of $\mathsf{S}T^2$ with $\mathcal{A}$ inserted, produces a factor of $|\mathcal{A}|$, and is left with $Y_{n-1}$ with its insertions of $\mathcal{A}$. Finally:

$$Z_{\mathcal{T}}[Y_n] = \left( Z_{\mathcal{T}}[\mathsf{S}T^2] \right)^n. \tag{2.46}$$

**Genus $g > 1$.** We will only consider two geometries: the handlebody $\mathsf{S}\Sigma_g$, and the cylinder $\Sigma_g \times I$. For the handlebody, the homology groups are $H_3(\mathsf{S}\Sigma_g; \mathcal{A}) = H_2(\mathsf{S}\Sigma_g; \mathcal{A}) = 0$ and $H_1(\mathsf{S}\Sigma_g, P; \mathcal{A})\big|_{b=0} = \mathcal{A}^g$, generated by the non-contractible cycles. Therefore the partition function of the gauged theory is

$$Z_{\mathcal{T}}[\mathsf{S}\Sigma_g] = \sum_{a_1, \ldots, a_g \in \mathcal{A}} Z_{\mathrm{CS}}[\mathsf{S}\Sigma_g; a_1, \ldots, a_g]. \tag{2.47}$$

According to (2.27), the sum on the RHS is over conformal blocks and thus this reproduces the modular invariant partition function (2.22) of the compact boson theory.

The completeness relation says that

$$Z_{\mathrm{CS}}[\Sigma_g \times I] = c_g \sum_{a_1, \ldots, a_g} Z_{\mathrm{CS}}[\mathsf{S}\Sigma_g; a_1, \ldots, a_g] Z_{\mathrm{CS}}[\mathsf{S}\Sigma_g; \check{a}_1, \ldots, \check{a}_g], \tag{2.48}$$

where the lines $a_i$ are inserted along the non-contractible cycles, and the coefficient $c_g$ is determined by the normalization of states: $c_g^{-1} = \langle \mathsf{S}\Sigma_g; a_1 \ldots a_g | \mathsf{S}\Sigma_g; a_1 \ldots a_g \rangle$. In order to compute

$c_g$, one uses the following argument. Represent $S^3 = S\Sigma_g \cup S\Sigma_g$ as the gluing of two handle-bodies:[19] the $g$ non-contractible cycles of the first handlebody form $g$ disconnected Hopf links with the cycles of the second handlebody. Since all states have the same normalization $c_g^{-1}$, we find

$$\langle S\Sigma_g; a_1, \ldots, a_g | S_1 \cdots S_g | S\Sigma_g; b_1, \ldots, b_g \rangle = \langle a_1 \cdot b_1, \ldots, a_g \cdot b_g \rangle_{S^3} \tag{2.49}$$
$$= e^{-2\pi i \sum_j (a_j, b_j)} \langle 1 \rangle_{S^3} = c_g^{-1} S_{a_1 \ldots a_g, b_1 \ldots b_g} .$$

Here $a_i \cdot b_i$ indicates a Hopf link, $(a, b)$ is the product that for $\mathfrak{u}(1)_k$ is $(\mu, \nu) = \frac{\mu\nu}{k}$, and in the last expression we have rewritten the first expression in terms of the matrix elements of $S \equiv S_1 \cdots S_g$, where each of the matrices $S_j$ performs an $S$-transformation on one of the handles of $\Sigma_g$. Equating the last two terms and using $\langle 1 \rangle_{S^3} = \mathcal{D}^{-1}$ and the expression (2.10) for the $S$-matrix gives $c_g = \mathcal{D}^{1-g}$. Thus the completeness relation reads

$$\mathbb{1}_{\Sigma_g} = \mathcal{D}^{1-g} \sum_{a_1, \ldots, a_g} |S\Sigma_g; a_1 \ldots a_g\rangle \langle S\Sigma_g; a_1 \ldots a_g| . \tag{2.50}$$

Consider now the cylinder $\Sigma_g \times I$, with homology groups $H_3[\Sigma_g \times I; \mathcal{A}] = 0$, $H_2[\Sigma_g \times I; \mathcal{A}] = \mathcal{A}$, and $H_1[\Sigma_g \times I, P; \mathcal{A}]\big|_{b=0} = \mathcal{A}^{2g}$. The partition function of $\mathcal{T}$ is

$$Z_{\mathcal{T}}[\Sigma_g \times I] = \frac{1}{|\mathcal{A}|} Z_{\text{CS}}[\Sigma_g \times I; \{\mathcal{A}\}_i, \{\mathcal{A}^\perp\}_j] . \tag{2.51}$$

Using the completeness relation we find:

$$Z_{\mathcal{T}}[\Sigma_g \times I] = |\mathcal{A}|^{-g} \sum_{a_1 \ldots a_g} Z_{\text{CS}}[S\Sigma_g; \{\mathcal{A}\}_i, \{(\mathcal{A})\}_j, \{a_i\}] Z_{\text{CS}}[S\Sigma_g; \{\breve{a}_i\}]$$
$$= \sum_{a_1 \ldots a_g \in \mathcal{A}} Z_{\text{CS}}[S\Sigma_g; \{\mathcal{A}\}_i] Z_{\text{CS}}[S\Sigma_g, \{\breve{a}_i\}] = \left( Z_{\mathcal{T}}[S\Sigma_g] \right)^2 . \tag{2.52}$$

Once again, the partition function factorizes as expected in the boundary CFT.

The CFT partition functions are modular invariant, which means that the partition function of $\mathcal{T}$ on a handlebody does not depend on which particular handlebody (distinguished by the set of boundary 1-cycles that are contractible) is attached to the boundary. This is a manifestation of the fact that the path integral of $\mathcal{T}$ is completely independent of the bulk geometry, since this theory is trivial in the bulk.

**Bulk independence and factorization.** Let us finally discuss the general case, after having analyzed several explicit examples. We can prove factorization of the partition function from the fact that the gauged theory $\mathcal{T}$ is trivial in the bulk.

Let $M$ be an oriented three-manifold with boundary $B = \bigsqcup_i \Sigma_{(i)}$, where each $\Sigma_{(i)}$ is a Riemann surface of genus $g_i$. In order to compute the partition function of $\mathcal{T}$ on $M$, we inductively use surgery around each of the boundary components $\Sigma_{(i)}$ [44] (see Figure 4). Since $Z_{\mathcal{T}} = 1$ on any closed 3-manifold, $Z_{\mathcal{T}}[M] = Z_{\mathcal{T}}[M \sqcup S^3]$. We divide $S^3$ in two handlebodies with the same genus $g_i$ as $\Sigma_{(i)}$. Then we use that the Hilbert space of $\mathcal{T}$ on any Riemann surface is one-dimensional, and that in a one-dimensional Hilbert space we can swap $\langle \chi_1 | \chi_2 \rangle \langle \chi_3 | \chi_4 \rangle = \langle \chi_1 | \chi_4 \rangle \langle \chi_3 | \chi_2 \rangle$. We end up with the disconnected sum of a handlebody with

---

[19]Cut $S^3$ along an $S^2$ so as to divide it in two balls, $S^3 = D_3 \cup D_3$. This is the case $g = 0$. Now modify one $D_3$ by removing from its interior a solid handle attached to its boundary $S^2$, and add that handle to the other $D_3$. This gives $S^3 = S T^2 \cup S T^2$, which is the case $g = 1$. By repeating the removal/addition of handles, one obtains $S^3 = S\Sigma_g \cup S\Sigma_g$ for any $g$.

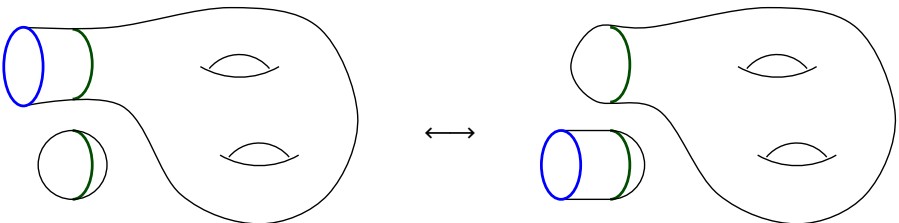

Figure 4: Surgery around a boundary component for a trivial bulk TQFT.

boundary $\Sigma_{(i)}$, and a manifold $M'$ whose boundary is $B \setminus \Sigma_{(i)}$. Repeating the procedure for all boundary components, we end up with a disconnected sum of handlebodies with boundaries $\Sigma_{(i)}$, and a closed manifold. The partition function of $\mathcal{T}$ on the latter is 1. Hence

$$Z_{\mathcal{T}}[M] = \prod_i Z_{\mathcal{T}}[\mathsf{S}\Sigma_{(i)}]. \tag{2.53}$$

This shows that the partition function is completely independent of the choice of $M$ with given boundary condition $B$, and that, therefore, it factorizes as expected in the CFT.

## 2.5 Correlators

Besides partition functions, one can also reproduce the correlation functions of local operators in the compact boson RCFT from the bulk $U(1)_k \times U(1)_{-k}$ description with gauged $\mathbb{Z}_k^{(\omega)}$ 1-form symmetry. In the bulk we follow a two-step procedure: first, we consider correlation functions of bulk lines with endpoints at the boundary, in the ungauged $U(1)_k \times U(1)_{-k}$ theory. Then, we gauge $\mathbb{Z}_k^{(\omega)}$ as in the previous section. After gauging, only a subset of the correlators is left, which is in bijection with the physical correlators of the boundary CFT. (We provide a slightly different and more general perspective in Section 4.3.)

**Sphere two-point function.** We start by considering the two-point correlation function on the sphere. In the $U(1)_k \times U(1)_{-k}$ theory, the two-point function is given by a $D_3$ path integral with a Wilson line anchored to two points at the boundary. We denote Wilson lines as $(\lambda, \bar{\mu})$ as before. We pick the same orientation for the two lines, so that one endpoint $z_1$ has charge $(\lambda, \bar{\mu})$ while the other endpoint $z_2$ has charge $(-\lambda, -\bar{\mu})$. For the purpose of obtaining the most general two-point function after gauging, it is sufficient to consider the case in which the line is unknotted in the bulk (by the same bulk-independence argument as in the previous section, any knotted line would yield the same result after gauging). The resulting two-point correlator in the ungauged theory is [44, 89][20]

$$\left\langle \lambda_{z_1 \to z_2} \bar{\mu}_{z_1 \to z_2} \right\rangle_{D_3} = \left( \frac{z_1 - z_2}{\ell} \right)^{-\frac{\lambda^2}{k}} \left( \frac{\bar{z}_1 - \bar{z}_2}{\ell} \right)^{-\frac{\bar{\mu}^2}{k}}. \tag{2.54}$$

The normalization depends on the arbitrary length scale $\ell$, and it reflects the arbitrariness in the normalization of vertex operators on the boundary.[21] Such a correlation function is not a singled-valued function of the endpoints, and its monodromy is related to the spin of the bulk line.

---

[20]Here we choose $-\frac{k}{2} < \lambda, \bar{\mu} \leq \frac{k}{2}$, representing correlators of primary operators of chiral algebra, because with that choice $h = \frac{\lambda^2}{2k}$ is the dimension of the primary. Shifts of $\lambda$ by $k$ represent chiral-algebra descendants.

[21]The length scale $\ell$ can be thought of as that appearing in the propagator of the chiral boson: $\left\langle \phi(z_1)\phi(z_2) \right\rangle \propto \log(\frac{z_1 - z_2}{\ell})$.

Next, we gauge $\mathbb{Z}_k^{(\omega)}$. This consists in inserting the object $\mathcal{A}$ along all generators of $H_1$, see eqn. (2.35). The insertion of the line $(\lambda, \bar{\mu})$ inside the ball adds an element to $H_1$, generated by the 1-cycle linking the line:

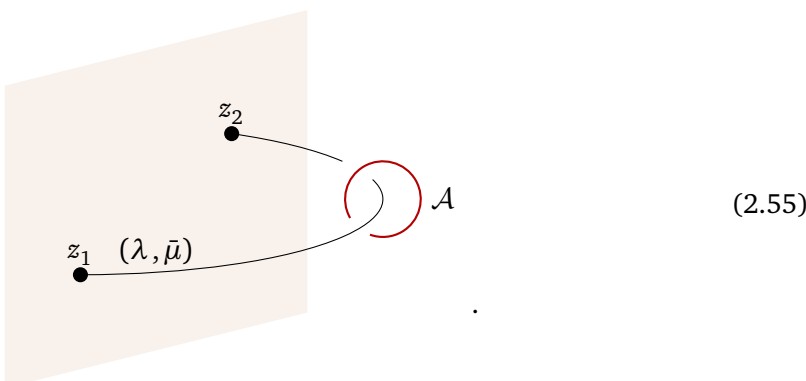

$$\text{(2.55)}$$

.

The effect of $\mathcal{A}$ is to project onto physical fields:

$$\left\langle \lambda_{z_1 \to z_2} \bar{\mu}_{z_1 \to z_2} \right\rangle_{\mathbb{Z}_k^{(\omega)} \text{ gauged}} = k \, \delta_{[\omega\lambda]_k, \bar{\mu}} \left( \frac{z_1 - z_2}{\ell} \right)^{-\frac{\lambda^2}{k}} \left( \frac{\bar{z}_1 - \bar{z}_2}{\ell} \right)^{-\frac{[\omega\lambda]_k^2}{k}}, \qquad (2.56)$$

where $[x]_k$ denotes the integer $-\frac{k}{2} < [x]_k \le \frac{k}{2}$ which equals $x \bmod k$. The only non-vanishing two-point functions coincide, up to normalization, with those of the primary operators of the compact boson RCFT, which have $(h, \bar{h}) = \left( \frac{\lambda^2}{2k}, \frac{[\omega\lambda]_k^2}{2k} \right)$. The constraint $\omega^2 = 1 \bmod 2k$ ensures that all correlation functions are single-valued.

**Sphere $n$-point function.** It is straightforward to extend the above procedure to the case of $n$-point function of vertex operators with $n = 2m$ an even integer, and assuming that the operators come in $m$ pairs of opposite charge. The starting point is now the correlator of $m$ lines of the $U(1)_k \times U(1)_{-k}$ theory. We denote the endpoints of the $i$-th line as $z_{1i}$ and $z_{2i}$, where $i$ runs from 1 to $m$. We take unknotted and unlinked lines. We have [89]:

$$\left\langle \prod_{i=1}^m \lambda^{(i)}_{z_{1i} \to z_{2i}} \bar{\mu}^{(i)}_{z_{1i} \to z_{2i}} \right\rangle = \prod_{i=1}^m \left( \frac{z_{1i} - z_{2i}}{\ell} \right)^{-\frac{\lambda^{(i)2}}{k}} \left( \frac{\bar{z}_{1i} - \bar{z}_{2i}}{\ell} \right)^{-\frac{\bar{\mu}^{(i)2}}{k}}$$

$$\times \prod_{i<j}^m \left[ \frac{(z_{1i} - z_{1j})(z_{2i} - z_{2j})}{(z_{1i} - z_{2j})(z_{1j} - z_{2i})} \right]^{\frac{\lambda^{(i)}\lambda^{(j)}}{k}} \left[ \frac{(\bar{z}_{1i} - \bar{z}_{1j})(\bar{z}_{2i} - \bar{z}_{2j})}{(\bar{z}_{1i} - \bar{z}_{2j})(\bar{z}_{1j} - \bar{z}_{2i})} \right]^{\frac{\bar{\mu}^{(i)}\bar{\mu}^{(j)}}{k}}. \quad (2.57)$$

Gauging $\mathbb{Z}_k^{(\omega)}$ entails inserting $\mathcal{A}$ along the $m$ 1-cycles of the complement of the lines inside the ball, *i.e.*, the cycles that link one of the $m$ lines. Each insertion of $\mathcal{A}$ projects the line that it links to the subspace of charges $\bar{\mu}^{(i)} = \left[ \omega\lambda^{(i)} \right]_k$ that are allowed for RCFT primaries. As a result:

$$\left\langle \prod_{i=1}^m \lambda^{(i)}_{z_{1i} \to z_{2i}} \bar{\mu}^{(i)}_{z_{1i} \to z_{2i}} \right\rangle_{\mathbb{Z}_k^{(\omega)} \text{ gauged}} = k^m \prod_{i=1}^m \delta_{[\omega\lambda^{(i)}]_k, \bar{\mu}^{(i)}} \left( \frac{z_{1i} - z_{2i}}{\ell} \right)^{-\frac{\lambda^{(i)2}}{k}} \left( \frac{\bar{z}_{1i} - \bar{z}_{2i}}{\ell} \right)^{-\frac{[\omega\lambda^{(i)}]_k^2}{k}}$$

$$\times \prod_{i<j}^m \left[ \frac{(z_{1i} - z_{1j})(z_{2i} - z_{2j})}{(z_{1i} - z_{2j})(z_{1j} - z_{2i})} \right]^{\frac{\lambda^{(i)}\lambda^{(j)}}{k}} \left[ \frac{(\bar{z}_{1i} - \bar{z}_{1j})(\bar{z}_{2i} - \bar{z}_{2j})}{(\bar{z}_{1i} - \bar{z}_{2j})(\bar{z}_{1j} - \bar{z}_{2i})} \right]^{\frac{[\omega\lambda^{(i)}]_k [\omega\lambda^{(j)}]_k}{k}}. \quad (2.58)$$

A few comments are in order:

- The $k^m$ prefactor may be adsorbed consistently by an appropriate normalization of the two-point function.[22] After that, all correlators coincide with those of the physical vertex operators in the boundary RCFT.

- As for the two-point function, these correlation functions are single-valued after gauging. Furthermore, since the surviving lines are transparent, the result does not depend on the bulk knotting pattern.

More general charge-conserving $n$-point correlation functions can be obtained starting from the $2m$-point functions above and performing OPEs. They can be obtained considering configurations of anchored lines that fuse in the bulk before gauging.

**Correlators on wormhole geometries.**    One may ask whether, after gauging, correlators on "wormhole" geometries factorize as it happens for partition functions. The topical example is a four-point function on $T^2 \times I$ (or, more generally, on $\Sigma_g \times I$) with two operator insertions on each end of the interval, connected by two Wilson lines threading the bulk. We denote doubled lines $(\lambda, \bar{\mu})$ by $a, b$:

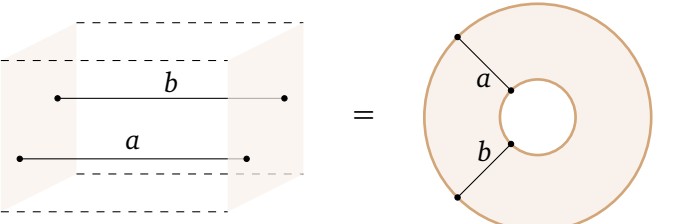

Before gauging, this does not factorize (see also [90]). Based on the previous discussion, after gauging $\mathbb{Z}_k^{(\omega)}$ one can explain factorization in two ways:

1. Lines stretching in the bulk can now "recombine" with the defect network for the gauging, as on the right in Figure 1. Resolving the four-valent junctions and using crossing leads to a factorized correlator.

2. One can use the fact that the bulk Hilbert space is one-dimensional. One performs surgery on the interval and glues a representative of the (only) state on the two-punctured torus (in the figure we keep the transverse $S^1$ implicit):[23]

$$\mathbb{1}_{\mathcal{H}(T^2; a, b)} = \delta_{\check{a}, b}\, \delta_{a \in \mathcal{A}} \quad \text{(two discs with } a \text{ and } \check{a}\text{)}.$$

This leads to the factorized answer:

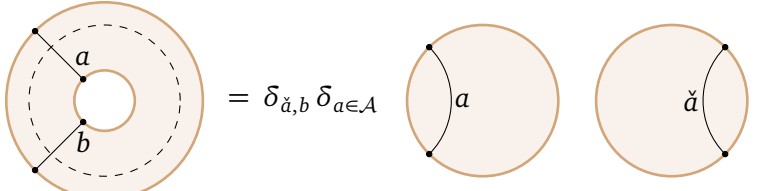

While we have presented the arguments for a fixed configuration, the generalization to more complicated geometries and operator insertions seems quite straightforward.

---

[22]This is achieved by rescaling an operator with scaling dimensions $(h, \bar{h})$ by the factor $\sqrt{k}\, \ell^{h+\bar{h}}$.

[23]In this case the bulk theory is *gauged*, but we do not draw the network for simplicity.

## 2.6 Comments on symmetries

**Dual symmetry.** In our discussion, two bulk theories play a prominent role: the Chern-Simons theory $\mathcal{U}$ and its gauging, *i.e.*, the two following gauge theories:

$$\mathcal{U} = U(1)_k \times U(1)_{-k}, \qquad \mathcal{T} = \frac{U(1)_k \times U(1)_{-k}}{\mathbb{Z}_k^{(\omega)}}. \tag{2.59}$$

Theory $\mathcal{T}$ is obtained from $\mathcal{U}$ by gauging a discrete 1-form symmetry, therefore it should feature a dual global $\mathbb{Z}_k$ 0-form symmetry [83]. On the other hand, $\mathcal{T}$ is trivial and thus the 0-form symmetry does not act on anything. If we gauge it, we go back to $\mathcal{U}$: this shows that $\mathcal{U}$ should be equivalent to a pure $\mathbb{Z}_k$ gauge theory. Restricting for simplicity to the diagonal case $\omega = 1$, this is easy to see with a simple field redefinition in the Lagrangian of $\mathcal{U}$:

$$\mathcal{U}: \qquad \mathcal{L} = \frac{k}{4\pi} a da - \frac{k}{4\pi} \tilde{a} d\tilde{a} = \frac{k}{4\pi} b db + \frac{k}{2\pi} b d\tilde{b}, \tag{2.60}$$

where we defined $a = b + \tilde{b}$, $\tilde{a} = \tilde{b}$, and lowercase fields are dynamical. We see that $\mathcal{U}$ is $(\mathbb{Z}_k)_k$, which is parity invariant (because CS level $\ell$ is equivalent to $\ell + 2k$). On the other hand, theory $\mathcal{T}$ can be written as

$$\mathcal{T}': \qquad \mathcal{L} = \frac{k}{4\pi} B dB + \frac{k}{2\pi} B d\tilde{b}, \tag{2.61}$$

where $B$ is a background field, and $\tilde{b}$ is a Lagrange multiplier restricting $B$ to $\mathbb{Z}_k$. We see that, if one activates a non-trivial background $B$ for the 0-form symmetry, $\mathcal{T}$ behaves as an invertible TQFT. We have used a prime here because, although $\mathcal{T} = \mathcal{T}'$ as theories, the two descriptions are different.

**Non-diagonal case.** The $\mathbb{Z}_2$ automorphism of $U(1)_k$ CS theory used to define $\mathbb{Z}_k^{(\omega)}$ can be given a Lagrangian description as follows.[24] We first integrate in two auxiliary gauge fields $a_1, a_2$ and consider the Lagrangian

$$\mathcal{L} = \frac{k}{4\pi} a da + \frac{1}{2\pi} a_1 da_2. \tag{2.62}$$

The fields $a_{1,2}$ are Lagrange multipliers that simply set each other to zero [86]. As in Section 2.1, let $k = 2pp'$ with $p, p'$ coprime integers, find $r_0, s_0$ such that $pr_0 - p's_0 = 1$, and define $\omega = pr_0 + p's_0$ so that $\omega^2 = 1 + 2r_0s_0 k$. Then perform the $SL(3, \mathbb{Z})$ field redefinition

$$\begin{pmatrix} a \\ a_1 \\ a_2 \end{pmatrix} \rightarrow \begin{pmatrix} \omega & 1 & -r_0 s_0 \\ k s_0 & p & s_0(1 - pr_0) \\ -k r_0 & -p' & r_0(1 + p's_0) \end{pmatrix} \begin{pmatrix} a \\ a_1 \\ a_2 \end{pmatrix}. \tag{2.63}$$

The matrix has unit determinant. The transformation leaves the action invariant, but it acts on the lines as

$$e^{iq \int a} \rightarrow e^{i\omega q \int a}, \tag{2.64}$$

where we used that $a_{1,2}$ are set to zero by the equations of motion.

By performing this transformation on $\tilde{a}$ in (2.60) before changing variables to $b, \tilde{b}$, we obtain the same Lagrangian for $\mathcal{U}$ on the RHS of (2.60), but with a redefinition $b = a - \omega\tilde{a}$ (up to auxiliary gauge fields that can be set to zero). This shows that $b$ gauges precisely the $\mathbb{Z}_k$ 0-form symmetry (parametrized by $\omega$) that does not act on physical operators.

---

[24]This symmetry has also been studied in [91]. The technique used here is similar to the one in [92].

**Global symmetry on the boundary.** Let us now discuss the global symmetries present in the boundary theory. In the standard holographic dictionary, a gauge symmetry in the bulk corresponds to a global symmetry on the boundary. In our setup, we have 0-form $U(1)$ gauge symmetries in the bulk associated to the two Chern-Simons gauge fields, and they induce $U(1)$ global symmetries of both chiralities on the boundary. An important role is played by the boundary action

$$S_\partial = \frac{k}{4\pi} \int_\partial d^2x \sqrt{g}\, g^{z\bar{z}} \left( a_z a_{\bar{z}} - \tilde{a}_z \tilde{a}_{\bar{z}} \right) \tag{2.65}$$

that is needed to impose holomorphic/antiholomorphic boundary conditions, as we review in Appendix B, where we also discuss the relevance of a certain boundary contact term.

In addition, since we gauge a discrete $\mathbb{Z}_k$ 1-form symmetry in the bulk, one might naively expect the existence of this global symmetry on the boundary. In the holographic dictionary the boundary charge operators are obtained by letting the bulk charge operators, over whose insertions we are summing to gauge the symmetry, end on the boundary. Therefore, for the $\mathbb{Z}_k$ 1-form symmetry, one might expect topological local operators on the boundary arising from the endpoint of bulk lines generating the $\mathbb{Z}_k$ subgroup:

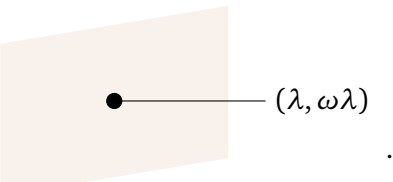

.

On the other hand, we have already seen that actually these endpoints give rise to the physical primary operators of the compact boson CFT, which are not topological. We conclude that this symmetry is explicitly broken by the holomorphic boundary conditions we have imposed. Indeed, the boundary action (2.65) gives rise to a boundary Sugawara stress tensor that assigns a non-zero energy to the endpoints of lines, corresponding to the conformal dimension of the primaries. More directly, one can check that the boundary term (2.65) would not be invariant under $\mathbb{Z}_k$ 1-form gauge transformations of $a$ that are non-vanishing at the boundary.[25]

Finally, let us comment on the implications of the dual $\mathbb{Z}_k$ 0-form symmetry on the boundary. We can let a dual symmetry charge operator end on a line on the boundary, defining a topological defect line. In the bulk, a line bounded by a $\mathbb{Z}_k$ surface can be thought of as a line of the initial theory $\mathcal{U}$, that is not gauge invariant under the $\mathbb{Z}_k$ one-form symmetry and therefore can be defined only as a semi-local operator with a surface attached to it. Such semi-local operators are defined modulo fusion with $\mathcal{A}$. For convenience let us label a basis of these lines by $L_i$. By the the braiding rules of the initial theory $\mathcal{U}$, these lines will act on operators $\phi_a$ at the boundary (which are the endpoints of $a \in \mathcal{A}$) by $L_i[\phi_a] = \frac{S_{ia}}{S_{0a}} \phi_a$. This is the action of Verlinde lines in RCFTs on primary operators [93–95]. Thus we conclude that this procedure gives rise to the set of Verlinde lines of the physical RCFT.

## 3 Abelian generalizations

The simple example of a 2d compact scalar, dual to the $U(1)_k \times U(1)_{-k}/\mathbb{Z}_k^{(\omega)}$ Chern-Simons theory, can be generalized in many ways. In this section we consider a few Abelian generalizations, including a multi-component compact scalar and a non-rational case.

---

[25]This is also related to the fact that when gauging $\mathbb{Z}_k^{(\omega)}$ in Section 2.3, we impose Dirichelet boundary conditions to the $\mathbb{Z}_k$ 2-form gauge fields at the boundary.

### 3.1 Multi-component compact scalar

Consider $D$ real free compact bosons $X^j \cong X^j + 2\pi$, with Euclidean action:[26]

$$S = \frac{1}{8\pi} \int d^2\sigma \left( G_{ij}\, \delta^{\alpha\beta}\, \partial_\alpha X^i\, \partial_\beta X^j + iB_{ij}\, \varepsilon^{\alpha\beta}\, \partial_\alpha X^i\, \partial_\beta X^j \right). \tag{3.1}$$

The target space is a torus $T^D$ with metric $G_{ij}$ and B-field $B_{ij}$. There exists a moduli space $\mathcal{M}_D$ worth of CFTs obtained by varying the metric and the B-field, modulo field redefinitions and dualities,[27] and it is known as the Narain moduli space [17]:

$$\mathcal{M}_D = O(D,D;\mathbb{Z})\backslash O(D,D;\mathbb{R})/O(D) \times O(D). \tag{3.2}$$

Moreover, the theory has $\mathfrak{u}(1)^D \times \mathfrak{u}(1)^D$ current algebra generated by the currents $\partial X^i$ and $\bar{\partial} X^i$, respectively.

Given a CFT identified by $m \in \mathcal{M}_D$, the torus partition function is

$$Z(m,\tau) = \frac{\Theta(m,\tau)}{|\eta(\tau)|^{2D}}, \tag{3.3}$$

where

$$\Theta(m,\tau) = \sum_{\vec{n},\vec{w}\in\mathbb{Z}^D} \exp\left[ -2\pi\tau_2\left( G^{ij} v_i v_j + \tfrac{1}{4} G_{ij} w^i w^j \right) + 2\pi i\tau_1 n_i w^i \right] = \sum_{\vec{n},\vec{w}\in\mathbb{Z}^D} q^{h_{\vec{n},\vec{w}}}\, \bar{q}^{\bar{h}_{\vec{n},\vec{w}}} \tag{3.4}$$

is the Siegel-Narain theta function, while

$$v_i = n_i + \tfrac{1}{2} B_{ij} w^j \tag{3.5}$$

is the velocity (in the presence of B-field). In the rightmost expression we used the left and right dimensions of primary operators,

$$h_{\vec{n},\vec{w}} = \frac{1}{2} \left| \vec{n} + \frac{G+B}{2}\vec{w} \right|_{G^{-1}}^2, \qquad \bar{h}_{\vec{n},\vec{w}} = \frac{1}{2} \left| \vec{n} - \frac{G-B}{2}\vec{w} \right|_{G^{-1}}^2. \tag{3.6}$$

In order to avoid clutter, we introduced the notation $|\vec{x}|_M^2 = x^i M_{ij} x^j$.

**Chiral algebras.** When $G, B \in \mathbb{Q}^{D\times D}$, the theory is a RCFT. In order to determine the chiral algebras, we proceed as follows. First define the matrix

$$M = \frac{G+B}{2} \quad \Rightarrow \quad M^\mathsf{T} = \frac{G-B}{2}, \qquad G = M + M^\mathsf{T}, \qquad B = M - M^\mathsf{T}. \tag{3.7}$$

The left-moving chiral algebra is given by all operators with $\bar{h}_{\vec{n},\vec{w}} = 0$, which are solutions to the equation

$$\vec{n} = M^\mathsf{T}\vec{w}, \qquad \text{with} \quad \vec{n}, \vec{w} \in \mathbb{Z}^D. \tag{3.8}$$

This equation gives rise to two lattices. One is the lattice $\Lambda_L$ of vectors $\vec{w}$ such that $M^\mathsf{T}\vec{w} \in \mathbb{Z}^D$. We package a set of generators of $\Lambda_L$ into an integer matrix $P_L$ (defined up to multiplication by unimodular matrices from the right) as its columns. In other words, $\Lambda_L = P_L\, \mathbb{Z}^D$. The other

---

[26]To make contact with (2.1), set $D = 1$ and then $G_{ii} = R^2$ here and $\varphi = RX$ there.
[27]The set of equivalences was clearly reviewed in [96]. It includes unimodular transformations of the fields $X^j$, as well as gauge transformations that shifts the components of $B_{ij}$ by even integers.

lattice is $M^{\mathsf{T}}\Lambda_L$ of image values of $\vec{n}$, and a set of generators is given by the columns of the integer matrix $\widetilde{P}_L = M^{\mathsf{T}}P_L$. We can rewrite this relation as

$$M^{\mathsf{T}} = \widetilde{P}_L P_L^{-1}, \tag{3.9}$$

which could be regarded as the matrix version of writing a fraction in the irreducible form.[28] The chiral operators are labelled by $\vec{n} = M^{\mathsf{T}}P_L\vec{\ell}$, $\vec{w} = P_L\vec{\ell}$ with $\vec{\ell} \in \mathbb{Z}^D$ and their left-moving dimensions are $h = \frac{1}{2}\vec{\ell}^{\,\mathsf{T}}P_L^{\mathsf{T}}GP_L\vec{\ell}$. We recognize the chiral algebra

$$\mathfrak{u}(1)_{K_L}^D, \qquad \text{with} \quad K_L = P_L^{\mathsf{T}}GP_L = \widetilde{P}_L^{\mathsf{T}}P_L + P_L^{\mathsf{T}}\widetilde{P}_L. \tag{3.10}$$

These relations should be compared with (2.6) and (2.7) in the case $D = 1$. We see that $K_L$ is a positive symmetric even integer matrix, namely $K_L > 0$, $(K_L)_{ij} \in \mathbb{Z}$ and $(K_L)_{ii} \in 2\mathbb{Z}$ (corresponding to a bosonic Chern-Simons theory).

The chiral algebra $\mathfrak{u}(1)_{K_L}^D$ has integrable representations labelled by the discriminant group $\mathcal{D}_L = \mathbb{Z}^D/K_L\mathbb{Z}^D$, where $K_L\mathbb{Z}^D$ is the integer lattice generated by the columns of $K_L$. The order of the discriminant is $|\mathcal{D}_L| = \det K_L$, the dimensions of chiral primary operators are

$$h = \frac{1}{2}|\vec{\lambda}|_{K_L^{-1}}^2 \tag{3.11}$$

where $\vec{\lambda}$ is the left-moving charge of the primary operator, and the characters are

$$K_{\vec{\lambda}}^{(K_L)}(\tau) = \frac{1}{\eta(\tau)^D}\sum_{\vec{\ell} \in \mathbb{Z}^D} q^{\frac{1}{2}|\vec{\lambda}+K_L\vec{\ell}|_{K_L^{-1}}^2}, \tag{3.12}$$

for $\vec{\lambda} \in \mathcal{D}_L$. Indeed, comparing (3.11) with (3.6), we identify the left-moving charge of a generic operator as $\vec{q}_L = P_L^{\mathsf{T}}\vec{n} + P_L^{\mathsf{T}}M\vec{w}$. This can be written as $\vec{\lambda} + K_L\vec{\ell}$, where the integer vector $\vec{\ell}$ parametrizes the chiral algebra descendants. The left-moving dimension $h$ mod 1 provides a quadratic function $h : \mathcal{D}_L \to \mathbb{R}/\mathbb{Z}$ that can be used to construct the bilinear form $K_L^{-1}(\cdot,\cdot) : \mathcal{D}_L \times \mathcal{D}_L \to \mathbb{R}/\mathbb{Z}$.

The right-moving chiral algebra is obtained in a similar way. Its operators are solutions to $h_{\vec{n},\vec{w}} = 0$, namely to $\vec{n} = -M\vec{w}$ with $\vec{n}, \vec{w} \in \mathbb{Z}^D$. One defines the lattice $\Lambda_R$ of integer vectors $\vec{w}$ such that $M\vec{w} \in \mathbb{Z}^D$, and packages a set of generators into the integer matrix $P_R$ as its columns. The lattice $M\Lambda_R$ of image values of $\vec{n}$ is generated by the columns of the integer matrix $\widetilde{P}_R = MP_R$. The chiral algebra is then $\mathfrak{u}(1)_{K_R}^D$ with

$$M = \widetilde{P}_R P_R^{-1}, \qquad\qquad K_R = P_R^{\mathsf{T}}GP_R = \widetilde{P}_R^{\mathsf{T}}P_R + P_R^{\mathsf{T}}\widetilde{P}_R. \tag{3.13}$$

Notice that $\det K_L = \det K_R$.

**Left-right pairing.** Let us understand how left- and right-moving representations of chiral algebra are paired into physical fields. The operator $(\vec{n}, \vec{w})$ has left- and right-moving charges $\vec{q}_L = P_L^{\mathsf{T}}\vec{n} + P_L^{\mathsf{T}}M\vec{w}$ and $\vec{q}_R = P_R^{\mathsf{T}}\vec{n} - P_R^{\mathsf{T}}M^{\mathsf{T}}\vec{w}$, respectively. If we mod out the lattice $\mathbb{Z}^{2D}$ of vectors $\binom{\vec{n}}{\vec{w}}$ by the holomorphic and anti-holomorphic fields corresponding to the sublattice $J\mathbb{Z}^{2D}$ where

$$J = \begin{pmatrix} M^{\mathsf{T}}P_L & -MP_R \\ P_L & P_R \end{pmatrix}, \tag{3.14}$$

---

[28]Special representatives for $P_L, \widetilde{P}_L$ can be found by performing Smith decomposition of $M^{\mathsf{T}}$, namely writing $M^{\mathsf{T}} = UD_{\mathbb{Q}}V$ where $U, V$ are integer unimodular matrices, while $D_{\mathbb{Q}}$ is a diagonal rational matrix. We write $D_{\mathbb{Q}} = \mathrm{Num}\cdot\mathrm{Den}^{-1}$ where Num, Den are the two diagonal integer matrices of numerators and denominators of the entries of $D_{\mathbb{Q}}$ written in the irreducible form. Then $P_L = V^{-1}\mathrm{Den}$ and $\widetilde{P}_L = U\,\mathrm{Num}$. We also find $P_R = U^{-1\mathsf{T}}\mathrm{Den}$ and $\widetilde{P}_R = V^{\mathsf{T}}\mathrm{Num}$ for the two other matrices defined below.

then we are left with the group $\mathbb{Z}^{2D}/J\,\mathbb{Z}^{2D}$ of order $\det J = \det K_L = \det K_R$. These are the integrable representations, paired by a group isomorphism. We say that two integer matrices $P,Q$ are coprime if $\mathbb{Z}^D$ is the only integer lattice such that both $P\mathbb{Z}^D$ and $Q\mathbb{Z}^D$ are sublattices thereof (this condition is invariant under multiplication of $P,Q$ by integer unimodular matrices from the right). In our case, the matrices $P_L^\mathsf{T}$ and $\widetilde{P}_L^\mathsf{T}$ are coprime.[29] It follows that there exist integer matrices $N_{1,2}$ such that

$$S = \begin{pmatrix} P_L^\mathsf{T} & P_L^\mathsf{T} M \\ N_1 & N_2 \end{pmatrix} \in SL(2D,\mathbb{Z})\,. \tag{3.15}$$

Indeed the columns of $P_L^\mathsf{T}$, $\widetilde{P}_L^\mathsf{T} = P_L^\mathsf{T} M$ generate $\mathbb{Z}^D$, and thus there exists a linear integer and invertible change of coordinates $S$ from $(\vec{n},\vec{w})$ to $(\vec{q}_L,\vec{\ell}\,)$, for some integer coordinate $\vec{\ell}$ that parametrizes the chiral algebra descendants on the anti-holomorphic side. Let the inverse integer matrix be

$$S^{-1} = \begin{pmatrix} R_0 & MN_3 \\ -S_0 & -N_3 \end{pmatrix}\,, \tag{3.16}$$

so that $\left(\begin{smallmatrix}\vec{n}\\\vec{w}\end{smallmatrix}\right) = S^{-1}\left(\begin{smallmatrix}\vec{q}_L\\\vec{\ell}\end{smallmatrix}\right)$. In particular[30] $P_L^\mathsf{T} R_0 - \widetilde{P}_L^\mathsf{T} S_0 = \mathbb{1}$. Substituting into the right-moving charge $\vec{q}_R$, we determine[31] that $\vec{q}_R = \omega\,\vec{q}_L \bmod K_R\mathbb{Z}^D$, where

$$\omega = P_R^\mathsf{T} R_0 + \widetilde{P}_R^\mathsf{T} S_0\,. \tag{3.17}$$

This integer matrix gives the group isomorphism $\omega : \mathcal{D}_L \to \mathcal{D}_R$. With some algebra, one can show that $\omega^\mathsf{T} K_R^{-1} \omega = K_L^{-1} + S_0^\mathsf{T} R_0 + R_0^\mathsf{T} S_0$. This implies that

$$\bar{h}(\omega\vec{\lambda}) = h(\vec{\lambda}) \pmod 1, \qquad \text{for all} \quad \vec{\lambda} \in \mathcal{D}_L\,, \tag{3.18}$$

namely, that $\omega$ maps the quadratic function $h$ on $\mathcal{D}_L$ to the quadratic function $\bar{h}$ on $\mathcal{D}_R$. As proven in [97], the Abelian Chern-Simons theories constructed with the matrices $K_L$ and $K_R$ are equivalent, either at the classical level (by a field redefinition, if $K_R = \check{V}^\mathsf{T} K_L \check{V}$ for some integer unimodular matrix $\check{V}$) or at the quantum level.

Eventually, the torus partition function can be written as

$$Z(m,\tau) = \sum_{\vec{\lambda}\in\mathcal{D}_L} K_{\vec{\lambda}}^{(K_L)}(\tau)\,\overline{K_{\omega\vec{\lambda}}^{(K_R)}(\tau)}\,. \tag{3.19}$$

In the following, in order to avoid clutter, we will assume that the left and right matrices defining the chiral algebras are equal,

$$K_L = K_R \equiv K\,, \tag{3.20}$$

so that $\omega$ defines an automorphism of $\mathcal{D}_0 = \mathbb{Z}^D/K\,\mathbb{Z}^D$.

---

[29]Suppose that they are not. Then there exists an integer lattice $L\mathbb{Z}^D$ (with $\det L > 1$) such that $P_L^\mathsf{T}\mathbb{Z}^D$ and $\widetilde{P}_L^\mathsf{T}\mathbb{Z}^D$ are sublattices thereof. Hence one can write $P_L^\mathsf{T} = LR^\mathsf{T}$, $\widetilde{P}_L^\mathsf{T} = L\widetilde{R}^\mathsf{T}$ in terms of integer matrices $R, \widetilde{R}$ providing a decomposition $M^\mathsf{T} = \widetilde{R}R^{-1}$. However $R\mathbb{Z}^D$, which is finer than $P_L\mathbb{Z}^D$, is mapped by $M^\mathsf{T}$ into $\mathbb{Z}^D$, in contradiction with the hypothesis that $P_L\mathbb{Z}^D$ is the totality of vectors $\vec{w}$ with that property.

[30]In the representation of footnote 28, $R_0, S_0$ are easily determined. Set $R_0 = V^\mathsf{T} R_0^{\mathrm{red}}$ and $S_0 = U^{-1\mathsf{T}} S_0^{\mathrm{red}}$. The reduced matrices satisfy: $\mathrm{Den}\,R_0^{\mathrm{red}} - \mathrm{Num}\,S_0^{\mathrm{red}} = \mathbb{1}$, which is a diagonal equation, hence $R_0^{\mathrm{red}}, S_0^{\mathrm{red}}$ are diagonal. Then $N_1 = S_0^{\mathrm{red}} V^{-1\mathsf{T}}$, $N_2 = R_0^{\mathrm{red}} U^\mathsf{T}$, and $N_3 = -P_R$.

[31]One finds $\vec{q}_R = P_R^\mathsf{T}(R_0 + M^\mathsf{T} S_0)\vec{q}_L + P_R^\mathsf{T} GN_3\vec{\ell}$. Since $\vec{\ell}$ parametrizes the anti-holomorphic sector, $N_3 = P_R\check{U}$ for some unimodular integer matrix $\check{U}$. Note that different solutions to the problem in (3.15)-(3.16) are related by $N_1 \to N_1 - WP_L^\mathsf{T}$, $N_2 \to N_2 - W\widetilde{P}_L^\mathsf{T}$, $R_0 \to R_0 + MN_3W$, $S_0 \to S_0 + N_3W$ for integer matrices $W$. The matrix $\omega$ shifts by $K_R(\check{U}W)$, showing that $\omega$ is well-defined as a map to $\mathcal{D}_R$. By some algebra one shows that $\omega K_L = -K_R\check{U}(N_1\widetilde{P}_L + N_2 P_L)$ so that $\omega$ is a map from $\mathcal{D}_L$ to $\mathcal{D}_R$.

**Chern-Simons bulk dual.** In the three-dimentional bulk we consider the Chern-Simons theory $U(1)_K^D \times U(1)_{-K}^D$ with action

$$S_{\text{CS}} = \frac{K_{ij}}{4\pi} \int \left( A_L^i dA_L^j - A_R^i dA_R^j \right), \tag{3.21}$$

where $i = 1, \dots, D$. Here $K$ is the positive symmetric even integer matrix constructed above, so that the resulting Chern-Simons theory is bosonic and has vanishing chiral central charge.[32] On the left side, $U(1)_K^D$, the lines correspond to the elements of $\mathcal{D}_0$, and the $S$ and $T$ matrices — that can be read off from the characters (3.12) — are

$$S_{\vec{\lambda},\vec{\mu}} = \frac{1}{|K|^{1/2}} e^{-2\pi i \vec{\lambda}^{\mathsf{T}} K^{-1} \vec{\mu}},$$

$$T_{\vec{\lambda},\vec{\mu}} = e^{-2\pi i D/24} \theta_{\vec{\lambda}} \delta_{\vec{\lambda},\vec{\mu}}, \qquad \text{with} \quad \theta_{\vec{\lambda}} = e^{\pi i |\vec{\lambda}|_{K^{-1}}^2}, \tag{3.22}$$

where $|K| = |\det K|$ is the order of $\mathcal{D}_0$. The phase of $\theta_{\vec{\lambda}}$ is the chiral dimension $h$ in (3.11) mod 1, *i.e.*, the spin, and it provides a quadratic refinement of the braiding matrix, $B(\vec{\lambda}, \vec{\mu}) = S_{\vec{\lambda},\vec{\mu}}^* S_{0,0}/(S_{0,\vec{\lambda}} S_{0,\vec{\mu}}) = \theta_{\vec{\lambda}+\vec{\mu}}/\theta_{\vec{\lambda}} \theta_{\vec{\mu}}$. In the presence of a boundary, we impose holomorphic boundary conditions for $U(1)_K^D$ and antiholomorphic for $U(1)_{-K}^D$.

The lines of an Abelian Chern-Simons theory form a group under fusion, the 1-form symmetry group, which in this case is $\mathcal{D}_0 \times \mathcal{D}_0$. We want to gauge a non-anomalous subgroup $\mathcal{A}$. The anomaly cancelation condition is that each line in $\mathcal{A}$ has integer spin [33, 55], implying that each line in $\mathcal{A}$ has trivial braiding with all other lines. Now the total number of lines is $|K|^2$, therefore if $\mathcal{A}$ has order $|K|$ then after gauging the theory has only one line — the identity — and is trivial in the bulk (without gravitational anomaly). A particularly simple way to satisfy both conditions is to find a group isomorphism $\omega : \mathcal{D}_0 \to \mathcal{D}_0$ that preserves the spin of the lines, eqn. (3.18).[33] Then

$$\mathcal{A} = \bigoplus_{\vec{\lambda} \in \mathcal{D}_0} (\vec{\lambda}, \omega\vec{\lambda}), \tag{3.23}$$

where $(\vec{\lambda}, \omega\vec{\lambda})$ is a line of $U(1)_K^D \times U(1)_{-K}^D$. The gauging procedure and the factorization of the partition functions follow exactly the same steps as in the previous section, with $k$ replaced by the matrix $K$. After gauging, this theory is holographically dual to the multi-component compact boson RCFT defined by $(K, \omega)$. In particular, the partition function on a solid torus is $Z(m, \tau)$ in (3.19). The triviality of the bulk theory implies factorization and independence of the bulk geometry.

## 3.2 More general Lagrangian subgroups

As repeated above, the lines of an Abelian Chern-Simons theory form the 1-form symmetry group of the theory. A subgroup $\mathcal{A}$ is non-anomalous if each line has integer spin. A non-anomalous subgroup $\mathcal{A}$ is called *Lagrangian* if it has maximal order, namely if $|\mathcal{A}|$ is equal to the square root of the total number of lines. Gauging a Lagrangian subgroup generates a theory that is trivial in the bulk.

We are interested in theories of the form $\mathcal{C}_L \times \overline{\mathcal{C}}_R$, where both $\mathcal{C}_L$ and $\mathcal{C}_R$ are Abelian Chern-Simons theories with positive-definite matrix. On general grounds, Lagrangian subgroups are in correspondence with topological boundary conditions [58, 59, 67], and we can use the

---

[32]More generally, one could relax the condition that $K$ is positive, or take a more general (symmetric even integer) Chern-Simons matrix that does not have the block-diagonal form $\left(\begin{smallmatrix} K & 0 \\ 0 & -K \end{smallmatrix}\right)$ as we did. These cases have been studied in [64].

[33]In the presence of generic left and right sectors $K_L$, $K_R$, the condition is that $\omega : \mathcal{D}_L \to \mathcal{D}_R$ satisfies $\omega^{\mathsf{T}} K_R^{-1} \omega = K_L^{-1} + N$ for some symmetric even integer matrix $N$.

folding trick to map them to topological interfaces between $\mathcal{C}_L$ and $\mathcal{C}_R$. One possibility is that the interface is invertible: it then represents an isomorphism between the 1-form symmetry groups of $\mathcal{C}_L$ and $\mathcal{C}_R$ that preserves the spin. This is precisely the isomorphism $\omega$ that we described before, and the Lagrangian subgroup is (3.23).

However, there are more general possibilities. For instance, $\mathcal{C}_L$ and $\mathcal{C}_R$ could be non-isomorphic. Or, $\mathcal{C}_L$ could have a Lagrangian subgroup $\mathcal{A}_L$, $\mathcal{C}_R$ could have a Lagrangian subgroup $\mathcal{A}_R$, so that $\mathcal{A}_L \otimes \mathcal{A}_R$ is a Lagrangian subgroup of $\mathcal{C}_L \times \overline{\mathcal{C}_R}$. This case would produce a partition function which is the product of a left and a right part, each separately modular invariant. In the general case, Lagrangian subgroups are still in correspondence with topological interfaces between $\mathcal{C}_L$ and $\mathcal{C}_R$, but these might be non-invertible. Two necessary conditions for the existence of Lagrangian subgroups are that the total number of lines is a perfect square, and that the signature of the total Chern-Simons matrix is a multiple of 24.[34]

We should mention an interesting subtlety: a model whose maximally extended chiral algebra is $\mathcal{C}$, can also be described in terms of a sub-algebra $\widetilde{\mathcal{C}} \subset \mathcal{C}$. Let us give a simple example. The theory $U(1)_2 \times U(1)_{-2}$ has a unique Lagrangian subgroup $\mathcal{A} = (0,0) \oplus (1,1)$, which gives rise to the compact boson CFT at self-dual radius $R^2 = 2$. Also the theory $U(1)_8 \times U(1)_{-2}$ has a unique Lagrangian subgroup $\mathcal{A} = (0,0) \oplus (2,1) \oplus (4,0) \oplus (6,1)$, isomorphic to $\mathbb{Z}_4$. Gauging of $\mathcal{A}$ produces the modular-invariant torus partition function[35]

$$Z = \left(K_0^{(8)}(\tau) + K_4^{(8)}(\tau)\right)\overline{K_0^{(2)}(\tau)} + \left(K_2^{(8)}(\tau) + K_6^{(8)}(\tau)\right)\overline{K_1^{(2)}(\tau)} = \left|K_0^{(2)}(\tau)\right|^2 + \left|K_1^{(2)}(\tau)\right|^2 , \quad (3.24)$$

which describes, once again, the compact boson at $R^2 = 2$. The two models describe the very same RCFT, but in the latter one uses the chiral algebra $\mathfrak{u}(1)_8$ which is a subalgebra of $\mathfrak{u}(1)_2$. The theory $U(1)_8 \times U(1)_{-8}$ has three Lagrangian subgroups: $\mathbb{Z}_8^{(1)}$ and $\mathbb{Z}_8^{(-1)}$ that we already described, as well as $\mathcal{A} = (0,0) \oplus (0,4) \oplus (4,0) \oplus (4,4) \oplus (2,2) \oplus (2,6) \oplus (6,2) \oplus (6,6)$ which is isomorphic to $\mathbb{Z}_2 \times \mathbb{Z}_4$. Gauging the first two gives the compact boson CFT at $R^2 = 8$ and $R^2 = 1/2$, respectively, that are dual to each other. Gauging the latter gives, once again, the compact boson CFT at $R^2 = 2$.

## 3.3 Non-rational theories

We conclude this section with a non-rational example. The starting point is a non-compact Abelian theory: Chern-Simons with gauge group $\mathbb{R}$ (which can be easily generalized to $\mathbb{R}^D$). We do not have full control over all details (in particular the normalization of states should be treated carefully), but we present some interesting ideas. When the gauge group is $\mathbb{R}$, the level is not physical (it can be rescaled by a field redefinition) and the action reads

$$S_{\mathbb{R}} = \frac{1}{4\pi} \int A\, dA . \quad (3.25)$$

This theory has a continuous family of Wilson lines $W_s = e^{is \int A}$ labeled by $s \in \mathbb{R}$. With this normalization, the solid torus partition function gives the $\mathfrak{u}(1)$ characters

$$\chi_s^{\mathbb{R}}(\tau) = \frac{1}{\eta(\tau)} q^{\frac{s^2}{2}} . \quad (3.26)$$

---

[34]The signature of the total CS matrix can just be a multiple of 8, if we allow stacking by the bosonic invertible theory $(E_8)_1$. Necessary and sufficient conditions, easily computable, for the existence of Lagrangian subgroups have recently been found in [67]. Unfortunately, they do not tell us which nor how many inequivalent Lagrangian subgroups there exist.

[35]One uses the identity $\sum_{i=0}^{p-1} K_{p(\lambda+ki)}^{(kp^2)}(\tau) = K_\lambda^{(k)}(\tau)$, which extends to the characters refined by $U(1)$.

The modular matrices are

$$T_{s,t} = e^{2\pi i\left(\frac{s^2}{2} - \frac{1}{24}\right)} \delta(s - t), \qquad S_{s,t} = e^{-2\pi i s t}, \qquad C_{s,t} = \delta(s + t). \qquad (3.27)$$

We can now stack the theory $\mathbb{R}$ with its parity reversal, that we indicate as $\overline{\mathbb{R}}$, and look for maximal non-anomalous subgroups of the 1-form symmetry to be gauged. The definition of a Lagrangian subgroup should be modified because the theory has an infinite number of lines. We define a Lagrangian (Abelian) subgroup by:

1)  The lines in $\mathcal{A}$ must form a group.

2)  The lines in $\mathcal{A}$ must have integer spin, and thus be mutually transparent.

3)  Every other line in the theory must link non-trivially with at least one line in $\mathcal{A}$.

Condition 3) is the same as asking that $\mathcal{A}$ be maximal, in the sense that no other line can be added to $\mathcal{A}$ while preserving 1) and 2). Each line in $\mathcal{A}$ will be of the form $L_s \otimes \overline{L}_t$. Asking the lines in $\mathcal{A}$ to have integer spin implies

$$(s + t)(s - t) = 0 \mod 2. \qquad (3.28)$$

This automatically implies that the lines in $\mathcal{A}$ are mutually transparent. There are two types of solutions.

One possibility to solve (3.28) is to set $s = \pm t$. If we include in $\mathcal{A}$ lines with both signs, their composition does not close inside $\mathcal{A}$. Therefore, without loss of generality, let us consider the diagonal case $s = t$ (the antidiagonal case $s = -t$ leads to the same result). The Lagrangian condition implies that all diagonal lines $L_s \otimes \overline{L}_s$ appear in $\mathcal{A}$:

$$\mathcal{A}_\infty = \int ds\, L_s \otimes \overline{L}_s. \qquad (3.29)$$

This is a Lagrangian subgroup, whose gauging gives the CFT of a non-compact free boson (with zero-mode removed). In particular the torus partition function is

$$Z_{T^2} = \int ds\, \left|\chi_s^{\mathbb{R}}(\tau)\right|^2 = \frac{1}{\sqrt{2\tau_2}\, |\eta(\tau)|^2}. \qquad (3.30)$$

The integral in (3.29) could have in principle included a non-trivial measure factor $\mu(s)$. A more refined argument[36] shows that the measure must be constant, and we set it to one.

Another possibility to solve (3.28) is to set

$$s + t = \frac{2n}{R}, \qquad s - t = wR, \qquad \text{with} \quad n, w \in \mathbb{Z}, \qquad (3.31)$$

for some real positive number $R$. The Lagrangian condition forces us to include all possible values of $n, w$. Thus:

$$\mathcal{A}_R = \bigoplus_{n,w \in \mathbb{Z}} L_{\frac{n}{R} + \frac{wR}{2}} \otimes \overline{L}_{\frac{n}{R} - \frac{wR}{2}}. \qquad (3.32)$$

Gauging of this Lagrangian subgroup gives the compact boson CFT at generic radius $R$, and the torus partition function is indeed (2.2).

The treatment above can be extended to the multi-component case ($\mathbb{R}^D$ Chern-Simons theory) in a straightforward way. As expected, one finds that all the free CFTs of $D$ compact or non-compact (with zero-modes removed) scalars are described by Lagrangian algebras.

---

[36]In Section 4.1 we describe a more general formalism, based on modular tensor categories. In (4.18), $Z_s^{\mathcal{A}}$ plays the role of $\mu(s)$. Since the $F$-matrix is $[F_{s+t+u}^{stu}]_{t+u,s+t} = 1$, the associativity condition (4.23) implies $\mu(s+t) = \mu(t+u)$ for all $s, t, u \in \mathbb{R}$, from which it follows that $\mu(s)$ must be constant.

# 4 Non-Abelian case

Let us now move to the case of non-Abelian Chern-Simons theories. They describe interacting boundary RCFTs, for instance Wess-Zumino-Witten (WZW) models.[37] In the bulk we take $G_k \times G_{-k}$ Chern-Simons theory, where $G$ is a compact Lie group and $k$ is its level. As before, we impose holomorphic and anti-holomorphic boundary conditions for the two factors, respectively, in order to obtain a $\mathfrak{g}_k \oplus \overline{\mathfrak{g}}_k$ chiral algebra [56].

In the non-Abelian case, the fusion of lines does not give rise to a group structure but rather to a full-fledged fusion algebra. There might be a subset of the simple lines, called Abelian, with the property that their fusion with any other simple line gives a single simple line (as opposed to a composite line). Under fusion, the set of Abelian lines has the structure of an Abelian group, the 1-form symmetry of the theory. A non-anomalous subgroup of it could be gauged, however this is not sufficient to render the theory trivial.[38] The reason is that most lines are not Abelian. The fact that they are topological, though, implies that they constitute a sort of symmetry. Indeed, the full set of lines form the so-called non-invertible 1-form symmetry of the theory, which is a modular tensor category (MTC).

There exists an analogous concept to the gauging of a symmetry group that applies to non-invertible symmetries, called anyon condensation [41, 42]. This is an algebraic procedure that, given an initial MTC and a piece of data called an "algebra object" $\mathcal{A}$, produces a new MTC corresponding to the TQFT after gauging. The algebra object must satisfy certain consistency conditions, that are the analog of picking a non-anomalous subgroup. If the algebra object has maximal dimension, it is called a Lagrangian algebra and its condensation produces a trivial theory. It turns out that in $G_k \times G_{-k}$ Chern-Simons theory there always are one or more Lagrangian algebra objects, whose condensation produces the holographic duals to many RCFTs.

## 4.1 Review of anyon condensation in modular tensor categories

To set our notation, let us review the main data of a modular tensor category and the procedure of anyon condensation. For more detailed expositions, we refer, *e.g.*, to [52, 54, 57, 98, 99] for the former and to [42] for the latter.

**Modular tensor categories (MTC).** To a Chern-Simons theory (or to a chiral algebra) one associates a category $\mathcal{C}$ whose objects are the Wilson lines (or the integrable representations of chiral algebra), and whose morphisms are maps between lines (or intertwiners between representations). We indicate the objects as $L$.

In a tensor category one can take tensor products of objects and morphisms, and there is a unique trivial line $L_0$ (sometimes indicated also as $L_{\mathbb{1}}$) providing the neutral element. The tensor product is associative. A MTC is semisimple: simple objects $L_a$ are the ones whose only endomorphisms are multiples of the identity, and a category is semisimple if every object is the direct sum of finitely many simple objects. We label the simple objects by $a, b, c, \dots$ (comparing with previous sections, they are the lines $(\lambda, \bar{\mu})$ discussed before). Morphisms from $L_a \otimes L_b$ to $L_c$ form a vector space $V_{ab}^c = \mathrm{Hom}(L_a \otimes L_b, L_c)$ over $\mathbb{C}$, and we label a basis of it by the index $\mu_{ab}^c$. The number of independent morphisms is the fusion coefficient $N_{ab}^c = \dim V_{ab}^c$, which is required to be symmetric in $ab$.

For each object $L$ there is a dual object $\check{L}$, associated to the conjugate representation of chiral algebra, such that the product of $L$ and $\check{L}$ contains the trivial line: $L \otimes \check{L} = L_0 + \dots$

---

[37]Coset models can also be described in this way [55], but we will not consider them here.

[38]For instance, the $SU(2)_k$ CS theory has $k+1$ lines $L_\lambda$ labeled by the Dynkin label $\lambda \in \{0, 1, \dots, k\}$. The 1-form symmetry group is $\mathbb{Z}_2 = \{L_0, L_k\}$. In $SU(2)_k \times SU(2)_{-k}$, the diagonal $\mathbb{Z}_2$ given by $(L_0, L_0)$ and $(L_k, L_k)$ is non-anomalous and can be gauged, however the resulting theory is a non-trivial TQFT.

(notice that $\check{L}_0 = L_0$). Dual objects allow us to define dual morphisms from $L_c$ to $L_a \otimes L_b$. Morphisms and dual morphisms are associated to trivalent junctions of lines:

$$\mu_{ab}^c \qquad \qquad \mu_c^{ab} \tag{4.1}$$

The dimensions of the morphism vector spaces satisfy $N_{ab}^c = N_{\check{a}c}^b$ as well as the other relations that follow from symmetry. Morphisms admit orthogonality and completeness relations:

$$= \delta_{ab}\,\delta_{\mu\mu'} \qquad \qquad = \sum_c \sum_{\mu \in V_{ab}^c} \tag{4.2}$$

On the left, $\delta_{\mu\mu'}$ can be non-vanishing only if $N_{cd}^a > 0$. Besides, one can define a trace and associate to each object $L$ a number called the quantum dimension and denoted by $\dim L$:

$$L \qquad = \mathrm{Tr}(\mathrm{id}_L) = \dim L = \dim \check{L} \,. \tag{4.3}$$

For simple lines we introduce the notation $d_a = \dim L_a$. Such quantum dimensions are completely fixed by $N_{ab}^c$. The total dimension of the category is $\dim \mathcal{C} \equiv \mathcal{D} = \sqrt{\sum_a d_a^2}$.

The MTC is characterized by braiding and fusion matrices, that define linear relations between isomorphic spaces of morphisms. The brading matrix $R$ is defined as

$$= \sum_{v \in V_{ab}^c} [R_c^{ab}]_{\mu v} \tag{4.4}$$

while the fusion matrix $F$ is defined as[39]

$$= \sum_f \sum_{\substack{\rho \in V_{ab}^f \\ \sigma \in V_{fc}^d}} [F_d^{abc}]_{(e;\mu v)(f;\rho\sigma)} \tag{4.7}$$

---

[39]One can also define the inverse $G$ of the fusion matrix $F$, satisfying

$$\sum_{f,\alpha\beta} [F_d^{abc}]_{(e;\mu v)(f;\alpha\beta)} [G_d^{abc}]_{(f;\alpha\beta)(g;\rho\sigma)} = \delta_{eg}\delta_{\mu\rho}\delta_{v\sigma} \,. \tag{4.5}$$

One can show that $G$ is expressed in terms of $F$ as

$$[G_d^{abc}]_{(e;\mu v)(f;\rho\sigma)} = \sum_{\alpha\beta\gamma\delta} [R_e^{ab}]_{\mu\alpha}^{-1}[R_d^{ec}]_{v\beta}^{-1}[F_d^{cba}]_{(e;\alpha\beta)(f;\gamma\delta)}[R_f^{bc}]_{\gamma\rho}[R_d^{af}]_{\delta\sigma} \,. \tag{4.6}$$

and it encodes associativity of the fusion product $a \otimes b \otimes c$. These matrices satisfy several consistency conditions [52]. One set is given by the pentagon equations

$$\sum_{\lambda}[F_e^{abf}]_{(g;\beta\gamma)(\ell;\sigma\lambda)}[F_e^{\ell cd}]_{(f;\alpha\lambda)(k;\psi\rho)} = \sum_{h}\sum_{\delta\mu\nu}[F_k^{abc}]_{(h;\delta\nu)(\ell;\sigma\psi)}[F_e^{ahd}]_{(g;\mu\gamma)(k;\nu\rho)}[F_g^{bcd}]_{(f;\alpha\beta)(h;\delta\mu)}. \quad (4.8)$$

The indices of morphisms run over the corresponding spaces (when they are not vanishing), in particular $\lambda \in V_{\ell f}^e$, $\delta \in V_{bc}^h$, $\mu \in V_{hd}^g$, and $\nu \in V_{ah}^k$. Another set is given by the hexagon equations:

$$\sum_{\gamma\delta}[R_e^{ac}]_{\alpha\gamma}[F_d^{bac}]_{(e;\gamma\beta)(f;\delta\lambda)}[R_f^{ab}]_{\delta\mu} = \sum_{g}\sum_{\nu\rho\sigma}[F_d^{bca}]_{(e;\alpha\beta)(g;\nu\rho)}[R_d^{ag}]_{\rho\sigma}[F_d^{abc}]_{(g;\nu\sigma)(f;\mu\lambda)}. \quad (4.9)$$

Here $\gamma \in V_{ac}^e$, $\delta \in V_{ba}^f$, $\nu \in V_{bc}^g$, $\rho \in V_{ga}^d$, and $\sigma \in V_{ag}^d$.

The matrices $R$ and $F$ are "gauge dependent", in the sense that they depend on a choice of basis in the vector spaces of morphisms. Two important gauge-invariant quantities are the topological spin $\theta_a$ and the $S$-matrix.[40] The former is defined by

$$\begin{matrix} & = & \theta_a & \text{or} & \theta_a = \theta_{\check a} = \frac{1}{d_a} \end{matrix}, \quad (4.10)$$

it is a phase, $|\theta_a| = 1$, and it satisfies

$$\theta_a = \frac{d_a[F_a^{a\check aa}]_{00}}{[R_0^{a\check a}]} = \sum_{c,\mu}\frac{d_c}{d_a}[R_c^{aa}]_{\mu\mu}, \qquad \sum_{\nu}[R_c^{ab}]_{\mu\nu}[R_c^{ba}]_{\nu\rho} = \frac{\theta_c}{\theta_a\theta_b}\delta_{\mu\rho}. \quad (4.11)$$

The absolute value of the first relation provides an alternative formula for $d_a$. The $S$-matrix is defined by

$$S_{ab} = \frac{1}{\mathcal{D}} \quad a \quad b \quad = \frac{1}{\mathcal{D}}\sum_c N_{\check ab}^c d_c \frac{\theta_c}{\theta_a\theta_b}, \quad (4.12)$$

which implies

$$a \quad \begin{matrix} \\ b \end{matrix} \quad = \quad \frac{S_{ab}}{S_{0b}} \quad \begin{matrix} \\ b \end{matrix}. \quad (4.13)$$

In particular $S_{00} = 1/\mathcal{D}$ and $S_{0a} = d_a/\mathcal{D}$. One can prove that $S_{ab} = S_{ba} = S_{\check ab}^*$ and that $S$ is a unitary matrix. The Verlinde formula [93] is

$$N_{ab}^c = \sum_{x \in \mathcal{C}}\frac{S_{ax}S_{bx}S_{cx}^*}{S_{0x}}. \quad (4.14)$$

Together with the matrix $T_{ab} = e^{-2\pi i c_-/24}\theta_a\delta_{ab}$ and the charge-conjugation matrix $C_{ab} = \delta_{a\check b}$, the $S$-matrix obeys the modular relations

$$(ST)^3 = S^2 = C, \qquad C^2 = \mathbb{1}. \quad (4.15)$$

The chiral central charge $c_-$, defined modulo 8, is given by $\frac{1}{\mathcal{D}}\sum_a \theta_a d_a^2 = e^{2\pi i c_-/8}$.

---

[40]Another one is the Frobenius-Schur indicator $\varkappa_a = d_a[F_a^{aaa}]_{00} = \pm 1$ for lines such that $\check a = a$.

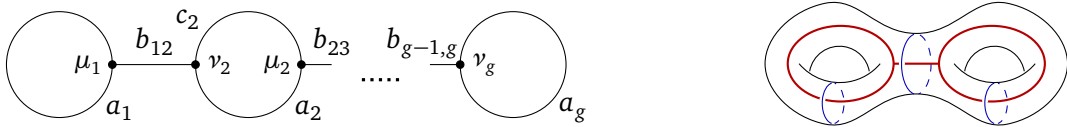

Figure 5: Left: a basis of line insertions in a genus-$g$ handlebody that generate the Hilbert space of states on the boundary Riemann surface $\Sigma_g$. Right: example of insertions for $g = 2$.

The Hilbert space $\mathcal{H}_g$ of the TQFT on a Riemann surface $\Sigma_g$ of genus $g$ is generated by a handlebody $S\Sigma_g$ with line insertions. A basis for the Hilbert space is provided by the insertion of simple lines $a_1, \ldots, a_g, c_2, \ldots, c_{g-1}$ along the non-contractible cycles (similarly to the Abelian case), together with $g - 1$ lines $b_{12}, b_{23}, \ldots, b_{g-1,g}$ such that $b_{i,i+1}$ connects the $i$-th to the $(i + 1)$-th non-contractible cycle and at each junction the homomorphism space is non-trivial. See Figure 5. The total number of states is thus

$$\dim \mathcal{H}_g = \sum_{\substack{a_1, \ldots, a_g \in \mathcal{C} \\ c_2, \ldots, c_{g-1} \in \mathcal{C}}} \sum_{b_{12}, \ldots, b_{g-1,g} \in \mathcal{C}} N_{a_1 \breve{a}_1}^{b_{12}} N_{a_2 c_2}^{b_{12}} N_{a_2 c_2}^{b_{23}} \cdots N_{a_{g-1} c_{g-1}}^{b_{g-1,g}} N_{a_g \breve{a}_g}^{b_{g-1,g}} . \tag{4.16}$$

Using Verlinde's formula, this reduces to

$$\dim \mathcal{H}_g = \mathcal{D}^{2g-2} \sum_{a \in \mathcal{C}} d_a^{2-2g} . \tag{4.17}$$

**Algebra objects and anyon condensation.** In the case of MTCs whose lines are not all Abelian, gauging a one-form symmetry is not enough to make the theory trivial in the bulk (and thus achieve a good holographic dual to a boundary CFT). Instead, one should invoke a procedure called *anyon condesation* which is the analog of gauging in the case of non-invertible symmetries. We keep following the notation of [57], and refer to [42] for details on condensation. The procedure can be summarized as follows.

First, one packages a set of lines (possibly with degeneracies) into a composite line $\mathcal{A}$:

$$\mathcal{A} = \sum_{a \in \mathcal{C}} Z_a^{\mathcal{A}} L_a , \qquad Z_a^{\mathcal{A}} \in \mathbb{Z}_{\geq 0} . \tag{4.18}$$

What substitutes the group structure is a morphism $m : \mathcal{A} \otimes \mathcal{A} \to \mathcal{A}$, which we denote by a red dot. The trivial line is required to be inside $\mathcal{A}$, and this is represented by a unit morphism $\eta : L_0 \to \mathcal{A}$:

$$\tag{4.19}$$

In components, $m$ is represented by a tensor $m_{a\alpha,b\beta}^{c\gamma;\mu}$ as follows:

$$\tag{4.20}$$

Here $\alpha = 1,\ldots,Z_a^{\mathcal{A}}$ labels a choice of basis vectors in $\mathrm{Hom}(L_a,\mathcal{A})$ (and similarly for $\beta,\gamma$), and the blue cup is the projector from $\mathcal{A}$ to the corresponding vector. The projectors satisfy orthogonality and completeness relations:

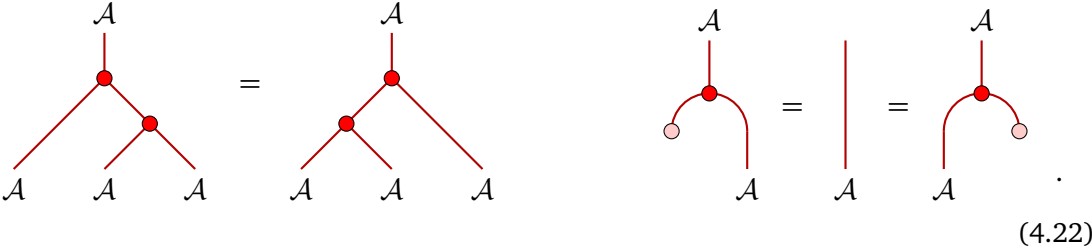

$$\tag{4.21}$$

The morphism $m$ is required to be associative and compatible with the unit morphism:

$$\tag{4.22}$$

The pictures corresponds to the equations

$$\sum_{\varphi=1}^{Z_f^{\mathcal{A}}} m_{a\alpha,b\beta}^{f\varphi;\rho} \, m_{f\varphi,c\gamma}^{d\delta;\sigma} = \sum_{e\in\mathcal{C}}\sum_{\varepsilon=1}^{Z_e^{\mathcal{A}}}\sum_{\mu\nu} m_{a\alpha,e\varepsilon}^{d\delta;\nu} \, m_{b\beta,c\gamma}^{e\varepsilon;\mu} \, [F_d^{abc}]_{(e;\mu\nu)(f;\rho\sigma)}\,, \tag{4.23}$$

$$\sum_{\gamma=1}^{Z_0^{\mathcal{A}}}\sum_{\mu} \eta_\gamma \, m_{0\gamma,b\beta}^{a\alpha;\mu} = \sum_{\gamma=1}^{Z_0^{\mathcal{A}}}\sum_{\mu} \eta_\gamma \, m_{b\beta,0\gamma}^{a\alpha;\mu} = \delta_{ab}\delta_{\alpha\beta}\,, \tag{4.24}$$

where $\eta_\gamma$ are the components of the unit morphism. These properties define a so-called *algebra object* $\mathcal{A}$. One requires it to be *haploid*, or *connected*: there is a unique copy of the trivial line inside $\mathcal{A}$, *i.e.*, $\dim\mathrm{Hom}(L_0,\mathcal{A}) = Z_0^{\mathcal{A}} = 1$.

Besides $m$, one also defines an associative co-morphism $\Delta : \mathcal{A} \to \mathcal{A}\otimes\mathcal{A}$ and a co-unit $\epsilon : \mathcal{A} \to L_0$, making $\mathcal{A}$ into a co-algebra as well. One requires $\mathcal{A}$ to be *special*: in a normalization of $\Delta$ such that $m\circ\Delta = \mathrm{id}_{\mathcal{A}}$, the condition is

$$\boxed{\phantom{XXX}} = \dim\mathcal{A} = \sum_{a\in\mathcal{C}} Z_a^{\mathcal{A}} d_a\,. \tag{4.25}$$

One requires the Frobenius condition, that allows one to relate morphisms and co-morphisms by crossing:[41]

$$\tag{4.27}$$

---

[41]In components, the relations read:

$$\sum_{e,\varepsilon}\sum_{\mu\nu} \Delta_{a\alpha;\mu}^{c\gamma,e\varepsilon} \, m_{e\varepsilon,b\beta}^{d\delta;\nu} \, [F_f^{ceb}]_{(d;\nu\sigma)(a;\mu\rho)} = \sum_{\phi} m_{a\alpha,b\beta}^{f\phi;\rho} \, \Delta_{f\phi;\sigma}^{c\gamma,d\delta} = \sum_{e,\varepsilon}\sum_{\mu\nu} \Delta_{b\beta;\mu}^{e\varepsilon,d\delta} \, m_{a\alpha,e\varepsilon}^{c\gamma;\nu} \, [G_f^{aed}]_{(c;\nu\sigma)(b;\mu\rho)}\,. \tag{4.26}$$

These properties define a special Frobenius algebra object $\mathcal{A}$.

As opposed to the Abelian case, the line $\mathcal{A}$ is in general not transparent with respect to itself. Instead, anomaly cancellation corresponds to the condition

$$(4.28)$$

which in components reads

$$m^{c\gamma;\nu}_{a\alpha,b\beta} = \sum_{\mu \in V^c_{ba}} m^{c\gamma;\mu}_{b\beta,a\alpha} \, [R^{ab}_c]_{\mu\nu} \,. \tag{4.29}$$

An algebra with this property is called *commutative*. Therefore, a gaugeable (or condensable) algebra $\mathcal{A}$ — the non-invertible analog of a non-anomalous 1-form symmetry — is a connected commutative special Frobenius algebra, or equivalently a connected commutative separable algebra, in $\mathcal{C}$ [42].

The partition function of the "gauged theory" after anyon condensation on an orientable 3-manifold $M$ is defined as the partition function of the original theory with the insertion of a fine mesh of lines $\mathcal{A}$ along the dual edges of a triangulation of $M$, with morphisms $m$ or $\Delta$ at the vertices (see for instance Figure 7).[42] Only in the case that $\mathcal{A}$ is Abelian (*i.e.*, that all lines $L_a$ appearing in the decomposition (4.18) are Abelian and $Z^{\mathcal{A}}_a \in \{0,1\}$), a fine mesh can be reduced to a copy of $H_1(M;\mathcal{A})$ — and the normalization factors appearing in (2.32) are reproduced. We give more details on the Abelian case below. When $M$ has boundaries with Dirichelet boundary conditions, there are dual edges of the triangulation that terminate at the boundary (inside the faces that triangulate it), and one should fix projection maps for all of them. To compute partition functions, one imposes trivial boundary conditions, as in Sections 2.3 and 2.4: one terminates each boundary dual edge with a unit/co-unit morphism $\eta$ (4.19) and includes a boundary term $(\dim\mathcal{A})^{-\chi(\text{boundary})/2}$ in order to normalize to 1 the $S^2$ partition function of the boundary theory. More general boundary conditions come into play when computing correlation functions of local operators placed at the boundary (as discussed in Section 4.3). In this case, one considers a triangulation such that each insertion point is inside a boundary face (and no face contains more than one point). The dual edges of the triangulation can end at those points, and for each of them one chooses a projector map $\alpha$, as in (4.20), from $\mathcal{A}$ to the representation $L_a$ of chiral algebra (with $Z^{\mathcal{A}}_a > 0$) that corresponds to the inserted 2d primary operator. When $Z^{\mathcal{A}}_a > 1$ there are multiple choices for the projector, as there exist multiple operators in the same representation $L_a$ in the spectrum of the RCFT.

The lines of the gauged theory are local $\mathcal{A}$-modules $\mathcal{M}$, and their category $\mathcal{C}^{\text{loc}}_{\mathcal{A}}$ inherits a MTC structure from $\mathcal{C}$. A (left) $\mathcal{A}$-module is a pair $(\mathcal{M}, \rho_{\mathcal{M}})$ of an object $\mathcal{M} \in \mathcal{C}$ and an action $\rho_{\mathcal{M}} : \mathcal{A} \otimes \mathcal{M} \to \mathcal{M}$ (below on the left), satisfying the compatibility condition in the middle of

---

[42]The dual edges of a triangulation have 4-valent vertices, each of which should be resolved into two 3-valent vertices. Associativity, commutativity, and the Frobenius property guarantee that the result is independent from the resolution, as well as from the chosen triangulation. Note that, since $\mathcal{A}$ is not transparent with respect to itself, generic meshes not dual to a triangulation would lead to different (and inconsistent) results.

the following picture:

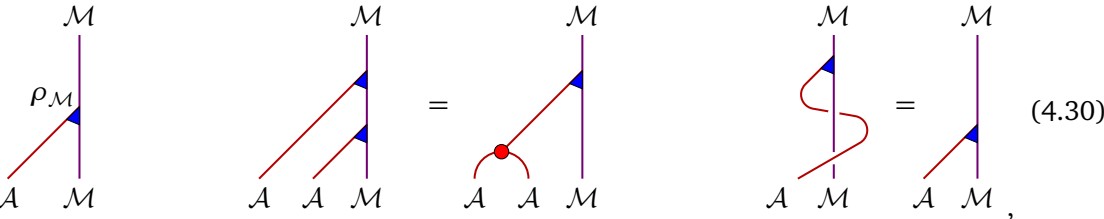

$$(4.30)$$

as well as the condition $\rho_{\mathcal{M}} \circ (\eta \otimes \mathrm{id}_{\mathcal{M}}) = \mathrm{id}_{\mathcal{M}}$. The module is local if it has trivial braiding with $\mathcal{A}$, as on the right in the picture above. This allows one to define a canonical right action in terms of the left action,

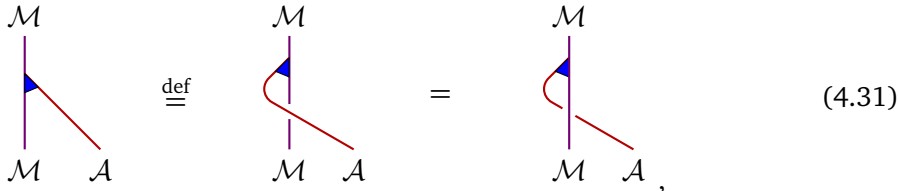

$$(4.31)$$

and, by virtue of the commutativity of $\mathcal{A}$, the two actions commute. This makes $\mathcal{M}$ into an $\mathcal{A}$-bimodule as well. Notice that $(\mathcal{A}, m)$ is always a local $\mathcal{A}$-module, which corresponds to the trivial line of $\mathcal{C}_{\mathcal{A}}^{\mathrm{loc}}$ and of the gauged theory. The total dimension of the category $\mathcal{C}_{\mathcal{A}}^{\mathrm{loc}}$ is $\dim \mathcal{C}_{\mathcal{A}}^{\mathrm{loc}} = \mathcal{D}/\dim \mathcal{A}$ [100, 101]. Hence, the gauged theory is trivial in the bulk if and only if

$$\dim \mathcal{A} = \mathcal{D}. \tag{4.32}$$

This condition defines a Lagrangian algebra object (or anyon) $\mathcal{A}$.

One can prove that a condensable anyon has trivial spin,[43] in other words

$$\theta_a = 1 \qquad \forall\, a \text{ such that } Z_a^{\mathcal{A}} > 0. \tag{4.33}$$

Moreover, a Lagrangian condensable anyon is modular invariant [57, 67]:

$$\sum_{b \in \mathcal{C}} S_{ab} Z_b^{\mathcal{A}} = Z_a^{\mathcal{A}}. \tag{4.34}$$

**Gauss law.** Given a commuting special Frobenius algebra $\mathcal{A}$, the effect of its condensation is encoded in a projector[44] $P_{\mathcal{A}}(L) \in \mathrm{Hom}(\mathcal{A} \otimes L, \mathcal{A} \otimes L)$. In the following, we will only apply it to simple lines $L_b$ and write $P_{\mathcal{A}}(b)$. The projector is defined as

$$P_{\mathcal{A}}(b) \equiv \qquad = \tag{4.35}$$

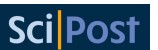

---

[43]One considers a loop as in (4.10), but where the anyon running in the loop is $\mathcal{A}$. On the one hand, $\mathcal{A}$ can be decomposed into simple lines. On the other hand, the properties of a special Frobenius algebra together with commutativity allow one to untwist the loop. This leads to the equation $\sum_a Z_a^{\mathcal{A}} d_a \theta_a = \sum_a Z_a^{\mathcal{A}} d_a$.

[44]This projector was defined in [57] for a generic Frobenius algebra object, but it simplifies in the commuting case.

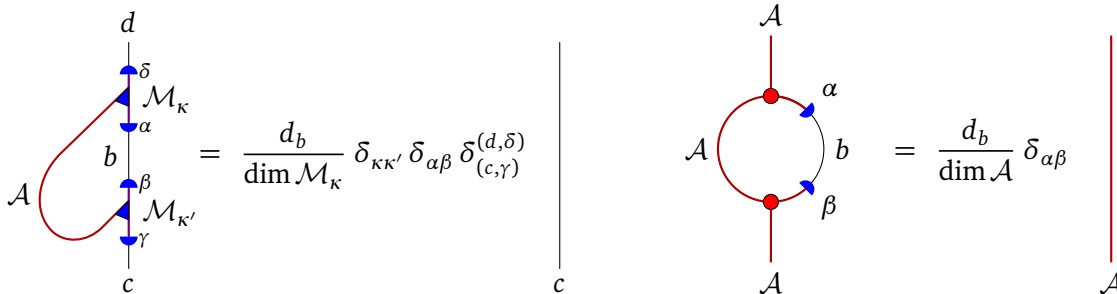

Figure 6: Left: orthogonality of $\mathcal{A}$-modules. The index $\kappa$ labels different modules. Right: this relation is obtained from the left by setting $\mathcal{M}_\kappa = \mathcal{M}_{\kappa'} = \mathcal{A}$ and $\rho_\mathcal{M} = m$, composing with the projections $\gamma$ below and $\delta$ above, and then summing over $c, \gamma, d, \delta$.

and is easily seen to satisfy $P_\mathcal{A}(b) \circ P_\mathcal{A}(b) = P_\mathcal{A}(b)$ [57]. In the case that $\mathcal{A}$ is Lagrangian, using modular invariance of the vector $Z_a^\mathcal{A}$, one computes the trace of $P_\mathcal{A}(b)$:

$$\operatorname{Tr} P_\mathcal{A}(b) = \quad = \quad = \mathcal{D} Z_b^\mathcal{A}. \qquad (4.36)$$

This shows that $P_\mathcal{A}$ only has support on the lines $L_b \in \mathcal{A}$.

For Lagrangian $\mathcal{A}$, the projector can equivalently be represented as

$$P_\mathcal{A}(b) = \frac{\dim \mathcal{A}}{d_b} \sum_{\beta=1}^{Z_b^\mathcal{A}} \qquad . \qquad (4.37)$$

To prove it, one uses the orthogonality between $\mathcal{A}$-modules [57], depicted in Figure 6 on the left. Specializing to the case that the two modules are $(\mathcal{A}, m)$, composing with projection maps $\delta$ and $\gamma$ above and below, respectively, and summing over them, one obtains the relation in Figure 6 on the right. Let us call $\widehat{P}_\mathcal{A}(b)$ the operator in (4.37) and relax the assumption that $\mathcal{A}$ is Lagrangian. The relation in Figure 6 implies that $\widehat{P}_\mathcal{A}(b)$ is a projector, $\widehat{P}_\mathcal{A}(b) \circ \widehat{P}_\mathcal{A}(b) = \widehat{P}_\mathcal{A}(b)$. One also easily shows that placing the operator $\widehat{P}_\mathcal{A}(b)$ along a 1-cycle is equivalent to placing the line $Z_b^\mathcal{A} \mathcal{A}$, that $\operatorname{Tr} \widehat{P}_\mathcal{A}(b) = Z_b^\mathcal{A} \dim \mathcal{A}$, as well as that $\mathcal{P}_\mathcal{A}(b) \circ \widehat{\mathcal{P}}_\mathcal{A}(b) = \widehat{\mathcal{P}}_\mathcal{A}(b) \circ \mathcal{P}_\mathcal{A}(b) = \widehat{\mathcal{P}}_\mathcal{A}(b)$. In the case that $\mathcal{A}$ is Lagrangian, they imply that $\widehat{P}_\mathcal{A}(b) = P_\mathcal{A}(b)$ [102].

**The Abelian case.** A line $L_a$ is said to be Abelian when $d_a = 1$. Abelian lines have unique fusion with all other lines, namely $\sum_c N_{ab}^c = 1$ for all $b$. The set of Abelian lines forms a finite Abelian group under fusion: the 1-form symmetry group of the bulk theory. An algebra object $\mathcal{A}$ is said to be invertible when it is made of Abelian lines with no degeneracies, namely when $Z_a^\mathcal{A} \in \{0, 1\}$ and $d_a Z_a^\mathcal{A} = Z_a^\mathcal{A}$. In this case the components $L_i$ of $\mathcal{A}$ form an Abelian subgroup of the total 1-form symmetry group, that we also denote as $\mathcal{A}$. The morphism components reduce to $m_{ij}^k$, which can be non-zero only when $i \otimes j = k$. We can thus regard them as a normalized group-cohomology 2-cochain $m \in C^2(\mathcal{A}; \mathbb{C}^*)$ labelled by $ij \in \mathcal{A} \otimes \mathcal{A}$, with $m_{0a}^a = m_{a0}^a$. If we

regard $[F_\ell^{ijk}]_{mn}$ as a 3-cochain labelled by $ijk$ (because $m = j \otimes k$, $n = i \otimes j$ and $\ell = i \otimes j \otimes k$), then the pentagon equation (4.8) reduces to $dF = 1$ (using multiplicative notation), while the associativity condition (4.23) reduces to

$$F = (dm)^{-1}, \tag{4.38}$$

namely $F$ must be trivial in $H^3(\mathcal{A}; \mathbb{C}^*)$. If we regard $[R_k^{ij}]$ as a 2-cochain labelled by $ij$, then the commutativity condition (4.28) reduces to $R_k^{ij} = m_{ij}^k / m_{ji}^k$ and one easily checks that the anomaly cancelation condition $\theta_i = 1$ is satisfied.

One can prove the following relation for invertible special Frobenius algebras $\mathcal{A}$:

$$\begin{array}{ccc} \text{[diagram]} & = & \dfrac{1}{\dim \mathcal{A}} \quad \text{[diagram]} \end{array} \tag{4.39}$$

The dotted notation means that the line should close, however there could be other lines passing in the middle, or the line could close through a topologically non-trivial 1-cycle. The relation holds because, in the invertible case, only the identity can run along the horizontal channel. Applying the relation to $P_\mathcal{A}(b)$ we immediately reproduce the construction we used in the Abelian case, in Section 2.4. In particular, $P_\mathcal{A}(b) = Z_b^\mathcal{A} \, \mathrm{id}_\mathcal{A} \otimes \mathrm{id}_b$.

## 4.2 Partition functions and factorization

There is a correspondence between RCFTs with a given chiral algebra, and condensable Lagrangian anyons in the MTC $\mathcal{C} = \mathcal{C}_L \times \overline{\mathcal{C}_R}$, where $\mathcal{C}_L$ and $\overline{\mathcal{C}}_R$ (the overline stands for orientation reversal) correspond to the left and right part of the chiral algebra (that in general could be different), respectively [53, 57, 59, 102]. On the one hand, given a (bosonic) RCFT, its torus partition function takes the form

$$Z_{T^2}^\mathcal{A} = \sum_{\mu \in \mathcal{C}_L, \ \bar{\nu} \in \mathcal{C}_R} Z_{\mu\bar{\nu}}^\mathcal{A} \, \chi_\mu(\tau) \, \chi_{\bar{\nu}}(\bar{\tau}), \tag{4.40}$$

where $Z_{\mu\bar{\nu}}^\mathcal{A}$ are non-negative integers while $\chi$ are characters. One can prove that

$$\mathcal{A} \equiv \sum_{\mu \in \mathcal{C}_L, \ \bar{\nu} \in \mathcal{C}_R} Z_{\mu\bar{\nu}}^\mathcal{A} \, L_\mu \otimes L_{\bar{\nu}}, \tag{4.41}$$

is a condensable Lagrangian anyon, with a suitable morphism $m$. With respect to our previous notation, simple lines are labeled by $a = (\mu, \bar{\nu})$. For instance, modular invariance of the torus partition function, $\sum_{a \in \mathcal{C}} S_{ab} Z_b^\mathcal{A} = Z_a^\mathcal{A}$, and uniqueness of the vacuum, $Z_0^\mathcal{A} = 1$ where the index $0 = (0,0)$, imply that $\mathcal{A}$ is Lagrangian (recall that $d_a = \mathcal{D} S_{0a}$):

$$\dim \mathcal{A} = \sum_{a \in \mathcal{C}} Z_a^\mathcal{A} d_a = \mathcal{D}. \tag{4.42}$$

Commutativity, instead, follows from the fact that physical conformal primaries $\mathcal{O}_a$, with $a \subset \mathcal{A}$, of the RCFT have trivial braiding.

On the contrary, consider a bulk Chern-Simons theory $G_k \times G_{-k}$ (that can be generalized in various ways) with a given condensable Lagrangian line $\mathcal{A}$ as in (4.41). We want to determine the partition functions of the theory $\mathcal{T}$ obtained after condensation of $\mathcal{A}$, on oriented three-manifolds with boundaries and holomorphic/anti-holomorphic boundary conditions for $G_k$ and $G_{-k}$, respectively. Let us start from handlebodies.

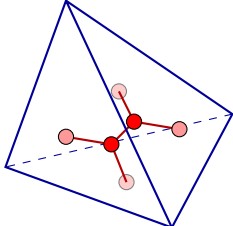
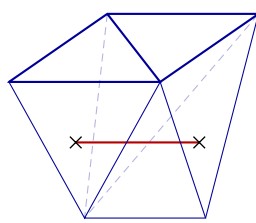

Figure 7: Left: triangulation of the ball $\mathsf{S}S^2 = D_3$ by a single tetrahedron, and dual network. Right: triangulation of the solid torus $\mathsf{S}T^2$ by three tetrahedra. We identify the right face with the left face, and the two faces in the front with the two in the back, while the two upper faces triangulate the boundary $T^2$. The dual network has been drawn only after simplification, and it wraps the non-contractible cycle.

The case of $\mathsf{S}S^2 = D_3$ is simple: this geometry can be triangulated by a single tetrahedron, and its faces triangulate the boundary $S^2$ (Figure 7 on the left). The dual network reduces to the segment in (4.25) on the right, equal to $\dim \mathcal{A} = \mathcal{D}$. The Chern-Simons partition function is $Z_{\mathrm{CS}}[D_3] = 1$, therefore, in order to normalize to 1 the $S^2$ partition function of the boundary theory, we include a boundary counterterm[45] $(\dim \mathcal{A})^{-\chi(\text{boundary})/2}$. This gives

$$Z_{\mathcal{T}}[D_3] = Z_{\mathrm{CS}}[D_3] = 1 \ . \tag{4.43}$$

The solid torus $\mathsf{S}T^2$ can be triangulated by three tetrahedra (with suitable gluing of the faces, see Figure 7 on the right), and two faces triangulate the boundary $T^2$. The dual network reduces to a loop of $\mathcal{A}$ along the non-contractible cycle. The partition function of Chern-Simons theory on the solid torus with holomorphic boundary conditions and a simple line insertion reproduces the corresponding character of chiral algebra (or extensions thereof, if $G$ is not simply connected) [56, 77, 79]. This has been shown very explicitly in [79]. The partition function of the gauged theory, thus, reproduces the torus partition function (4.40).

Triangulation of a handlebody $\mathsf{S}\Sigma_g$ produces (after simplifications) a dual network as in Figure 5 on right, in which the lines are $\mathcal{A}$ and the (co)morphisms are $m$ or $\Delta$. Once again, the partition function of the three-dimensional gauged theory $\mathcal{T}$, with holomorphic boundary conditions, is equal to the partition function of the RCFT on $\Sigma_g$. Using the identity on the right of (4.21) on each segment of the line $\mathcal{A}$, and the definition (4.20) of the components of $m$ (and the similar definition for $\Delta$), the network gets rewritten in terms of the insertions in Figure 5 on the left, with: $(i)$ for each segment of $\mathcal{A}$, a sum over representations $a \in \mathcal{A}$ and a sum over projectors $\alpha : \mathcal{A} \to L_a$ at the two ends; $(ii)$ for each junction between lines with labels $a, b, c$, a sum over morphisms $\mu \in V_{ab}^c$ (or co-morphisms $\mu \in V_c^{ab}$, depending on orientation). Each term in these sums contains a coefficient $m_{a\alpha, b\beta}^{c\gamma;\mu}$ for each morphism and $\Delta_{c\gamma;\mu}^{a\alpha, b\beta}$ for each co-morphism.[46] The partition function of $G_k \times G_{-k}$ Chern-Simons theory with the insertions of Figure 5 (left) gives the (product of left and right) conformal blocks of the chiral algebra on $\Sigma_g$ [44]. The conformal blocks are labeled by the primaries $a, \ldots$ and the morphisms $\mu, \ldots$ in Figure 5, while do not depend on the projector labels $\alpha, \ldots$ which instead reflect the multiplicities of chiral algebra representations in the operator content of the RCFT (if

[45]The choice of counterterm is related to the choice of normalizations in (4.25). For instance, in a normalization in which the bubble is equal to $\dim \mathcal{A}$ while the segment is equal to 1, the boundary counterterm would be trivial, but there would be a bulk counterterm to ensure independence from the triangulation.

[46]Each segment connecting the loops in the network of Figure 5 will have a morphism at one junction, and a co-morphism at the other junction. The choice of normalization (4.25) for the bubble fixes the normalization of the components of $\Delta$ in terms of those of $m$. Since we have $g - 1$ insertions of $\Delta$, we see that this normalization enters with a power of $g - 1$, and is related to the Euler counterterm $(\dim \mathcal{A})^{1-g}$.

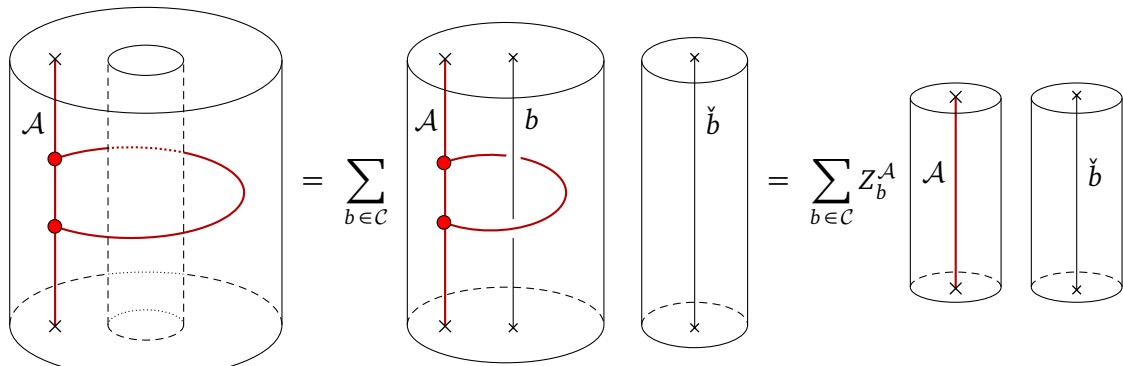

Figure 8: Factorization of the $T^2 \times I$ partition function of the gauged theory $\mathcal{T}$. Left: the $T^2 \times I$ geometry (the upper and lower faces are indentified in all pictures) with the network of algebra object $\mathcal{A}$ inserted. Middle: two solid tori, obtained by inserting the completeness of states at some point along $I$. We recognise the projector $\mathcal{P}_{\mathcal{A}}(b)$. Right: The $\mathsf{S}T^2 \times \mathsf{S}T^2$ partition function of $\mathcal{T}$. This picture is the non-Abelian analog of Figure 3.

one works with the maximally extended chiral algebra, then there are no multiplicities [53]). What we described from the condensation of $\mathcal{A}$ are precisely the ingredients of the partition function of the RCFT on $\Sigma_g$ [52–54] (in the Abelian case one recovers the construction of higher-genus partition functions of, *e.g.*, [74] that we reviewed in Section 2.1). Indeed, when the bulk geometry is a handlebody, it is always possible to pull the condensation network of $\mathcal{A}$ to the boundary: the lines become Verlinde lines [93] and one reduces to the pair-of-pants decomposition of 2d partition functions.

The partition functions of $\mathcal{T}$ on general oriented three-dimensional manifolds $M$ can be computed using the fact that $\mathcal{T}$ is completely trivial in the bulk — the Hilbert space on any closed two-dimensional spatial manifold is one-dimensional, and the partition function on any closed three-manifold is 1 — and then applying the very same formal argument that led to (2.53) and depicted in Figure 4. As in the Abelian case, it implies that:

- The partition function only depends on the boundary conditions, and not on the particular 3d geometry that is used in the bulk. In particular, such a background independence implies that no sum over geometries should be performed.

- When the boundary consists of a connected Riemann surface $\Sigma_g$, the partition function is the one computed by the corresponding handlebody. When the boundary consists of multiple components, the partition function factorizes as the product of partition functions on handlebodies, as in (2.53).

Alternatively, one can exhibit factorization by explicitly constructing a triangulation of the multi-boundary geometry and computing the expectation value of the dual network of $\mathcal{A}$. Let us show how this works for the simple case of the $T^2 \times I$ Euclidean wormhole, confirming that it factorizes into the product of two partition functions on solid tori, as expected from the point of view of the boundary theory. The manifold $T^2 \times I$ is an annulus times a circle, and we represent it on the LHS of Figure 8. In the same figure we indicate the network of $\mathcal{A}$ produced by a triangulation [57]. We cut $T^2 \times I$ along a $T^2$ at some point of the interval $I$, and insert a complete basis of states on $T^2$ represented by solid tori with the insertion of lines $L_b$. This produces the expression in the middle of Figure 8, in which the projector $\mathcal{P}_{\mathcal{A}}(b)$ of (4.35) appears. Using the expression (4.37) for $\mathcal{P}_{\mathcal{A}}(b)$, and its reduction to $Z_b^A \mathcal{A}$ when placed on a

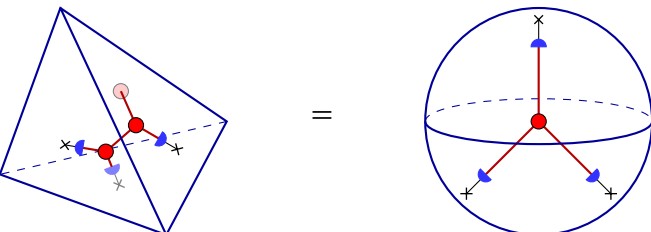

Figure 9: Triangulation of the ball $SS^2 = D_3$ with three local operator insertions on the boundary. It is triangulated by a single tetrahedron, with three of the boundary faces containing an insertion point. The dual triangulation is attached to those faces via a projector from $\mathcal{A}$ to the chiral algebra representation of the corresponding operator. The network that computes the three-point function can be simplified as shown on the right.

cycle, we finally obtain the RHS of Figure 8, which is the partition function of $\mathcal{T}$ on $ST^2 \times ST^2$. A similar argument may be extended to the more general case $\Sigma_g \times I$ (see Appendix D).

## 4.3 Correlators

Let us now discuss how to obtain correlation functions of the RCFT using the dual description in terms of the topological theory $\mathcal{T}$ after condensation of $\mathcal{A}$. The physical primary operators are labeled by the representations $a$ of the (left and right) chiral algebra with $Z_a^{\mathcal{A}} > 0$, and in addition by the projectors $\alpha$ from $\mathcal{A}$ to $L_a$, that run over the $Z_a^{\mathcal{A}}$ copies of the representation in the physical spectrum of the RCFT. In order to obtain correlations functions, as explained in Section 4.1, we take a fine enough triangulation of the bulk three-manifold $M$ such that each boundary local operator is inside one boundary face, and no face contains more than one operator. One of the edges of the dual triangulation will terminate on the local operator $(a, \alpha)$ at the boundary, and the prescription is to place the projector $\alpha$ at that point (while edges landing at boundary points where no operator is inserted, terminate with the unit projector $\eta$). This is depicted in Figure 9 for the case of a three-point function on $S^2$. The properties of $\mathcal{A}$ ensure that the correlators obtained this way satisfy modular invariance in the extended sense of a surface with punctures, that includes crossing symmetry. This construction of physical correlators, when the bulk geometry is a handlebody, can be mapped to that of [52–54] by pulling the network to the boundary.

## 4.4 Examples

We conclude this section by presenting some concrete examples of algebra objects in $U(1)_k$ and $SU(2)_k$, and by algebraically checking the factorization of the $T^2 \times I$ partition function after condensation. We exploit a formula given in [57] for the topological partition function $Z_{ab}^{\mathcal{A}}$ with insertion of a dual network of the algebra object $\mathcal{A}$ in the topology $T^2 \times I$, in the case that $N_{ab}^c$ and $Z_a^{\mathcal{A}}$ take values in $\{0,1\}$:[47]

$$Z_{ab}^{\mathcal{A}} = \sum_{d,e,f \in \mathcal{A}} m_{de}^f \Delta_f^{de} \sum_{g \in \mathcal{C}} \frac{\theta_g}{\theta_b} [F_{\hat{a}}^{deb}]_{gf} [G_{\hat{a}}^{deb}]_{fg} . \tag{4.44}$$

This quantity is the transition amplitude from the state $|a\rangle$ in $\mathcal{H}(T^2)$ to the state $|b\rangle$, and the physical partition function of the gauged theory is then

$$Z_{\mathcal{T}}[T^2 \times I] = \sum_{(\mu,\bar{\nu}),(\rho,\bar{\sigma}) \in \mathcal{C}} Z_{(\mu,\bar{\nu})(\rho,\bar{\sigma})}^{\mathcal{A}} \chi_\mu(\tau_1) \chi_{\bar{\nu}}(\bar{\tau}_1) \chi_\rho(\tau_2) \chi_{\bar{\sigma}}(\bar{\tau}_2) , \tag{4.45}$$

---

[47]With the notation $a, b, c, \dots \in \mathcal{A}$ we indicate all simple anyons $L_a$ such that $Z_a^{\mathcal{A}} > 0$.

where $\tau_1$, $\tau_2$ are the modular parameters of the two torus boundaries. Here we have specialized the formula of [57] to the case of a commuting algebra $\mathcal{A}$. We also introduced the inverse matrix $G$, as in (4.6), and the co-morphism $\Delta$, which can be expressed as

$$[G_d^{abc}]_{ef} = \frac{R_f^{bc} R_d^{af}}{R_e^{ab} R_d^{ec}} [F_d^{cba}]_{ef} , \qquad \Delta_c^{ab} = \frac{1}{\dim \mathcal{A}} \frac{m_{\breve{a}c}^b [F_c^{a\breve{a}c}]_{b0}}{m_{\breve{a}a}^0 [F_a^{a\breve{a}a}]_{00}} . \qquad (4.46)$$

Since here we consider the non-chiral case that the total category is $\mathcal{C} = \mathcal{C}_L \times \overline{\mathcal{C}}_L$, using the double-index notation we have[48]

$$R_{(\rho,\bar{\rho})}^{(\mu,\bar{\mu})(\nu,\bar{\nu})} = R_\rho^{\mu\nu} (R_{\bar{\rho}}^{\bar{\mu}\bar{\nu}})^* , \qquad \left[F_{(\sigma,\bar{\sigma})}^{(\mu,\bar{\mu})(\nu,\bar{\nu})(\rho,\bar{\rho})}\right]_{(\lambda,\bar{\lambda})(\kappa,\bar{\kappa})} = [F_\sigma^{\mu\nu\rho}]_{\lambda\kappa} [F_{\bar{\sigma}}^{\bar{\mu}\bar{\nu}\bar{\rho}}]_{\bar{\lambda}\bar{\kappa}}^* , \qquad (4.48)$$

and so on. Factorization is the statement that

$$Z_{ab}^{\mathcal{A}} = Z_a^{\mathcal{A}} Z_b^{\mathcal{A}} . \qquad (4.49)$$

This immediately follows from Figure 8, *i.e.*, it is a special case of the Gauss law action.

**The case of $U(1)_k$.** We consider $k \in 2\mathbb{N}$. There are $k$ simple lines labelled by $\mu \in \mathbb{Z}_k$, forming the Abelian group $\mathbb{Z}_k$ under fusion, $L_\mu \otimes L_\nu = L_{\mu+\nu}$, thus $N_{\mu\nu}^\rho = \delta_{\mu+\nu=\rho \bmod k}$. The conjugate line is $\breve{\mu} = k - \mu$. The theory is Abelian and the quantum dimensions of all simple lines are $d_\mu = 1$. The topological spins, $R$-matrices and $F$-matrices are

$$\theta_\mu = e^{i\pi \frac{\mu^2}{k}} , \qquad R_\rho^{\mu\nu} = (-1)^{\mu\lfloor \frac{\nu}{k} \rfloor + \nu\lfloor \frac{\mu}{k} \rfloor} e^{i\pi \frac{\mu\nu}{k}} , \qquad [F_\sigma^{\mu\nu\rho}]_{\lambda\kappa} = (-1)^{\mu(\lfloor \frac{\nu+\rho}{k} \rfloor - \lfloor \frac{\nu}{k} \rfloor - \lfloor \frac{\rho}{k} \rfloor)} , \qquad (4.50)$$

where $\lfloor x \rfloor \in \mathbb{Z}$ is the floor function. Recall that $R_\rho^{\mu\nu}$ is defined only when $N_{\mu\nu}^\rho > 0$, and similarly $[F_\sigma^{\mu\nu\rho}]_{\lambda\kappa}$ is defined only when $N_{\nu\rho}^\lambda N_{\mu\lambda}^\sigma N_{\mu\nu}^\kappa N_{\kappa\rho}^\sigma > 0$. The total quantum dimension is $\mathcal{D} = \sqrt{k}$ and the chiral central charge is $c_- = 1$. The $S$-matrix is $S_{ab} = e^{-2\pi i \frac{ab}{k}}$. The two Frobenius-Schur indicators are $\varkappa_0 = 1$, $\varkappa_{k/2} = (-1)^{k/2}$.

The modular invariant associated to the integer $\omega$ is described by the Lagrangian condensable anyon

$$\mathcal{A} = \bigoplus_{\mu \in \mathbb{Z}_k} L_\mu \otimes L_{\omega\mu} , \qquad m_{ab}^c \equiv m_{(\mu,\omega\mu)(\nu,\omega\nu)}^{(\rho,\omega\rho)} = (-1)^{\mu\lfloor \frac{\nu}{k} \rfloor + \omega\mu\lfloor \frac{\omega\nu}{k} \rfloor} , \qquad (4.51)$$

where $m_{ab}^c$ is defined only when $N_{\mu\nu}^\rho > 0$, and the unit morphism is $\eta = 1$. In particular $Z_{(\mu,\bar{\mu})}^{\mathcal{A}} = \delta_{\bar{\mu}=\omega\mu \bmod k}$. One can check that the factorization property (4.49) is indeed satisfied.

**The case of $SU(2)_k$.** Next, we consider the WZW model $SU(2)_k$. To this end, we recall some basic facts about that the $SU(2)_k$ TQFT. There are $k+1$ simple lines labeled by the $SU(2)$ Dynkin label $\mu = 0, 1, \dots, k$. The fusion algebra is

$$L_\mu \otimes L_\nu = \sum_{\substack{\rho=|\mu-\nu| \\ \rho=\mu-\nu \bmod 2}}^{\min(\mu+\nu, 2k-\mu-\nu)} L_\rho , \qquad (4.52)$$

---

[48]The matrices of $\overline{\mathcal{C}}_L$ are obtained from those of $\mathcal{C}_L$ with a parity transformation. This leads to

$$\left[\overline{R}_c^{ab}\right]_{\mu\nu} = [R_c^{ab}]_{\nu\mu}^{-1} , \qquad \left[\overline{F}_d^{abc}\right]_{(e;\mu\nu)(f;\rho\sigma)} = [G_d^{abc}]_{(f;\rho\sigma)(e;\mu\nu)} . \qquad (4.47)$$

Exploiting the fact that $R_c^{ab}$, $F_d^{abc}$ and $G_d^{abc}$ are unitary matrices (and specializing to the case that the fusion coefficients $N \in \{0,1\}$), one obtains the formulas in the main text.

from which the fusion coefficients $N_{\mu\nu}^{\rho}$ follow. Each line is self-conjugate: $\check{\mu} = \mu$. The quantum dimensions, topological spins, $R$-matrices and $F$-matrices are

$$d_\mu = [\mu + 1], \qquad \theta_\mu = e^{2\pi i \Delta_\mu}, \qquad \Delta_\mu = \frac{\mu(\mu+2)}{4(k+2)},$$

$$R_\rho^{\mu\nu} = (-1)^{\frac{\mu+\nu-\rho}{2}} e^{i\pi(\Delta_\rho - \Delta_\mu - \Delta_\nu)} N_{\mu\nu}^\rho,$$

$$[F_\sigma^{\mu\nu\rho}]_{\lambda\kappa} = \begin{Bmatrix} \rho/2 & \nu/2 & \lambda/2 \\ \mu/2 & \sigma/2 & \kappa/2 \end{Bmatrix} N_{\mu\lambda}^\sigma N_{\nu\rho}^\lambda N_{\mu\nu}^\kappa N_{\kappa\rho}^\sigma,$$
(4.53)

where we introduced the $q$-number and $q$-factorial:

$$[n] = \frac{\sin\left(\frac{\pi n}{k+2}\right)}{\sin\left(\frac{\pi}{k+2}\right)}, \qquad [n]! = \prod_{m=1}^{n}[m], \qquad [0]! = [1]! = 1.$$
(4.54)

Then the curly bracket symbol is defined by

$$\begin{Bmatrix} a & b & e \\ c & d & f \end{Bmatrix} = (-1)^{a+b-c-d-2e} \sqrt{[2e+1][2f+1]} \, \Delta(a,b,e)\,\Delta(a,d,f)\,\Delta(c,b,f)\,\Delta(c,d,e)$$

$$\times \sum_{z=z_{\min}}^{z_{\max}} (-1)^z [z+1]! \Big([z{-}a{-}b{-}e]!\,[z{-}a{-}d{-}f]!\,[z{-}c{-}b{-}f]!\,[z{-}c{-}d{-}e]!$$

$$[a{+}b{+}c{+}d{-}z]!\,[a{+}c{+}e{+}f{-}z]!\,[b{+}d{+}e{+}f{-}z]!\Big)^{-1},$$
(4.55)

where the integer $z$ is summed in the interval

$$z_{\max} = \min(a+b+c+d,\, a+c+e+f,\, b+d+e+f),$$
$$z_{\min} = \max(a+b+e,\, a+d+f,\, c+b+f,\, c+d+e).$$
(4.56)

We used the function

$$\Delta(a,b,c) = \sqrt{\frac{[-a+b+c]!\,[a-b+c]!\,[a+b-c]!}{[a+b+c+1]!}}.$$
(4.57)

The total quantum dimension and the chiral central charge are

$$\mathcal{D} = \sqrt{\frac{k+2}{2}} \frac{1}{\sin\left(\frac{\pi}{k+2}\right)}, \qquad c_- = \frac{3k}{k+2}.$$
(4.58)

The $S$-matrix is

$$S_{\mu\nu} = \frac{1}{\mathcal{D}}\left[(\mu+1)(\nu+1)\right].$$
(4.59)

It is real, symmetric, and it squares to the identity. The FS indicators are $\varkappa_\mu = (-1)^\mu$.

The modular invariants of $SU(2)_k$ follow an ADE classification [103]. The simplest one, that exists for every $k \geq 1$, is the diagonal $A_{k+1}$ modular invariant:

$$Z_{(\mu,\bar{\mu})}^{\mathcal{A}} = \delta_{\mu\bar{\mu}}, \qquad m_{(\mu,\mu)(\nu,\nu)}^{(\rho,\rho)} = N_{\mu\nu}^\rho,$$
(4.60)

with unit morphism $\eta = 1$. One can explicitly verify that the factorization (4.49) holds.

The other permutation modular invariant (*i.e.*, in which the left and right modes are paired through a permutation) exists for $k = 2 \bmod 4$ with $k \geq 6$, and is called the $D_{\frac{k}{2}+2}$ invariant. The permutation $\pi$ acts as

$$\pi(\mu) = \begin{cases} \mu & \text{for } \mu \text{ even}, \\ k-\mu & \text{for } \mu \text{ odd}, \end{cases}$$
(4.61)

and is an isomorphism of the $SU(2)_k$ TQFT. The algebra object is given by

$$Z_{(\mu,\bar{\mu})}^{\mathcal{A}} = \delta_{\mu_\pi\bar{\mu}}\,, \qquad m_{(\mu,\mu_\pi)(\nu,\nu_\pi)}^{(\rho,\rho_\pi)} = N_{\mu\nu}^\rho \times \begin{cases} 1 & \text{for } m_{\text{even,even}}^{\text{even}}\,, \\ i^{\rho+\mu+1} & \text{for } m_{\text{odd,odd}}^{\text{even}}\,, \\ i^\nu & \text{for } m_{\text{odd,even}}^{\text{odd}}\,, \\ i^{\mu+\nu-\rho} & \text{for } m_{\text{even,odd}}^{\text{odd}}\,, \end{cases} \tag{4.62}$$

where we used the notation $\mu_\pi \equiv \pi(\mu)$, and with unit morphism $\eta = 1$.

In general, a permutation invariant is defined by an automorphism $\pi$ such that

$$R_\rho^{\mu\nu} = R_{\rho_\pi}^{\mu_\pi\nu_\pi}\frac{\Pi_{\mu\nu}^\rho}{\Pi_{\nu\mu}^\rho}\,, \qquad [F_\sigma^{\mu\nu\rho}]_{\lambda\kappa} = [F_{\sigma_\pi}^{\mu_\pi\nu_\pi\rho_\pi}]_{\lambda_\pi\kappa_\pi}\frac{\Pi_{\nu\rho}^\kappa\Pi_{\mu\kappa}^\sigma}{\Pi_{\mu\nu}^\lambda\Pi_{\lambda\rho}^\sigma}\,, \tag{4.63}$$

where $\Pi_{\mu\nu}^\rho$ is a gauge transformation. Using the fact that $F$ and $R$ are unitary, it is simple to show that the choice

$$m_{(\mu,\bar{\mu})(\nu,\bar{\nu})}^{(\rho,\bar{\rho})} = \delta_{\mu\bar{\mu}_\pi}\,\delta_{\nu\bar{\nu}_\pi}\,\delta^{\rho\bar{\rho}_\pi}\left(\Pi_{\mu\nu}^\rho\right)^{-1}N_{\mu\nu}^\rho\,, \tag{4.64}$$

satisfies both associativity and commutativity. We can then perform a gauge transformation on the right movers by $\Pi^{-1}$. This sets $\tilde{m}_{(\mu,\bar{\mu})(\nu,\bar{\nu})}^{(\rho,\bar{\rho})} = \delta_{\mu\bar{\mu}_\pi}\,\delta_{\nu\bar{\nu}_\pi}\,\delta^{\rho\bar{\rho}_\pi}N_{\mu\nu}^\rho$, and the computation of the partition function reduces to the one of the diagonal invariant, which exhibits factorization.

The case of non-permutation invariants, such as the $D_{\frac{k}{2}+2}$ invariant for $k = 0 \bmod 4$, is a bit more complicated. Writing the theory as $SU(2)_{4l}$, the D-type invariant is described by the Lagrangian anyon

$$\mathcal{A} = \left[\bigoplus_{\substack{\lambda=0 \\ \lambda \text{ even}}}^{k/2-2}\left(L_\lambda + L_{k-\lambda}\right)\otimes\left(L_\lambda + L_{k-\lambda}\right)\right] + 2L_{k/2}\otimes L_{k/2}\,. \tag{4.65}$$

Note in particular that $Z_{\frac{k}{2},\frac{k}{2}}^{\mathcal{A}} = 2$. Let us give some details on the simplest case, $SU(2)_4$:

$$\mathcal{A} = L_0 \otimes L_0 + L_0 \otimes L_4 + L_4 \otimes L_0 + L_4 \otimes L_4 + 2L_2 \otimes L_2\,. \tag{4.66}$$

Denoting $e = (0,0)$, $q = (0,4)$, $\tilde{q} = (4,0)$, $b = (4,4)$, and $n = (2,2)$ with $\alpha = 1,2$, the non-vanishing components of the algebra morphism are:[49]

$$\begin{aligned} & m_{a\alpha,e}^{a\alpha} = m_{a\alpha,a\alpha}^e = m_{e,a\alpha}^{a\alpha} = 1\,, \\ & m_{q,\tilde{q}}^b = m_{q,b}^{\tilde{q}} = m_{\tilde{q},q}^b = m_{\tilde{q},b}^q = m_{b,q}^{\tilde{q}} = m_{b,\tilde{q}}^q = 1\,, \\ & m_{q,n2}^{n1} = -m_{q,n1}^{n2} = m_{\tilde{q},n1}^{n2} = -m_{\tilde{q},n2}^{n1} = m_{n1,q}^{n2} = -m_{n2,q}^{n1} = m_{n2,\tilde{q}}^{n1} = -m_{n1,\tilde{q}}^{n2} = i\,, \\ & m_{n1,n2}^q = -m_{n2,n1}^q = m_{n2,n1}^{\tilde{q}} = -m_{n1,n2}^{\tilde{q}} = -i\,, \\ & m_{b,n1}^{n1} = m_{b,n2}^{n2} = m_{n1,b}^{n1} = m_{n2,b}^{n2} = m_{n1,n1}^b = m_{n2,n2}^b = -1\,, \\ & m_{n1,n1}^{n1} = -m_{n1,n2}^{n2} = -m_{n2,n1}^{n2} = -m_{n2,n2}^{n1} = 1\,. \end{aligned} \tag{4.67}$$

With this choice and the version of formula (4.44) valid when $Z_a^{\mathcal{A}}$ can be bigger than 1, one can check that factorization holds.

---

[49]There is a continuous family of solutions that depends on the chosen basis in the space of projectors $\mathcal{A} \to n$. This is not a physical parameter, and we set it to a nice value to simplify the expressions.

# 5 Discussion

In this paper we have shown how the known relation [52,57,59] between modular invariants in 2d RCFT, Lagrangian commutative special Frobenius algebras, and topological boundary conditions in 3d TQFTs, can be recast in the following form. A three-dimensional theory $\mathcal{T}$, constructed by starting with a Chern-Simons theory and then gauging a maximal one-form symmetry (or, in the non-Abelian case, performing anyon condensation of a maximal non-invertible one-form symmetry), behaves non-perturbatively as a theory of gravity and indeed is holographically dual, in the standard sense, to a well-defined (rational) conformal field theory. The bulk theory $\mathcal{T}$ has a number of interesting properties. First, it is background independent: the full non-perturbative partition function is computed by choosing any three-dimensional manifold with the correct boundary conditions, with no need to sum over geometries (in a sense, the theory automatically sums over all needed geometries). A similar behavior has been observed in examples of full-fledged string theories in AdS$_3$ [45, 46]. Second, $\mathcal{T}$ is a unitary theory of gravity: its partition functions with fixed boundary conditions are equal to the partition functions of a unitary 2d boundary CFT. In particular, the partition functions with disconnected boundaries factorize. Third, $\mathcal{T}$ does not have any bulk observable, rather, it is completely blind to the bulk geometry, which follows from the fact that it is trivial in the bulk. This is what one would expect from a holographic theory, once the full non-perturbative path integral has been done. Fourth, the bulk theory does not have any global symmetry: bulk symmetries are precisely removed by gauging the one-form symmetry.

Our initial motivation was to establish a link between the presence of (possibly non-invertible) global symmetries in the bulk, and the fact that the dual boundary system is an ensemble of theories (as signalled by the failure of factorization of the partition functions). At least in the class of models we have considered here — 3d bulk theories constructed out of Chern-Simons or other topological field theories — it is easy to convince oneself that bulk (possibly non-invertible) global symmetries imply averaging. Symmetries imply that there are local or non-local topological operators [33], therefore the bulk theory has non-trivial Hilbert spaces, hence there is a dependence on 3d topologies and factorization fails. It would be interesting to extend this argument to more general theories of gravity, in order to provide yet another argument that quantum gravities cannot have global symmetries.

A given Chern-Simons theory, in general, admits multiple maximal gaugings — described by multiple Lagrangian condensable anyons $\mathcal{A}$. Each gauging produces the dual to a precise 2d CFT, while the partition functions of the original CS theory, summed over 3d geometries in order to ensure modular invariance, display ensemble averaging. Therefore, it is tempting to interpret the gauging of $\mathcal{A}$ as a concrete realization of the projectors to the so-called $\alpha$-states of the bulk system (using the language of [49]).

In this paper we have only considered very simple topological theories, dual to RCFTs. It would be fascinating if similar ideas could be extended to non-compact Chern-Simons theories (or other non-semisimple TQFTs), possibly related to three-dimensional Einstein gravity. We considered in Section 3.3 the straighforward example of $\mathbb{R}$ CS theory, but it is clear that such an example is too simplistic to be representative of the general case.

Let us conclude with a few remarks and comments.

**Semiclassical limits.**   One might want to compare the results emerging from the formalism used here to some sort of semiclassical gravity. To do this we need to discuss possible semiclassical limits. In a semiclassical limit the central charge $c = c_L + c_R$ should go to infinity. In WZW models for the current algebra $\widehat{\mathfrak{g}}_k$, the central charge is $c_L = k \dim(\mathfrak{g})/(k + \check{c}_{\mathfrak{g}})$, where $\check{c}_{\mathfrak{g}}$ is the dual Coxeter number. One possibility is to take $\dim(\mathfrak{g})$ large, which is somewhat akin to the standard large $N$ limit. Another possibility is to take $k \to -\check{c}_{\mathfrak{g}}$. At least for $\widehat{\mathfrak{su}}(2)_k$, it

might be possible to interpret this limit through analytic continuation to $\widehat{\mathfrak{sl}}(2,\mathbb{R})_k$, for which the central charge formula is $c_L = k\dim(\mathfrak{g})/(k - \check{c}_\mathfrak{g})$ (see [104] where ideas along these lines have been applied to 3d de Sitter gravity).

Let us also notice that the dimensions of chiral vertex operators are

$$h_\lambda = \frac{\mathcal{Q}_\lambda}{2(k + \check{c}_\mathfrak{g})}\,, \tag{5.1}$$

where $\mathcal{Q}_\lambda$ the quadratic Casimir of the representation $\lambda$. This implies that, in the second type of limit, a gap of order $\Delta_{\text{gap}} \sim c$ opens up between the identity module and the first integral representation of chiral algebra. This behavior resembles a semiclassical gravity.

**Dissecting Poincaré sums.** Our construction offers a suggestive way to think of Poincaré sums of RCFT characters (*i.e.*, sums over their modular images). Consider the Poincaré sum

$$Z_{\lambda\bar{\mu}}^{\text{avg}}(\tau, \bar{\tau}) = \sum_{\gamma \in \Gamma} \chi_\lambda(\gamma \cdot \tau)\overline{\chi_{\bar{\mu}}(\gamma \cdot \tau)}\,, \tag{5.2}$$

for a "seed" character $\chi_\lambda \bar{\chi}_{\bar{\mu}}$. Here $\Gamma$ is the quotient of the modular group that acts non-trivially on the seed character. For RCFTs this is a finite group, so the sum is finite. Such sums were considered, *e.g.*, in [18, 21] and could be thought of as an avatar of sums over gravitational geometries. Since $Z_{\lambda\bar{\mu}}^{\text{avg}}$ is a modular invariant function, constructed as a linear combination of character bilinears $\chi_\rho \bar{\chi}_{\bar{\sigma}}$, it can be expanded in a complete basis of modular invariant functions

$$Z^{\mathcal{A}}(\tau, \bar{\tau}) = \sum_{\rho, \bar{\sigma}} Z_{\rho\bar{\sigma}}^{\mathcal{A}}\, \chi_\rho(\tau)\overline{\chi_{\bar{\sigma}}(\tau)}\,, \tag{5.3}$$

labelled by the index $\mathcal{A}$ and forming a finite-dimensional vector space. The expansion reads

$$Z_{\lambda\bar{\mu}}^{\text{avg}}(\tau, \bar{\tau}) = \sum_{\mathcal{A}} p_{\lambda\bar{\mu}}^{\mathcal{A}}\, Z^{\mathcal{A}}(\tau, \bar{\tau})\,, \tag{5.4}$$

in terms of coefficients $p_{\lambda\bar{\mu}}^{\mathcal{A}}$. In general the basis elements $Z^{\mathcal{A}}$ cannot be all chosen to correspond to physical modular invariants of the corresponding chiral algebra: some of them might involve coefficients $Z_{\rho\bar{\sigma}}^{\mathcal{A}}$ that are not non-negative integers. However [18,21] found by inspection that, in some cases, the $Z^{\mathcal{A}}$'s involved are all physical. We will assume that we are in such a favorable situation.

The expansion (5.4) looks like an ensemble average over theories, in a basis of Coleman's $\alpha$-states [49]. In our formalism, we can give each of them a bulk interpretation as the (unique) state resulting from the condensation of the bulk Lagrangian anyon $\mathcal{A}$. Thus, our prescription could be viewed as defining a projection to a fixed $\alpha$-state.

We can be more explicit and compute the "probabilities" $p_{\lambda\bar{\mu}}^{\mathcal{A}}$ by inverting (5.4). One first defines the positive-definite scalar product

$$M_{\mathcal{A}\mathcal{A}'} = \langle \mathcal{A}|\mathcal{A}'\rangle = \sum_{\rho, \bar{\sigma}} Z_{\rho\bar{\sigma}}^{\mathcal{A}}\, Z_{\rho\bar{\sigma}}^{\mathcal{A}'}\,. \tag{5.5}$$

Then, using modular invariance of the elements $Z^{\mathcal{A}}$, one finds

$$p_{\lambda\bar{\mu}}^{\mathcal{A}} = |\Gamma| \sum_{\mathcal{A}'} [M^{-1}]_{\mathcal{A}\mathcal{A}'}\, Z_{\lambda\bar{\mu}}^{\mathcal{A}'}\,. \tag{5.6}$$

Unfortunately, the "probabilities" $p^{\mathcal{A}}$ are not guaranteed to be non-negative. This is an important indicator for the possibility of interpreting the sum as a proper ensemble average.

The $\alpha$-states associated to two distinct Lagrangian anyons $\mathcal{A}$, $\mathcal{A}'$ fail, in general, to be orthogonal using the product (5.5), because both anyons contain the vacuum line. Since different $\alpha$-states should be orthogonal as they represent different superselection sectors, this seems to hint that a different scalar product should be used.

The procedure we outlined allows us to "dissect" Poincaré sums with one boundary in terms of bulk data. The case with multiple boundaries is in principle also tractable. One could start from a sum over higher-genus handlebodies and take a pinching limit in which the complex structure $\Omega$ becomes block diagonal, as in [105]. The computation of overlaps in the non-Abelian case is tedious, although in principle doable using surgery techniques.

**Non-compact theories.** It would be very interesting to understand whether the formulation of modular invariants in terms of Lagrangian condensable anyons can be extended to the case of Chern-Simons theories with non-compact gauge group, or to non-semisimple TQFTs. A natural objective would be towards Einstein gravity in $AdS_3$ in its Chern-Simons formulation.

# Acknowledgements

We are grateful to Andreas Blommaert, Luca Iliesiu, Ying-Hsuan Lin, Pavel Putrov, and Rajath K. Radhakrishnan for useful discussions and correspondence. We also thank the organizers and participants of the GGI workshop "Topological properties of gauge theories", where part of these results were announced. The authors are partially supported by the INFN "Iniziativa Specifica ST&FI". F.B. and C.C. are supported by the ERC-COG grant NP-QFT No. 864583 "Non-perturbative dynamics of quantum fields: from new deconfined phases of matter to quantum black holes", and by the MIUR-PRIN contract 2015 MP2CX4. F.B. is also supported by the MIUR-SIR grant RBSI1471GJ. L.D. also acknowledges support by the program "Rita Levi Montalcini" for young researchers.

# A Higher-genus partition functions

Here we review the computation of the non-holomorphic function $\Phi$ appearing in the higher-genus partition functions (2.18) and (2.27). From the point of view of the 2d CFT, $\Phi$ can be written as (see Section 3 of [74] and references therein):

$$\Phi = \left| \det{}'_\Gamma \bar{\partial}_0 \right|, \tag{A.1}$$

where $\bar{\partial}_0$ is the Dolbeault operator mapping functions to $(0,1)$-forms, and the symbol $\det'$ denotes that the zero-modes are removed. The subscript $\Gamma$ reminds us that this regularized determinant depends on a choice of Lagrangian (primitive) sublattice $\Gamma \subset H_1(\Sigma, \mathbb{Z})$, *i.e.*, on a choice of contractible cycles in the corresponding three-dimensional handlebody. On the other hand, using two-dimensional holomorphic factorization [106–108]

$$\det{}' \Delta_0 = \left( \det \mathbb{Im}\,\Omega \right) \cdot \left| \det{}'_\Gamma \bar{\partial}_0 \right|^2, \tag{A.2}$$

where $\Delta_0$ is the scalar Laplacian on $\Sigma$ and the choice of $\Omega$ is related to $\Gamma$ (see [16], sections 3.1 and 4.2), one gets the alternative expression

$$\Phi = \left( \frac{\det{}' \Delta_0}{\det \mathbb{Im}\,\Omega} \right)^{1/2}. \tag{A.3}$$

Notice that both $\left| \det'_\Gamma \bar{\partial}_0 \right|$ and $(\det' \Delta_0)^{1/2}$ suffer from the conformal anomaly for $g \neq 1$. The Laplacian determinant $\det' \Delta_0$ is modular invariant.

On the other hand, following the discussion in Section 4 of [16], we can make contact with the partition function of 3d CS theory by using Zograf's three-dimensional holomorphic factorization formula [109, 110]:

$$\Phi = \left(\frac{\det' \Delta_0}{\det \mathbb{Im}\, \Omega}\right)^{1/2} = e^{-S_\mathrm{L}/24\pi} \left|\prod_{\gamma \in \mathcal{P}} \prod_{n=1}^{\infty} (1 - q_\gamma^n)\right|^2 . \tag{A.4}$$

Here $G$ is a Schottky group such that $\mathbb{H}^3/G$ is a handlebody whose boundary is $\Sigma$ and whose contractible cycles are indicated by $\Gamma$, while $\mathcal{P}$ is the set of primitive subgroups of $G$ and $\gamma$ are their generators. For each generator $\gamma$, we write $\gamma \in G \subset PSL(2, \mathbb{C})$ as a $2 \times 2$ matrix, then use $2 \cos \pi \tau_\gamma = \mathrm{Tr}\, \gamma$ to define $\tau_\gamma$ (the real part of $\tau_\gamma$ is only defined mod 1, while we choose the sign of $\tau_\gamma$ in such a way that $\mathbb{Im}\, \tau_\gamma > 0$), and finally define $q_\gamma = e^{2\pi i \tau_\gamma}$. The function $S_\mathrm{L}$ is a Liouville action, defined in [111], which reproduces the conformal anomaly and obstructs holomorphic factorization [112].

It was proven in [113] that $S_\mathrm{L}$ is equal to the holographically regularized volume of the handlebody $\mathbb{H}^3/G$:

$$S_\mathrm{L} = -4\,\mathrm{Vol}\!\left(\mathbb{H}^3/G\right) + \text{counterterms} . \tag{A.5}$$

For instance, for $g = 1$ one finds $\mathrm{Vol}(\mathbb{H}^3/G) = -\pi^2 \tau_2$ while $\mathcal{P}$ has a unique element, so that $\Phi = e^{-\pi \tau_2/6} \left|\prod_n (1 - q^n)\right|^2 = \left|\eta(\tau)\right|^2$.[50] Finally, one can use heat kernel methods to express the ratio of determinants in (2.26) [16, 114], arising from the perturbative partition function of $U(1)_k \times U(1)_{-k}$ Chern-Simons theory on a handlebody with conformal boundary conditions, in terms of the quantities discussed above:

$$Z_\mathrm{CS}^\mathrm{pert} \equiv \frac{(\det' \Delta_0)^{3/2}}{(\det' \Delta_1)^{1/2}} = \exp\!\left[-\frac{\mathrm{Vol}(\mathbb{H}^3/G)}{6\pi}\right] \left|\prod_{\gamma \in \mathcal{P}} \prod_{n=1}^{\infty} \frac{1}{1 - q_\gamma^n}\right|^2 = \frac{1}{\Phi} . \tag{A.6}$$

This partition function does not include the sum over disconnected flat connections.

# B  Boundary symmetry and contact term

Consider the Abelian Chern-Simons theory with action

$$S_\mathrm{CS} = i\frac{k}{4\pi} \int_{M_3} a\, da . \tag{B.1}$$

We study it on a manifold $M_3$ with a boundary $\partial M_3 = M_2$. We pick a complex structure on $M_2$ and we write the boundary term

$$S_\partial = \frac{k}{4\pi} \int_{M_2} d^2x\, \sqrt{g}\, g^{z\bar{z}}\, a_z a_{\bar{z}} , \tag{B.2}$$

using a Hermitean metric. The Hermitean metric and the volume form which are compatible with the complex structure are related by $\epsilon^{z\bar{z}} = i g^{z\bar{z}}$. Using this, we get the following variation of the total action:

$$\delta\!\left(S_\mathrm{CS} + S_\partial\right) = i\frac{k}{2\pi} \int_{M_3} \delta a\, da + \frac{k}{2\pi} \int_{M_2} d^2x\, \sqrt{g}\, g^{z\bar{z}}\, a_z\, \delta a_{\bar{z}} . \tag{B.3}$$

---

[50] Note that in this case of $g = 1$, the prefactor $e^{-S_\mathrm{L}/24\pi}$ in eqn. (A.4) gives a factorized contribution $e^{-\pi \tau_2/6} = q^{\frac{1}{24}} \bar{q}^{\frac{1}{24}}$. We can also compare with eqn. (2.3) of [79], taking into account that they include a factorized contribution in their definition of $\widetilde{F}$, see their footnote 3.

Therefore the equation of motion is $da = 0$ and the boundary condition that makes the action stationary is $a_{\bar{z}}\big|_{M_2} = A_{\bar{z}}$, where the uppercase letter denotes a fixed $c$-number connection.

Evaluating the EOM on the boundary we get

$$0 = da\big|_{M_2} = (\partial_z a_{\bar{z}} - \partial_{\bar{z}} a_z)\big|_{M_2} dz \wedge d\bar{z} = \left(\partial_z A_{\bar{z}} - \partial_{\bar{z}} a_z\big|_{M_2}\right) dz \wedge d\bar{z} . \tag{B.4}$$

We see that $a_z\big|_{M_2}$ is a holomorphic current when the boundary value $A_{\bar{z}}$ is set to zero, and otherwise it satisfies the "anomalous" conservation equation

$$\partial_{\bar{z}} a_z\big|_{M_2} = \partial_z A_{\bar{z}} . \tag{B.5}$$

Defining the partition function

$$\begin{aligned} Z[A_{\bar{z}}] &= \int_{a_{\bar{z}}|_{M_2}=A_{\bar{z}}} \mathcal{D}a \; \exp\left[-S_{\text{CS}} - S_\partial\right] \\ &= \int_{a_{\bar{z}}|_{M_2}=A_{\bar{z}}} \mathcal{D}a \; \exp\left[-i\frac{k}{4\pi} \int_{M_3} a\,da - \frac{k}{4\pi} \int_{M_2} d^2x \; \sqrt{g}\, g^{z\bar{z}} A_{\bar{z}} a_z\right], \end{aligned} \tag{B.6}$$

a natural normalization of the holomorphic current on $M_2$ is to take

$$\langle j_z \rangle_{A_{\bar{z}}} = -\frac{g_{z\bar{z}}}{\sqrt{g}} \frac{\delta}{\delta A_{\bar{z}}} \log Z[A_{\bar{z}}] , \tag{B.7}$$

from which we get

$$j_z = \frac{k}{4\pi} a_z\big|_{M_2} . \tag{B.8}$$

There are several related unpleasant features in these formulas:

(i) The current is not invariant under gauge transformations $a \to a + d\lambda$, namely

$$j_z \to j_z + \frac{k}{4\pi} \partial_z \Lambda , \tag{B.9}$$

where here and in the following $\Lambda(z,\bar{z})$ denotes the restriction of the gauge parameter $\lambda$ to $M_2$.

(ii) The current does not satisfy the standard anomaly equation:

$$\partial_{\bar{z}} j_z = \frac{k}{4\pi} \partial_z A_{\bar{z}} \neq \frac{k}{4\pi} F_{z\bar{z}} , \tag{B.10}$$

which is a trivial consequence of the fact that the holomorphic component $A_z$ does not enter in our formulas.

(iii) The partition function does not transform in the expected way under a background gauge transformation. To compute how the partition function transforms under a gauge transformation $a \to a + d\lambda \equiv a'$, we perform a change of variables in the path integral setting $a = a' - d\lambda$. We obtain

$$\begin{aligned} Z[A_{\bar{z}}] = \int_{a'_{\bar{z}}|_{M_2}=A_{\bar{z}}+\partial_{\bar{z}}\Lambda} \mathcal{D}a' \; \exp\Big[&-i\frac{k}{4\pi} \int_{M_3} a'\,da' - \frac{k}{4\pi} \int_{M_2} d^2x \; \sqrt{g}\, g^{z\bar{z}}\big(A_{\bar{z}} + \partial_{\bar{z}}\Lambda\big)a'_z \\ &- \frac{k}{2\pi} \int_{M_2} d^2x \; \sqrt{g}\, g^{z\bar{z}} \Lambda \partial_z A_{\bar{z}} + \frac{k}{4\pi} \int_{M_2} d^2x \; \sqrt{g}\, g^{z\bar{z}} \partial_z \Lambda \partial_{\bar{z}}\Lambda\Big]. \end{aligned} \tag{B.11}$$

Collecting the *c*-numbers on the second line as a factor in front of the path integral, we recognize the remaining path integral to be precisely the partition function with source $A_{\bar{z}} + \partial_{\bar{z}}\Lambda$, so we obtain

$$Z[A_{\bar{z}} + \partial_{\bar{z}}\Lambda] = \exp\left[\frac{k}{2\pi}\int_{M_2} d^2x \sqrt{g}\, g^{z\bar{z}}\Lambda\,\partial_z A_{\bar{z}} - \frac{k}{4\pi}\int_{M_2} d^2x \sqrt{g}\, g^{z\bar{z}}\partial_z\Lambda\,\partial_{\bar{z}}\Lambda\right]Z[A_{\bar{z}}]. \quad \text{(B.12)}$$

All of these unpleasant features can be fixed simply by adding a boundary contact term:

$$S_{\text{ct}} = -\frac{k}{4\pi}\int_{M_2} d^2x \sqrt{g}\, g^{z\bar{z}} A_z A_{\bar{z}}, \quad \text{(B.13)}$$

and defining accordingly a new partition function

$$\widetilde{Z}[A_z, A_{\bar{z}}] = \exp\left[\frac{k}{4\pi}\int_{M_2} d^2x \sqrt{g}\, g^{z\bar{z}} A_z A_{\bar{z}}\right]Z[A_{\bar{z}}]. \quad \text{(B.14)}$$

With the same definition as before,

$$\langle \tilde{j}_z \rangle_{A_z, A_{\bar{z}}} = -\frac{g_{z\bar{z}}}{\sqrt{g}}\frac{\delta}{\delta A_{\bar{z}}}\log \widetilde{Z}[A_z, A_{\bar{z}}], \quad \text{(B.15)}$$

we now get

$$\tilde{j}_z = \frac{k}{4\pi}\left(a_z\big|_{M_2} - A_z\right), \quad \text{(B.16)}$$

which has the advantage of being gauge-invariant, at least if we accompany the gauge transformation of the dynamical field by a background gauge transformation with the same parameter. At the same time, with this modification the current now satisfies the standard anomaly equation:

$$\partial_{\bar{z}}\tilde{j}_z = \frac{k}{4\pi}F_{z\bar{z}}. \quad \text{(B.17)}$$

Moreover, using (B.12) we can easily check that the new partition function has a standard anomalous transformation under background gauge transformations:

$$\begin{aligned}
\widetilde{Z}[A_z + \partial_z\Lambda, A_{\bar{z}} + \partial_{\bar{z}}\Lambda] &= \exp\left[\frac{k}{4\pi}\int_{M_2} d^2x \sqrt{g}\, g^{z\bar{z}}\Lambda\, F_{z\bar{z}}\right]\widetilde{Z}[A_z, A_{\bar{z}}] \\
&= \exp\left[-i\frac{k}{4\pi}\int_{M_2}\Lambda F\right]\widetilde{Z}[A_z, A_{\bar{z}}].
\end{aligned} \quad \text{(B.18)}$$

# C  Gauging formulas with boundary

In this appendix we derive formula (2.34) for the partition function of a 3d theory in which a discrete (Abelian) 1-form symmetry is gauged, on a manifold with boundary.[51]

Let $M$ be a closed (oriented) 3-manifold. Gauging of a discrete 1-form symmetry $\mathcal{A}$, as in (2.32), leads to the partition function

$$Z^{\text{gauged}}[M] = \frac{|H^0(M)|}{|H^1(M)|}\sum_{a\in H^2(M)} Z[M;a]. \quad \text{(C.1)}$$

---

[51] We are grateful to Pavel Putrov for explaining the context of this appendix to us.

Here and in the following we make explicit the manifolds on which the Euclidean path integrals $Z$ are computed. On the other hand, all cohomology groups are with coefficients in the Abelian group $\mathcal{A}$, which we then keep implicit.

Now, let $M = M_+ \cup M_-$ such that $\partial M_+ = \partial M_- = \Sigma$ and $M_+ \cap M_- = \Sigma$. Here $\Sigma$ is a 2-dimensional closed manifold, not necessarily connected. We would like to define the partition functions $Z[M_\pm]$ on $M_\pm$ with Dirichelet boundary conditions on $\Sigma$, in such a way that $Z[M]$ factorizes into them, with a sum over the boundary conditions. We would like to use the Mayer-Vietoris sequence (see, *e.g.*, [85]):

$$\ldots \xrightarrow{\Phi} H^{n-1}(\Sigma) \xrightarrow{\delta} H^n(M) \xrightarrow{\Psi} H^n(M_+) \oplus H^n(M_-) \xrightarrow{\Phi} H^n(\Sigma) \xrightarrow{\delta} \ldots, \qquad \text{(C.2)}$$

to express $H^2(M)$ in terms of $H^2(M_+) \oplus H^2(M_-)$. The map $\Phi$ restricts cocycles in $M_+$ and in $M_-$ to $\Sigma$ and then takes the difference. Therefore the kernel of $\Phi$ inside $H^2(M_+) \oplus H^2(M_-)$ consists of a sum over 1-form bundles on $M_\pm$ with equal boundary conditions (summed over such fixed and equal boundary conditions). However this is not isomorphic to $H^2(M)$ because $\Psi$ has a kernel, equal to $\delta(H^1(\Sigma))$.

To solve the problem, we consider cohomology relative to a subspace of $\Sigma$, such that $H^1$ of $\Sigma$ becomes trivial but $H^2$ of $\Sigma$ remains the same. This can be achieved using $\Sigma \smallsetminus P$, where $P$ consists of one point in each connected component of $\Sigma$. Heuristically, the relative cohomology group $H^2(\Sigma, \Sigma \smallsetminus P)$ focuses the 2-cocycles to a point in each connected component, and this indeed captures the full cohomology. Thus, we apply the Mayer-Vietoris sequence to $(M, \Sigma \smallsetminus P) = (M_+, \Sigma \smallsetminus P) \cup (M_-, \Sigma \smallsetminus P)$.

Let us first apply the long exact sequence to the pair $(\Sigma, \Sigma \smallsetminus P)$, which involves the local cohomology groups $H^n(\Sigma, \Sigma \smallsetminus P)$:

$$
\begin{aligned}
0 &\longrightarrow \underbrace{H^0(\Sigma, \Sigma \smallsetminus P)}_{0} \longrightarrow H^0(\Sigma) \xrightarrow{\approx} H^0(\Sigma \smallsetminus P) \longrightarrow \\
&\longrightarrow \underbrace{H^1(\Sigma, \Sigma \smallsetminus P)}_{0} \longrightarrow H^1(\Sigma) \xrightarrow{\approx} H^1(\Sigma \smallsetminus P) \xrightarrow{0} \\
&\xrightarrow{0} H^2(\Sigma, \Sigma \smallsetminus P) \xrightarrow{\approx} H^2(\Sigma) \longrightarrow \underbrace{H^2(\Sigma \smallsetminus P)}_{0} \longrightarrow 0.
\end{aligned}
\qquad \text{(C.3)}
$$

Here $\approx$ means isomorphism. For a connected 2-manifold $N_2$, $x \in N_2$ a point, and $D_2$ a disk around $x$, Poincaré duality implies $H^n(N_2 \smallsetminus \{x\}) \approx H^n(N_2 \smallsetminus \mathring{D}_2) \approx H_{2-n}(N_2 \smallsetminus \mathring{D}_2, \partial D_2)$. The Mayer-Vietoris sequence then gives:

$$
\begin{aligned}
0 &\longrightarrow H^0(M, \Sigma \smallsetminus P) \xrightarrow{\approx} H^0(M_+, \Sigma \smallsetminus P) \oplus H^0(M_-, \Sigma \smallsetminus P) \longrightarrow \underbrace{H^0(\Sigma, \Sigma \smallsetminus P)}_{0} \longrightarrow \\
&\longrightarrow H^1(M, \Sigma \smallsetminus P) \xrightarrow{\approx} H^1(M_+, \Sigma \smallsetminus P) \oplus H^1(M_-, \Sigma \smallsetminus P) \longrightarrow \underbrace{H^1(\Sigma, \Sigma \smallsetminus P)}_{0} \longrightarrow \\
&\longrightarrow H^2(M, \Sigma \smallsetminus P) \longrightarrow H^2(M_+, \Sigma \smallsetminus P) \oplus H^2(M_-, \Sigma \smallsetminus P) \xrightarrow{\Phi} \underbrace{H^2(\Sigma, \Sigma \smallsetminus P)}_{H^2(\Sigma)} \longrightarrow \ldots,
\end{aligned}
$$
$$\text{(C.4)}$$

which implies

$$H^2(M, \Sigma \smallsetminus P) \approx \operatorname{Ker} \Phi \subset H^2(M_+, \Sigma \smallsetminus P) \oplus H^2(M_-, \Sigma \smallsetminus P). \qquad \text{(C.5)}$$

The inclusion $\Sigma \overset{i}{\hookrightarrow} M_+$ induces the homomorphism $H^2(M_+, \Sigma \smallsetminus P) \xrightarrow{i^*} H^2(\Sigma, \Sigma \smallsetminus P) \approx H^2(\Sigma)$ therefore

$$H^2(M_+, \Sigma \smallsetminus P) = \bigoplus_{b \in H^2(\Sigma)} H^2(M_+, \Sigma \smallsetminus P)\Big|_{\operatorname{Im} i^* = b}, \qquad \text{(C.6)}$$

where the groups on the RHS have fixed boundary conditions, and the sum is over those boundary conditions. The restriction to $\text{Ker}\,\Phi$ corresponds to imposing that the boundary conditions are the same on $M_+$ and $M_-$.

On the other hand, we would like to obtain an expression for the group $H^2(M)$, as opposed to the relative cohomology group $H^2(M, \Sigma \smallsetminus P)$. The long exact sequence for the pair $(M, \Sigma \smallsetminus P)$ is:

$$
\begin{aligned}
0 \longrightarrow H^0(M, \Sigma \smallsetminus P) \longrightarrow H^0(M) \longrightarrow & \underbrace{H^0(\Sigma \smallsetminus P)}_{H^0(\Sigma)} \longrightarrow \\
\longrightarrow H^1(M, \Sigma \smallsetminus P) \longrightarrow H^1(M) \longrightarrow & \underbrace{H^1(\Sigma \smallsetminus P)}_{H^1(\Sigma)} \overset{\gamma}{\longrightarrow} \\
\overset{\gamma}{\longrightarrow} H^2(M, \Sigma \smallsetminus P) \longrightarrow H^2(M) \longrightarrow & \underbrace{H^2(\Sigma \smallsetminus P)}_{0} \longrightarrow \dots,
\end{aligned}
\tag{C.7}
$$

and similar sequences can be written for $M_\pm$. In particular

$$
H^2(M) \approx H^2(M, \Sigma \smallsetminus P) / \text{Im}\left( H^1(\Sigma \smallsetminus P) \overset{\gamma}{\longrightarrow} H^2(M, \Sigma \smallsetminus P) \right).
\tag{C.8}
$$

Notice that $H^2(M, \Sigma \smallsetminus P)$ is given by 2-cocycles that vanish on $\Sigma \smallsetminus P$ modulo coboundaries of 1-cochains that vanish on $\Sigma \smallsetminus P$; while the former represent the whole cohomology $H^2(\Sigma)$ — see (C.3) — the latter do not represent any element of $H^1(\Sigma)$, and as a result $H^2(M, \Sigma \smallsetminus P)$ is larger than $H^2(M)$. However $Z[M; a]$ only depends on $a \in H^2(M)$, therefore if we vary $a$ by an element in the kernel of the map $H^2(M, \Sigma \smallsetminus P) \to H^2(M)$ (which is equal to $\text{Im}\,\gamma$) then $Z[M; a]$ remains invariant. This shows that if we parametrize bundles using $H^2(M, \Sigma \smallsetminus P)$ we only suffer from an overcounting problem, which can be taken care of by dividing by $|\text{Im}\,\gamma|$. From (C.7) we obtain

$$
\begin{aligned}
|\text{Im}\,\gamma| &= \frac{|H^1(\Sigma)|\,|H^1(M, \Sigma \smallsetminus P)|\,|H^0(M)|}{|H^1(M)|\,|H^0(\Sigma)|\,|H^0(M, \Sigma \smallsetminus P)|} \\
&= \frac{|H^1(\Sigma)|\,|H^0(M)|}{|H^0(\Sigma)|\,|H^1(M)|} \prod_{i=+,-} \frac{|H^1(M_i, \Sigma \smallsetminus P)|}{|H^0(M_i, \Sigma \smallsetminus P)|},
\end{aligned}
\tag{C.9}
$$

where in the second equality we used (C.4).

We conclude that the partition function (C.1) takes the form:

$$
Z^{\text{gauged}}[M] = \left| \frac{H^0(M)}{H^1(M)} \right| \frac{1}{|\text{Im}\,\gamma|} \sum_{a \in H^2(M, \Sigma \smallsetminus P)} Z[M; a]
\tag{C.10}
$$

$$
= \left| \frac{H^0(M)}{H^1(M)} \right| \frac{1}{|\text{Im}\,\gamma|} \sum_{b \in H^2(\Sigma)} \sum_{\substack{a_+ \in H^2(M_+, \Sigma \smallsetminus P) \\ i^*(a_+)=b}} \sum_{\substack{a_- \in H^2(M_-, \Sigma \smallsetminus P) \\ i^*(a_-)=b}} Z[M_+; a_+] Z[M_-; a_-].
$$

This prompts the following definition of the gauged partition function with Dirichelet boundary conditions $b \in H^2(\Sigma)$, which coincides with (2.34):

$$
Z_b^{\text{gauged}}[M_+] = \left| \frac{H^0(M_+, \Sigma \smallsetminus P)}{H^1(M_+, \Sigma \smallsetminus P)} \right| \sum_{\substack{a \in H^2(M_+, \Sigma \smallsetminus P) \\ i^*(a)=b}} Z[M_+; a],
\tag{C.11}
$$

and similarly for $M_-$. It yields the gluing formula

$$
Z^{\text{gauged}}[M] = \left| \frac{H^0(\Sigma)}{H^1(\Sigma)} \right| \sum_{b \in H^2(\Sigma)} Z_b^{\text{gauged}}[M_+] Z_b^{\text{gauged}}[M_-].
\tag{C.12}
$$

The sum over boundary conditions corresponds to gauging a 1-form symmetry on the two-dimensional closed manifold $\Sigma$, and the normalization factor agrees with (2.32).

If $M$ is orientable, we can express (C.11) and (2.34) in terms of homology, as opposed to cohomology, groups. Using Poincaré duality and with a few steps, we can prove

$$H^n(M_+, \partial M_+ \smallsetminus P) \approx H_{d-n}(M_+, P).\qquad(\text{C.13})$$

In particular with $d = 3$ we have $H^0(M_+, \Sigma \smallsetminus P) \approx H_3(M_+)$ and $H^1(M_+, \Sigma \smallsetminus P) \approx H_2(M_+)$. The group $H_1(M_+, P)$ is larger than $H_1(M_+)$ because it includes relative 1-cycles going from one connected component of the boundary to another, however these extra 1-cycles are precisely fixed by the boundary conditions. This leads to the alternative formula (2.35).

To finish, let us prove (C.13). We let $M_+$ be an orientable compact manifold of dimension $d$ with boundary, and $P \subset \partial M_+$ be a set of points consisting of one point in each connected component of $\partial M_+$. Moreover, let $D \subset \partial M_+$ be a set of closed disks $D_{d-1}$, one disk around each point of $P$. The long exact sequence for the triple $(M_+, \partial M_+ \smallsetminus P, \partial M_+ \smallsetminus \mathring{D})$ is

$$\ldots \longrightarrow H^n(M_+, \partial M_+ \smallsetminus P) \longrightarrow H^n(M_+, \partial M_+ \smallsetminus \mathring{D}) \longrightarrow H^n(\partial M_+ \smallsetminus P, \partial M_+ \smallsetminus \mathring{D}) \longrightarrow \ldots,\quad(\text{C.14})$$

Using excision and homotopy invariance, for each $i$-th connected component of $\partial M_+$ we get

$$H^n(\partial M_+ \smallsetminus P, \partial M_+ \smallsetminus \mathring{D})\Big|_i \approx H^n(\mathbb{R}^{d-1} \smallsetminus \{0\}, \mathbb{R}^{d-1} \smallsetminus \mathring{D}) \approx H^n(\mathbb{R}^{d-1} \smallsetminus \mathring{D}, \mathbb{R}^{d-1} \smallsetminus \mathring{D}) = 0,\quad(\text{C.15})$$

where, by abuse of notation, $D$ is also a $(d-1)$-dimensional disk in $\mathbb{R}^{d-1}$ around the origin. This implies $H^n(M_+, \partial M_+ \smallsetminus P) \approx H^n(M_+, \partial M_+ \smallsetminus \mathring{D})$. To the spaces on the RHS we can apply Poincaré duality, $\approx H_{d-n}(M_+, D)$, and by homotopy invariance $\approx H_{d-n}(M_+, P)$.

# D Factorization for higher genus

Here we discuss factorization in the case of two boundaries with higher genus. We explicitly perform the calculation that proves factorization on $\Sigma_2 \times I$, while for the case of $g > 2$ we explain all the relevant ingredients but do not perform the full calculation.

The defect network for gauging on $\Sigma_2 \times I$ can be reduced to a network lying on $\Sigma_2$ at a certain point in $I$. We represent that $\Sigma_2$ by gluing the following building blocks:

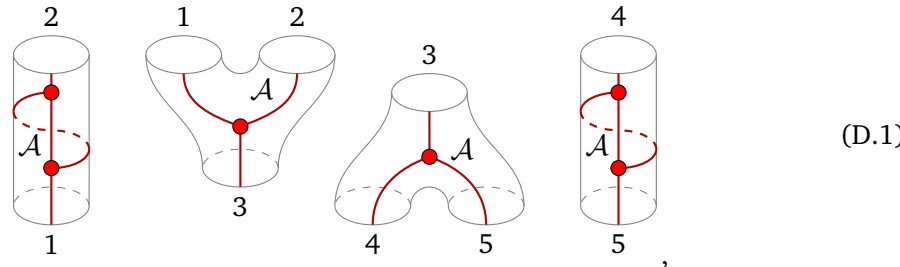

$$(\text{D.1})$$

with the understanding that we glue along matching numbers. We will henceforth omit $\mathcal{A}$ labels for ease of notation. As a convention, the lines are oriented upwards, which automatically defines for us which intersections are morphisms or co-morphisms. Such choice is carried throughout the computations.

We perform surgery along a $\Sigma_2$ slice by inserting a complete set of states on $\Sigma_2$:

$$\mathbb{1} = \sum_{\substack{a,b,c \in \mathcal{A} \\ \mu \in \text{Hom}(a \otimes \breve{a}, b),\ \nu \in \text{Hom}(c \otimes \breve{c}, b)}} c_{abc}^{\mu\nu} |a,b,c;\mu,\nu\rangle \langle a,b,c;\mu,\nu|,\qquad(\text{D.2})$$

where $c_{abc}^{\mu\nu} = \langle a, b, c; \mu, \nu | a, b, c; \mu, \nu \rangle^{-1}$. The state $|a, b, c; \mu, \nu\rangle$ is defined by the following path integral on $S\Sigma_2$ (gluings are as in the previous figure, and brown disks indicate that the surfaces are filled):

$$|a, b, c; \mu, \nu\rangle = \qquad\qquad\qquad\qquad\qquad\qquad\qquad\qquad . \tag{D.3}$$

We can also represent this more economically as follows:

$$= \qquad\qquad\qquad\qquad\qquad , \tag{D.4}$$

where crosses indicate non contractible cycles. A computation leads to[52]

$$c_{abc}^{\mu\nu} = \frac{d_a \, d_c}{\mathcal{D} \, d_b} \, . \tag{D.5}$$

On one side of the surgery we find the diagram

$$[S\Sigma_2]^+ = \qquad\qquad\qquad\qquad\qquad\qquad\qquad\qquad$$

$$= \qquad\qquad\qquad\qquad\qquad\qquad\qquad . \tag{D.6}$$

Inside the first and fourth block we recognize the projectors $P_{\mathcal{A}}(a)$ and $P_{\mathcal{A}}(c)$, which, for Lagrangian $\mathcal{A}$, we can write as in (4.37). After that, we move all morphisms into the pairs of pants. This leads to considering the following identity:

$$\frac{\mathcal{D}}{d_a} \sum_{\alpha=1}^{Z_a^{\mathcal{A}}} \qquad\qquad = \frac{\mathcal{D}}{d_a} \sum_{\alpha, \beta} \Delta_{a\alpha;\mu}^{a\alpha, b\beta} \qquad\qquad . \tag{D.7}$$

---

[52]The gist of the computation is as follows: when one glues together the two $S\Sigma_2$, the contractible disks combine (locally) into an $S^2$. Thus the glued geometry locally looks like $S^2 \times \mathbb{R}$, with the $\mathbb{R}$ direction winding around in a non-trivial way carrying the lines $a, \hat{a}, b, \hat{b}$, and $c, \hat{c}$. Since the Hilbert space on a bi-punctured $S^2$ is one dimensional if the punctures are conjugate and empty otherwise, we use this to perform surgery locally. We cut transversally to the local $\mathbb{R}$ direction, and on each side of the cut we glue a $D_3$ containing a line that connects the two conjugate punctures, for instance $a, \hat{a}$. When we do that, we need to divide by the norm of the state, which is $d/\mathcal{D}$, where $d$ is the dimension of the line. We cut in this way each of the lines $a, b, c$ (with their conjugates). Eventually, we are left with a simple configuration of lines in $S^3$.

The numbers here correspond to those in (D.1). To show this equality, one uses crossing twice to isolate a bubble containing the projectors, the relationship

$$
\left(
\begin{array}{c}
c \\
a \;\; \mu \;\; b \\
\alpha \quad \beta \\
\mathcal{A}
\end{array}
\right)
= \sum_{\gamma} \Delta_{c\gamma;\mu}^{a\alpha,b\beta}
\left(
\begin{array}{c}
c \\
\gamma \\
\mathcal{A}
\end{array}
\right)
,
\tag{D.8}
$$

and a bit of care with the orientations of lines. The same is done in the second pair of pants, with now $c$ instead of $a$ and the morphism $m$ instead of $\Delta$. We then find the following

$$
\left(
\begin{array}{c}
\times \quad \dfrac{\beta \;\; b \;\; \beta'}{\phantom{x}} \quad \times
\end{array}
\right)
= \frac{d_b}{\mathcal{D}} \delta_{\beta\beta'}
\left(
\begin{array}{c}
\times \quad\quad\quad \times
\end{array}
\right).
\tag{D.9}
$$

Putting everything together we get

$$
[S\Sigma_2]^+ = \frac{\mathcal{D} d_b}{d_a d_c} c_{abc}^{\mu\nu} \sum_{\alpha,\beta,\gamma} \Delta_{a\alpha;\mu}^{a\alpha,b\beta} m_{c\gamma,b\beta}^{c\gamma;\nu}
\left(
\begin{array}{c}
\times \quad\quad\quad \times
\end{array}
\right).
\tag{D.10}
$$

The overall factor cancels with $c_{abc}^{\mu\nu}$ and the diagram on the right is the gauging network for $S\Sigma_2$. The factors of $m$ and $\Delta$ can now be brought over to the other part $[S\Sigma_2]^-$ of the surgery. Summing over $a, b, c$ and $\mu, \nu$, and combining with the factors of $m$ and $\Delta$, one reconstructs the gauging network on $[S\Sigma_2]^-$ as well, showing factorization.

The case of higher genus is tractable in a similar way, by replacing locally the projector $P_{\mathcal{A}}$ with the expression in (4.37). On $\Sigma_g \times I$, in order to perform surgery along a $\Sigma_g$ slice, it is convenient to choose a complete set of states in the Hilbert space on $\Sigma_g$ defined by the following network of lines:

$$
\left(
\begin{array}{c}
\times \quad \times \quad\quad \times \quad \times \\
\times \quad\quad \cdots \quad\quad \times
\end{array}
\right).
\tag{D.11}
$$

This is labeled by the choice of lines on each segment, and the choice of morphisms at each junction. This basis of states is alternative to the one in Figure 5. The network in (D.11) comes from the building blocks in (D.3) for a certain choice of pair-of-pants decomposition of $\Sigma_g$. We will take as the network for the condensation of $\mathcal{A}$ on $\Sigma_g \times I$ the one that is obtained from the same pair-of-pants decomposition, but using the building blocks in (D.1) instead. After surgery along $\Sigma_g$, the goal is to obtain two factors of $S\Sigma_g$ each with the insertion of the condensation network on $S\Sigma_g$. The latter takes the same form of the network (D.11), only with the line $\mathcal{A}$ on the segments and the algebra (co)morphisms at the junctions.

To simplify the sides of the surgery that contains the condensation network, we apply to each of the loops around the non-contractible cycles in (D.11) the identity (D.7) (the segment with endpoints 1 and 2 wraps the non-contractible cycle). In this way, up to factors of quantum

dimensions and of algebra (co)morphisms, we obtain the following configuration:

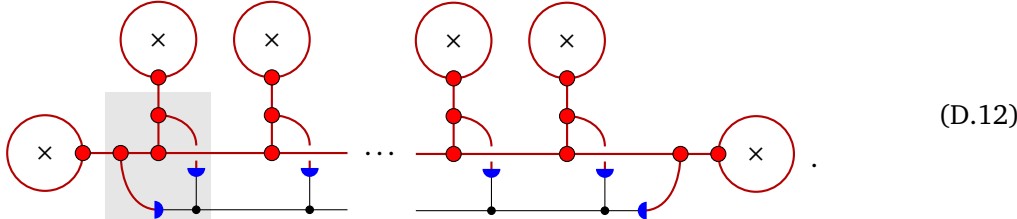

$$(D.12)$$

To simplify further, we apply the following generalization of (D.7) to the shaded region:

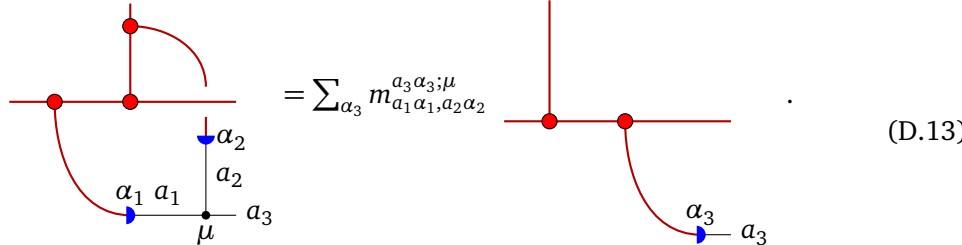

$$(D.13)$$

This simplification can be iterated, and at the end one finds precisely the gauging network on $S\Sigma_g$, times (co)morphisms factors for every junction of the initial network (D.11). These factors can be used to convert the network on the other side of the surgery to a gauging network. The leftover factors of quantum dimensions, and a factor of the state normalization (D.11) which appears when inserting the completeness relation, cancel yielding factorization.

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
