# Peer review of "Factorization and global symmetries in holography"

_SciPost Physics, doi:SciPost Phys. 14, 019 (2023)_

## Round 2 · Referee Report · Anonymous (Referee 1) · 2022-8-10

Strengths

Beautiful idea that is implemented in full technical detail.

Weaknesses

Perhaps not fully self-contained for the reader not already well-versed in generalized symmetries, homology classes, or modular tensor categories.

Report

Dear Editor,

In this work, the authors combine results on higher-form symmetries with the perspective that quantum gravity cannot contain global symmetries into a compelling story. In line with recent toy models on holography and ensemble averaging, they study a pair of oppositely oriented Chern-Simons theories. These have a $Z_k \times Z_{k}$ one-form global symmetry, and the authors gauge a $Z_k$ subgroup of this symmetry. Their result is that with a single torus boundary, one gets a modular invariant result. Instead of summing over topology the projection onto the gauge-invariant sector produces this sum. Higher topologies become trivial and should not be summed over in this construction. The results are worked out in full technical detail, and address questions of current interest in gravity, its ensemble interpretation, and the factorization puzzles.

I have a couple of small questions:

  • The authors show how the different gaugeable Lagrangian subgroups lead to the different modular invariants of the boundary CFT. Is this relation one-to-one in general?

  • On p53 the authors briefly mention non-compact gauge groups and how that would be important for gravity. Is there anything already known on this? Is it known for instance whether the $SL(2,R) \times SL(2,R)$ model has a global 1-form symmetry?

  • Could one run the story in one dimension less? In the bulk, one would take a compact group BF model. The non-compact generalization $SL(2,R)$ would then correspond to JT gravity. It is not obvious what modular invariance would mean here, even though there is an ensemble interpretation given by Saad, Shenker and Stanford.

I would appreciate the authors giving some brief comments on these questions, but strongly recommend the paper for publication regardless.

  • validity: top
  • significance: top
  • originality: top
  • clarity: high
  • formatting: perfect
  • grammar: perfect

Author:  Lorenzo Di Pietro  on 2022-09-19  [id 2828]

(in reply to Report 1 on 2022-08-10)

Dear referee,

thank you for your report and for your interesting questions. Here are our answers:

  • We believe the answer is yes but we do not have a proof of this statement, that is why we did not comment explicitly on this in the paper;
  • In the paper we discussed the simple case of an abelian non-compact group. We do not have much to add for the case of SL(2,R), it is much harder to set up the problem because we do not have a non-perturbative definition that is under control in that case. Rather than the one-form symmetry, which certainly exist in the form of a Z_2 center symmetry, in this non-abelian case one would have to gauge the non-invertible symmetry given by a certain subset of the full set of bulk lines. It is not clear how to do this in the case of SL(2,R);
  • We believe there should be an analogous story in 2d, with BF theory as a starting point. To make contact with gravity one would have to address the differences between non-compact BF theory and JT gravity. This sounds like an interesting direction for the future but we do not have much to say about it at the moment.

---

## Round 2 · Referee Report · Anonymous (Referee 2) · 2022-8-17

Report

In recent years, a new holographic paradigm has emerged in which simple theories of gravity (primarily in low spacetime dimensions) are holographically dual not to particular quantum systems but rather to statistical ensembles of quantum systems. These are puzzling because the boundary theory does not obey the laws of quantum mechanics. A sharp signature of such averaged holographic dualities is non-factorization of the partition function on configurations with multiple disconnected boundaries. From the perspective of the bulk, non-factorization is typically attributed to the existence of spacetime wormhole solutions connecting multiple disjoint boundaries in semiclassical gravity; in general, the mechanism for factorization in consistent conventional holographic dualities is not well understood given the existence of such wormhole solutions. In this paper, the authors propose a novel mechanism for factorization in toy models of holography based on Chern-Simons theory that is intertwined with the absence of global symmetries in quantum gravity.

The toy models the authors consider are based on Chern-Simons theory in three spacetime dimensions. Recent bottom-up approaches to 3d quantum gravity involve fixing boundary conditions and summing the contributions of all smooth bulk geometries subject to those boundary conditions. The resulting sum over gravitational instantons typically suffers from pathologies that prevent a straightforward interpretation as a consistent theory of quantum gravity, and even the question of which bulk configurations ought to be included in the sum is in general not resolved. Indeed, it has recently been shown that the sum over hyperbolic three-manifolds in abelian Chern-Simons theory is precisely dual to an ensemble average of boundary free boson CFTs, and hence e.g. suffers from non-factorization.

The authors' insight in this paper is based on the observation that toy theories of 3d gravity based on Chern-Simons theory possess a global one-form symmetry, conflicting with the general expectation informed by black holes that there ought not to be global symmetries in quantum gravity. The authors propose to eliminate the global symmetry by gauging an appropriate maximal non-anomalous subgroup (in the non-abelian case the symmetry is non-invertible and this process is referred to as anyon condensation). The global symmetry is generated by the topological line operators in the theory. The gauged subgroup is generated by a particular subset of these line operators, and is not unique. The partition functions in the gauged theory are computed by summing over insertions of these distinguished lines on the homology cycles of the bulk manifold.

The authors ague that the effect of the gauging is to render the theory completely trivial in the bulk, in the sense that the partition function on any closed three-manifold is 1. For three-manifolds with asymptotic boundaries, the sum over lines has the effect of pairing the holomorphic and anti-holomorphic boundary conformal blocks in such a way as to compute a modular-invariant partition function coinciding with that of a specific rational CFT (that depends on the particular gauged subgroup) on the boundary. In particular, the resulting partition function is completely independent of the bulk topology, and hence there is no need to sum over geometries. Moreover, that the theory is trivial in the bulk ensures that partition functions with multiple asymptotic boundaries factorize, although the detailed mechanism is nontrivial. Similar effects have recently been observed in the context of string theory on $AdS_3\times S^3\times T^4$ supported by NSNS fluxes in the tensionless limit.

The authors first demonstrate this gauging procedure extensively and pedagogically in the context of $U(1)_k\times U(1)_{-k}$ CS theory, which upon gauging is dual to the free compact boson CFT in 2d. This example clearly exhibits most of the main ideas of the paper. They then briefly discuss some generalizations of their techniques within the realm of abelian CS theory.

Adapting their ideas to non-abelian CS theory requires the introduction of some mathematical machinery --- namely, anyon condensation in modular tensor categories --- which the authors devote some time to explaining clearly. Despite the heightened mathematical sophistication the main conceptual points remain essentially intact.

This is a beautiful paper that presents a genuinely novel mechanism for factorization, at least within the context of toy models for 3d gravity defined through CS theory with compact gauge group. It seems to be an urgent objective to understand to what extent their results can be adapated to non-compact CS theory, which is more directly related to Einstein gravity in 3d. The paper is well-written and its arguments are to the best of my knowledge sound. I enthusiastically recommend its publication in SciPost.

Requested changes

None needed.

---

## Editorial Decision

published